# Convexified Filtered ANN via Attribute-Vector Fusion

## Abstract

Vector search powers transformers technology, but real-world use demands hybrid queries that combine vector similarity with attribute filters (e.g., "top document in category X, from 2023"). Current solutions trade off recall, speed, and flexibility, relying on fragile index hacks that don't scale. We introduce fused-based ANN, a geometric framework that elevates filtering to ANN optimization constraints and introduces a convex fused space via a Lagrangian-like relaxation. Our method jointly embeds attributes and vectors through transformer-based convexification, turning hard filters into continuous, weighted penalties that preserve top-k semantics while enabling efficient approximate search. We prove that our fused method reduces to exact filtering under high selectivity, gracefully relaxes to semantically nearest attributes when exact matches are insufficient, and preserves downstream ANN -approximation guarantees. Empirically, fused-based method improves query throughput by eliminating brittle filtering stages, achieving superior recall–latency trade-offs on standard hybrid benchmarks without specialized index hacks, delivering up to $3\times$ higher throughput and better recall than state-of-the-art hybrid and graph-based systems. Theoretically, we provide explicit error bounds and parameter selection rules that make the fusion practical for production. This establishes a principled, scalable, and verifiable bridge between symbolic constraints and vector similarity, unlocking a new generation of filtered retrieval systems for large, hybrid, and dynamic NLP/ML workloads.

## 1 Introduction

The approximate nearest neighbor search (ANNS) is fundamental to many data science and AI applications, enabling efficient retrieval of similar vectors in high-dimensional spaces (Chen et al., 2021; Malkov & Yashunin, 2018; Subramanya et al., 2019b). However, real-world applications increasingly require hybrid queries that combine vector similarity with attribute constraints (Gollapudi et al., 2023; Wang et al., 2023; 2021a; Wei et al., 2020; Taipalus, 2024; Pinecone, 2021; Japan, 2016; Wu et al., 2022; Microsoft, 2020; Heidari et al., 2025b;a; Heidari & Zhang, 2025). These constraints typically appear as either exact filters (e.g., "images with tag 'sunset'") or range filters (e.g., "products priced between $20-$50") (Pan et al., 2024; Ren et al., 2020).

Existing approaches to hybrid queries can be categorized into three strategies: (1) Filter-first methods like AnalyticDB-V (Wei et al., 2020) and Weaviate (Taipalus, 2024), which use attribute information to narrow the search space before vector similarity search. (2) ANN-first methods such as NGT (Japan, 2016), Vearch (Jingdong, 2020), FAISS-IVF (Douze et al., 2024), and Pinecone (Pinecone, 2021), where vector search is performed before applying attribute filters. (3) Hybrid methods that integrate both filter and vector information into specialized index structures, including Filtered-DiskANN (Gollapudi et al., 2023), which uses a graph index with label-aware connections; NHQ (Wang et al., 2023), which builds a composite proximity graph with joint pruning; DEG (Yin et al., 2025), which performs hybrid similarity search by building a Pareto-pruned graph and using an weighted traversal to retrieve results along approximate Pareto frontiers; HQANN (Wu et al., 2022), which leverages attribute-guided navigation and fused search; as well as recent approaches like ACORN (Patel et al., 2024), NaviX (Sehgal & Salihoğlu, 2025), CAPS (Gupta et al., 2023), and Milvus (Wang et al., 2021a) in its advanced partitioning modes. Hybrid methods such as ACORN, NaviX, and CAPS employ predicate-aware or cost-aware partitioning schemes to jointly optimize filter and vector search, while modern versions of Milvus leverage offline data structures to partition vectors based on historical filter conditions, thus improving search efficiency under complex predicates. Range filters, which

constrain results to specified intervals of attribute values, present additional challenges compared to exact filters (Pan et al., 2024). Efficiently handling range constraints requires consideration of attribute continuity and potential overlap between ranges, which can lead to increased candidate set sizes and higher computational complexity. This is especially critical in real-world workloads, where range predicates are common and may be applied to high-cardinality or correlated attributes. As a result, designing hybrid query systems that support fast and scalable range filtering remains an open problem, with recent research exploring new geometric and algorithmic approaches to overcome these difficulties, including HM-ANN (Ren et al., 2020), which enables graph search for heterogeneous memory; SeRF (Zuo et al., 2024), which uses a compressed segment graph for ranges; iRangeGraph (Xu et al., 2025), which constructs elemental graphs for on-demand ranges; and UNIFY (Liang et al., 2024), which builds a unified segmented graph for all ranges.

Although these approaches involve different tradeoffs, they share a fundamental limitation: attribute filtering is treated as an auxiliary operation layered onto the vector search process or index structure, rather than as a transformation of the underlying data space itself. This paradigm imposes intrinsic performance bottlenecks, especially when supporting multiple attributes with varying priorities or adapting to shifts in attribute distributions. In particular, state-of-the-art methods typically forego direct use of the original data, instead constructing specialized index replicas tailored for hybrid queries. As a result, whether the transformation occurs at the data or index level is largely insignificant-further motivating a data-centric perspective for hybrid search.

To address these limitations, we present FUSEDANN, a hybrid query framework merging attribute filters with vector data at the representation level. FUSEDANN uses a filter-centric vector indexing method, a mathematically grounded transformation, that unifies attribute filtering with vector similarity search, analogous to introducing a Lagrange multiplier into a convex objective and fusion of information signals (Boyd & Vandenberghe, 2004; Heidari et al., 2024; 2020c; 2019). A unified space where: (1) the dimensionality remains unchanged, (2) the distance ordering of elements with identical attributes is preserved, and (3) a tunable parameter increases distances between differently attributed elements.

**Contributions.** Our key contributions are: **(I)** A general framework for hybrid queries compatible with existing ANN indexing algorithms (§3). **(II)** Support for multiple attributes with intuitive priority hierarchies (§4). **(III)** Efficient handling of range filters through geometric interpretation (§5). **(IV)** Comprehensive experimental evaluation demonstrating FUSEDANN's superior effectiveness, efficiency, and stability (§6). The theoretical analysis provided in the Appendix offers rigorous guarantees on FUSEDANN's performance characteristics, including precise bounds on transformation parameters and candidate set sizes required for specific error probabilities.

## 2 PRELIMINARIES

In this section, we present preliminaries; the main notations are summarized in Table 2.

**Definition 1** (**Record Set** $\mathcal{D}^{(\mathbb{F})}$)**.** *A record is an $\mathbb{F} + 1$-tuple vector $o_i^{(\mathbb{F})} = [v(o_i), f^{(1)}(o_i), \ldots, f^{(\mathbb{F})}(o_i)]$, where $v(o_i) \in \mathbb{R}^d$ is a content vector (e.g., from BERT (Devlin et al., 2019)), and $f^{(j)}(o_i) \in \mathbb{R}^{m_j}$ is the j-th attribute vector in a metric space, also from a neural network. The record set is $\mathcal{D}^{(\mathbb{F})} = \{o_1^{(\mathbb{F})}, \ldots, o_n^{(\mathbb{F})}\}$, containing n records, each consisting of a content vector of dimension d and $\mathbb{F}$ attribute vectors of dimensions $m_1, \ldots, m_F$. Let $\mathcal{X} = \{v(o_i) \mid o_i \in \mathcal{D}^{(\mathbb{F})}\}$ and, for each $j \in [1, \mathbb{F}]$, $\mathcal{F}_j = \{f^{(j)}(o_i) \mid o_i \in \mathcal{D}^{(\mathbb{F})}\}$.*

If $\mathbb{F} = 0$, we have the regular ANN setup. If $\mathbb{F} = 1$ (one attribute), we use $\mathcal{D}$ and $o_i$ instead of $\mathcal{D}^{(1)}$ and $o_i^{(1)}$.

**Example 1.** *Record sets can represent various data types, such as images or videos. For example, as illustrated in Fig. 1(a), each record may correspond to an image described by attributes such as "Tag", "Category", and "Date". By embedding both the images and the "Tag" in appropriate metric spaces, we obtain the record set $\mathcal{D}^{(1)}$ (or simply $\mathcal{D}$ because we only use one attribute $\mathbb{F} = 1$).*

Given a record $o \in \mathcal{D}^{(\mathbb{F})}$, its content vector $v(o) \in \mathcal{X}$ is represented as $v(o) = [v(o)[0], v(o)[1], \ldots, v(o)[d-1]]$, where $v(o)[i]$ denotes the $i$-th dimension. We primarily consider high-dimensional cases, where $d$ is typically in the hundreds or thousands. For any two records $o, r \in \mathcal{D}^{(\mathbb{F})}$, their similarity is commonly measured using a metric such as Euclidean

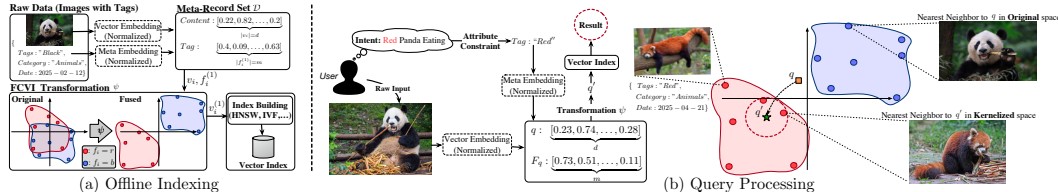

(a) Offline Indexing              (b) Query Processing

Figure 1: Data and queries are embedded into content and attribute vectors and fused by a transformation $\Psi$ parameterized by $\alpha > 1$ and $\beta > 1$. The fused vectors are indexed for efficient retrieval. At query time, the same transformation is applied, enabling unified search and re-ranking based on attribute-content similarity.

distance or cosine similarity. The Euclidean distance between their content vectors is defined as $\rho(v(o), v(r)) = \sqrt{\sum_{i=0}^{d-1}(v(o)[i] - v(r)[i])^2}$.

**Definition 2** (**Approximate Nearest Neighbor Search (ANNS)**). *Let $\mathcal{D}^{(\mathbb{F})}$ be a record set and $q$ a query with content vector $v(q)$. The exact $k$-nearest neighbors ($k$-NN) of $q$ in $\mathcal{D}^{(\mathbb{F})}$ with respect to the distance metric $\rho$ is defined as:*

$$\mathrm{NN}_k(q) = \arg\min_{S \subseteq \mathcal{D}^{(\mathbb{F})}, \, |S|=k} \sum_{o \in S} \rho\big(v(q), v(o)\big) \tag{1}$$

*Finding exact $k$-NN is computationally expensive in high-dimensional spaces (Abbasifard et al., 2014; Wang et al., 2021b). Therefore, approximate nearest neighbor search (ANNS) aims to efficiently return a set $\mathrm{ANN}_k(q)$ such that, with high probability,*

$$\max_{o \in \mathrm{ANN}_k(q)} \rho\big(v(q), v(o)\big) \leq (1 + \epsilon) \max_{o \in \mathrm{NN}_k(q)} \rho\big(v(q), v(o)\big), \tag{2}$$

*where $\epsilon > 0$ is the approximation factor. ANNS methods typically search based only on content vectors (Malkov & Yashunin, 2018; Douze et al., 2024; Bernhardsson, 2024).*

**Definition 3** (**Hybrid Query (HQ) with Monotone Attribute Priority**). *Given a set of records $\mathcal{D}^{(\mathbb{F})}$ and query $q = [v(q), F_q^{(1)}, \ldots, F_q^{(\mathbb{F})}]$, let $\mathcal{F}_{\pi(1)} \succ \cdots \succ \mathcal{F}_{\pi(\mathbb{F})}$ be the attribute priority order (with the content as the lowest priority). For any candidate set $S \subseteq \mathcal{D}^{(\mathbb{F})}$ of size $k$, let*

$$\mu_S^{(j)} = \frac{1}{k}\sum_{o \in S}\sigma_j(f^{(j)}(o), F_q^{(j)}), \qquad \mathrm{Var}_S^{(j)} = \frac{1}{k}\sum_{o \in S}\left[\sigma_j(f^{(j)}(o), F_q^{(j)}) - \mu_S^{(j)}\right]^2 \tag{3}$$

*where $\mu_S^{(j)}$ and $\mathrm{Var}_S^{(j)}$ denote the mean and variance of an attribute distance, and $\sigma_j$ is a distance metric on $\mathcal{F}_j$, which we assume to be Euclidean in this study. We say $S$ satisfies monotone attribute priority if: $\mathrm{Var}_S^{(\pi(1))} \leq \cdots \leq \mathrm{Var}_S^{(\pi(\mathbb{F}))}$. The hybrid query returns the set $S^*$ of size $k$ that minimizes the mean distances subject to monotone attribute priority:*

$$S^* = \arg\min_{\substack{S \subseteq \mathcal{D}^{(\mathbb{F})}, \, |S|=k \\ \text{monotone attribute priority}}} \left(\mu_S^{(\pi(1))}, \ldots, \mu_S^{(\pi(\mathbb{F}))}, \frac{1}{k}\sum_{o \in S}\rho(v(q), v(o))\right) \tag{4}$$

*in lexicographic order, i.e., by first minimizing the mean distance for the highest-priority attribute, then the next in order, and finally the average content distance $\rho$ (content vector distance metric). Eq. 4 relaxes filters, prioritizing exact matches on higher-priority attributes to fill $k$; if still short, it fills the rest via nearest attribute clusters ($k$-NN)—a native expansion unlike classical filtered ANN, which requires exact matches and offers no fallback. Users can still keep exact-only, as in Alg. 1.*

When $\mathbb{F} = 1$ (i.e., there is only one attribute), the hybrid query problem becomes a simplified version commonly considered in previous work (Chen et al., 2021; Gollapudi et al., 2023; Jingdong, 2020; Tan et al., 2023; Wang et al., 2021a; Wu et al., 2022; Heidari & Zhang, 2025; Xu et al., 2025; Zhu et al., 2020; Zuo et al., 2024).

## 3 FUSEDANN FRAMEWORK

**Model Overview.** Our framework enables hybrid search by fusing content and attribute information in a way that gives explicit control over their relative influence. Given a content vector $v(o_i) \in \mathbb{R}^d$ and

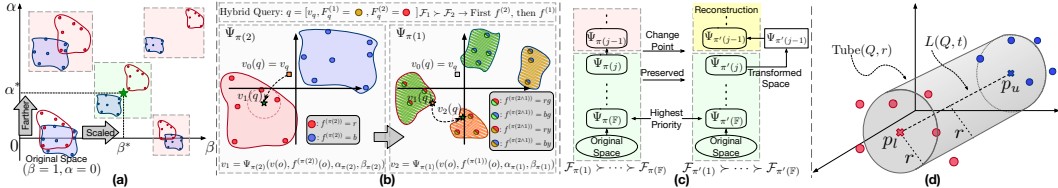

Figure 2: **(a):** The effect of $\alpha$ and $\beta$. **(b):** Multi-attribute iterative space overview. **(c):** Attribute or prority effect on our approach. **(d):** Range filter ANN analogy of cylinder

an attribute vector $f(o_i) \in \mathbb{R}^m$ with $m < d$, we partition $v(o_i)$ into $d/m$ blocks $v^{(1)}, \ldots, v^{(\lceil d/m \rceil)}$, each in $\mathbb{R}^m$. We then define the transformation:

$$\Psi(v, f, \alpha, \beta) = \left[ \frac{v^{(1)} - \alpha f}{\beta}, \ldots, \frac{v^{(\lceil d/m \rceil)} - \alpha f}{\beta} \right] \in \mathbb{R}^d \qquad (5)$$

where $\alpha > 1$ and $\beta > 1$ are scaling parameters. For exposition we assume $m \mid d$; otherwise, on the last block $v(\lceil d/m \rceil)$ we use the truncated attribute vector $f[: d - m\lfloor d/m \rfloor]$ to match dimensions. As illustrated in Fig. 1, the transformation is first applied to the data offline to build the index, and the same transformation is used online to process queries for retrieval (Fig. 1(b)). This creates a combined space that incorporates both the content representation and the attribute filters (such as Tag), where the attribute vectors can be generated using models like BERT (Devlin et al., 2019) or CLIP (Radford et al., 2021) to embed tags into a metric space before integration into the content vector (For a numerical example, see §B).

The parameter $\alpha$ increases the separation between records with different attribute values, while $\beta$ compresses all distances to regularize the fused space (see Fig. 2(a)). In practice, $\alpha$ and $\beta$ should be chosen large enough to ensure sufficient separation and regularization, but not so large as to introduce unnecessary computational complexity. Alg. 1 shows the building of the fused space and index, query generation, and result processing; as in line 15, we choose to include approximate attributes or only exact ones.

At query time, the query $q = [v(q), F_q]$ (content $v(q)$ and attribute $F_q$) is transformed as $q' = \Psi(v(q), F_q, \alpha, \beta)$.

For each attribute $a$, we define $R_a$ as the radius of the smallest hypersphere containing all transformed records with attribute $a$, $d_{min}(a, b)$ as the minimum distance between records with attributes $a$ and $b$, and $\gamma_a = \min_{b \neq a} \frac{d_{min}(a,b)}{R_a} - 1$ as the cluster separation metric. $N_a$ denotes the number of records with attribute $a$, and $N$ is the total number of records. These statistics are used to determine the optimal candidate set size $k'$(line 11) when processing queries; we pick the smallest $k'$ satisfying the candidate-size guarantee so the true top-$k$ are covered in the fused-space region, with $k' \geq k$.

Our theoretical analysis (detailed in the Supplementary Material §E) proves that the transformation $\Psi$ has several key properties: (i) it preserves the order of k-NN within clusters of records with identical attributes, enabling accurate content-based ranking within attribute groups; (ii) it increases separation between records with different attributes proportionally to $\alpha$, improving filtering effectiveness; and (iii) it scales all distances by $1/\beta$, control-

---

**Algorithm 1** Single-Attribute Hybrid Vector Indexing (FusedANN)

1: **[Offline Indexing] Require:** Dataset $\mathcal{D}$, Optimal parameters $\alpha > 1$, $\beta > 1$
2: **for** each $o_i$ in $\mathcal{D}$ **do**
3:     Partition $v(o_i)$ into $v^{(1)}, \ldots, v^{(d/m)}$
4:     Transform using given parameters $\alpha$, $\beta$: $v_i' = \Psi(v(o_i), f(o_i), \alpha, \beta) = \left[ \frac{v^{(1)} - \alpha f}{\beta}, \ldots, \frac{v^{(d/m)} - \alpha f}{\beta} \right]$
5:     Add $v_i'$ to index, retaining reference to $o_i$
6: **end for**
7: Precompute for each attribute $a$: radius $R_a$, minimum inter-cluster distance $d_{min}(a, b)$, and cluster separation metric $\gamma_a = \min_{b \neq a} \frac{d_{min}(a,b)}{R_a} - 1$
8: **[Online Query Processing] Require:** Query $q = [v(q), F_q]$, $k$, $\alpha$, $\beta$, error probability $\epsilon$, Boolean $AttrApprox$
9: Partition $v(q)$ into $v_q^{(1)}, \ldots, v_q^{(d/m)}$
10: Transform: $q' = \Psi(v(q), F_q, \alpha, \beta)$
11: Compute $k'$ (Thm. 2)
12: Retrieve top-$k'$ candidates from index using $q'$
13: **for** each candidate $o_i$ **do**
14:     Compute attribute distance: $s_f = \sigma(f(o_i), F_q)$
15:     **if** $AttrApprox$ = False AND $s_f \neq 0$ **then**
16:         continue;
17:     **end if**
18:     Compute content distance: $s_v = \rho(v(o_i), v(q))$
19:     Compute combined score: $\text{score}(o_i) = \alpha s_f + \beta s_v$
20: **end for**
21: Sort candidates by score and return top-$k$

---

ling overall concentration. These properties enable principled parameter selection: $\alpha$ should satisfy $\alpha > \frac{\beta \cdot \delta_{max}}{\sigma_{min} \cdot \sqrt{d/m}} \cdot (1 + \frac{\epsilon_f \cdot \beta}{\delta_{max}})$ where $\delta_{max}$ is the maximum content distance and $\sigma_{min}$ is the minimum attribute distance, while $\beta > \frac{\delta_{max}}{\epsilon_f}$ ensures intra-cluster distances are bounded by $\epsilon_f$ (Thm. 4). The optimal values for $\alpha$ and $\beta$ involves setting inequalities to equal (Cor. 1). The formula for $k'$ handles special cases like single-record attributes and identical-content records within an attribute ($R_a = 0$), providing probabilistic guarantees for retrieving the true top-k results.

**Complexity.** The offline phase requires $O(Nd)$ time to transform the dataset of $N$ records with $d$-dimensional vectors, plus $O(|\mathcal{F}|^2 N)$ time to compute cluster statistics where $|\mathcal{F}|$ is the number of distinct attributes. The storage overhead is $O(N)$ for vectors plus $O(|\mathcal{F}|^2)$ for cluster statistics, and we do not duplicate base vectors: the index stores only the transformed vector and a pointer to the original record. For online processing, query transformation takes $O(d)$ time, followed by $O(k' \log N)$ time for retrieving candidates and $O(k'd)$ time for pointer-only re-ranking. As $\alpha$ increases, the required $k'$ approaches $k$, so post-processing becomes akin to standard ANN re-ranking, minimizing overhead. This makes FUSEDANN efficient for practical applications where the number of distinct attributes is much smaller than the dataset size.

## 4 MULTI-ATTRIBUTES AND ATTRIBUTE HIERARCHY

Real-world search scenarios often involve multiple filtering attributes, each with different levels of importance (Def. 3). For example, an e-commerce platform might prioritize matching product categories first, then brands, and finally price ranges. In this section, we extend FUSEDANN to elegantly handle multiple attributes by applying our transformation sequentially, creating a natural hierarchy that controls their relative importance. In this section, we also assume $\forall j \in [1, \mathbb{F}] : m_j \mid d$.

**Recursive Transformations.** The key insight of our approach is remarkably simple: by applying the transformation $\Psi$ repeatedly for each attribute, we create a unified space that respects attribute priorities. Starting with the original content vector $v_0 = v(o_i)$, we apply each transformation in sequence:

$$v_j = \Psi_j(v_{j-1}, f^{(j)}(o_i), \alpha_j, \beta_j) \quad \text{for } j = 1, 2, \ldots, \mathbb{F} \tag{6}$$

where each transformation uses its own parameters $\alpha_j > 1$ and $\beta_j > 1$. The final transformed vector $v_{\mathbb{F}}$ integrates information from all attributes. As illustrated in Fig. 2(b), this process can be visualized with a simple example using two attributes, where each transformation progressively incorporates attribute information to refine the grouping of records.

This sequential approach creates a natural priority structure with three powerful properties (formally proven in the appendix). First, the order of elements with identical attributes is preserved through all transformations, ensuring that content-based ranking remains accurate within attribute-matched groups (Thm. 6).

Second, and crucially, the order of transformation application establishes a clear priority hierarchy: the later an attribute is applied, the higher its effective priority in determining the vector space structure (Thm. 7). As shown in Fig. 2(b), the transformation of the attribute with lower priority, $\pi(2)$, is applied first, followed by the higher-priority attribute, $\pi(1)$, resulting in the desired hierarchical organization. This occurs because later transformations' effects are scaled by fewer $\beta$ factors, giving them greater influence on the final distances. We show that when transformations are applied in reverse priority order, the resulting space inherently satisfies the monotone attribute priority property defined in Def. 3 (Thm. 8).

Third, our framework creates a natural stratification of records based on how many attributes match the query (Thm. 9). Records with more matching attributes will always be closer to the query than those with fewer matches, regardless of the content similarity. This creates well-defined "layers" in the vector space, with the innermost layer containing records matching all attributes, the next layer containing those matching all but one, and so on. Moreover, there always exist suitable transformation settings such that this attribute matching hierarchy holds for all cross-clusters pair of records (Thm. 10).

**Example 2.** *Imagine a product catalog with transformations applied in the order* $(f^{(color)}, f^{(size)}, f^{(brand)})$. *This makes brand the highest-priority attribute, followed by size, and then color. When searching, the retrieved products primarily match the brand specified in the query,*

*then the size, and finally the color. Additionally, products are ranked by content similarity. For more intuition, see §F.2.*

For multi-attribute retrieval, we extend Alg. 1 to apply transformations sequentially for each attribute (see Alg. 3). The key differences are: (1) transformations are applied iteratively as $v_j \leftarrow \Psi_j(v_{j-1}, f^{(j)}(o), \alpha_j, \beta_j)$ over all records for each attribute $j \in \{1, \ldots, \mathbb{F}\}$; (2) the optimal parameters $\alpha$ and $\beta$ of subsequent fused space is computed for each new attribute transformation iteratively; and (3) the candidate set size $k'$ is determined using Thm. 11, reflecting the narrowing effect of multiple filters (§F.6).

Time complexity remains $O(Nd)$ for preprocessing transformations, though computing statistics for all attribute combinations increases with the number of attributes. The query transformation is efficient at $O(\mathbb{F}d)$ time. Importantly, as $\mathbb{F}$ increases, fewer candidates are typically needed due to better separation in the transformed space, improving search efficiency..

### 4.1 ATTRIBUTE UPDATES IN FUSEDANN

Real-world applications often need to add new attributes (as metadata) or change priority orderings as requirements evolve. FUSEDANN handles these scenarios efficiently without requiring complete index reconstruction. When adding a new attribute with the highest priority, we simply apply an additional transformation to the already transformed vectors. For attributes inserted at lower priorities, a partial reconstruction is needed, but only from the insertion point forward. Similarly, when the priority orderings change, we need only to recompute transformations beyond the point where the old and new orderings differ: $j = \min\{k : \forall i \geq k, \pi(i) = \pi'(i)\}$. This limits the computational complexity to $O(N.j.d)$, substantially lower than the full recomputation, since only a partial reconstruction is required for indexes lower than $j$ (see Fig. 2(c)). This update efficiency makes FUSEDANN particularly well-suited for dynamic applications where attribute importance evolves over time, such as in recommendation systems where feature relevance changes based on user behavior. Note that standard data updates (e.g., new attribute values) are handled via native ANN insertion; the reconstruction bound $O(N \cdot j \cdot d)$ applies only to infrequent global re-prioritizations or when a new attribute field is added to the filtered index. See §F.7 for analysis.

## 5 RANGE FILTER ON FUSEDANN

Range queries seek records whose attribute values fall within a specified attribute range $[l, u]$, ranked by similarity to a query vector $q$. Formally, a range query is $Q = (q, l, u)$ where $q \in \mathbb{R}^d$ and $l, u \in \mathbb{R}^m$. Our fused space has an elegant geometric characteristics that allows us instead of indexing the points and then create feasible range at the runtime, index range queries and approximate nearest range query at runtime. A range query can be defined as a cylinder in the fused space that precisely captures all potential eligible nearest points to $q$ within $[l, u]$. The axis of the cylinder (*line segment*) obtains by $\Psi$ to the boundaries: $p_l := \Psi(q, l, \alpha, \beta)$, $p_u := \Psi(q, u, \alpha, \beta)$ parameterized as $L(Q, t) = (1 - t) \cdot p_l + t \cdot p_u$, where $t \in [0, 1]$ (or $L_Q$ in short).

For attribute values $f \in [l, u]$ in range-filtered query, the transformed query points $p_f := \Psi(q, f, \alpha, \beta)$ lie exactly on $L_Q$ in the fused embedding space (Thm. 15). Moreover, the vertical distance from $L_Q$ measures how well $q$ is approximated and scales with its vector similarity. Geometrically, each range-filtered query maps to a cylinder in the fused space. This relationship offers a unified geometric framework for jointly handling attribute range filtering and vector similarity. Formally, if $v \neq q$, the transformation of $v_f = \Psi(v, f, \alpha, \beta)$ vertical distance to $L_Q$ is exactly $\frac{\|v-q\|}{\beta}$ (Thm. 16). Leveraging this traceability, we define a cylindrical range query by introducing a radius-$r$ query cylinder around $L_Q$: $\mathbf{Tube}(\mathbf{Q}, \mathbf{r}) = \{z \in \mathbb{R}^d \mid \min_{t \in [0,1]} \|z - L(Q, t)\| \leq r\}$ (see Fig. 2(d)).

During indexing, we create cylinders that cover the fused space with an optimal radius $r = R$. This radius—ensuring high-probability top-$k'$ recall—is precomputed for each indexed line segment using the $k'$-th neighbor distance, dataset size, and similarity distribution. (Thm. 18). To efficiently cover the fused space with cylinders defined by pairs of offline data, which is crucial for fast retrieval, we use an adaptive sampling strategy during indexing over the fused space. At query time, for a top-$k$ query $Q' = (q', l', u')$ (where $k < k'$), we must find the nearest indexed cylindrical Tube$(Q, R)$ with Hausdorff distance closest axis $L_Q$ to $L_{Q'}$. The gap between $k$ and $k'$ guarantees high recall by providing the flexibility needed to approximate $L_{Q'}$ with Tube$(Q, R)$. The base radius $R$ is stored

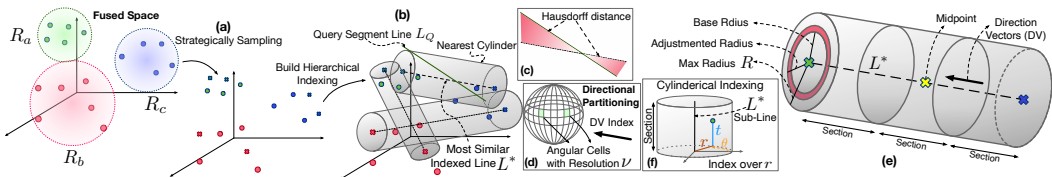

Figure 3: Hierarchical Indexing Components

with each line in the index and determines the extent of the corresponding cylindrical region or the maximum radius coverage.

**Hierarchical Indexing Framework.** We briefly sketch the idea here; details appear in §G. To enable efficient range queries in the fused space, we introduce a hierarchical framework of three levels in Alg. 2.

① Leveraging the guaranteed sample complexity, which is derived from the data pattern and range distributions (Livshits et al., 2020; Heidari et al., 2020b), we strategically take a **sufficiently large sample** from the space of possible range queries (Fig. 3(a) and Thm. 17). This approach ensures that any potential query line will closely match a pre-indexed line, while minimizing storage (see §G.3) and accounting for the varying importance of regions in the fused space (Alg. 4).

② We build a specialized **line similarity index** that efficiently identifies the pre-indexed line most similar to $L_Q$ (Fig. 3(b)). Our line similarity combines directional, positional, and length components to provide strong correlation with the Hausdorff distance between lines (See Fig. 3(c) and Thm. 19). Note that ANN indexing of a finite $L_Q$ within the cylinder defining the approximate range query differs from the approximate line nearest-neighbor methods (Andoni et al., 2009), which assume infinite lines. Our line index organizes lines first by their direction vectors and then by their spatial locations. The directional partitioning creates angular cells on the *unit sphere with resolution $\nu$* (Fig. 3(d,e)), assigning each line to a cell based on its orientation. Within each directional group, we further organize the lines using spatial indices based on their midpoints (Alg. 5). This hierarchical structure enables logarithmic time retrieval of the indexed line most similar to any query line (Alg. 6).

**Algorithm 2** Concise version of Alg. 10

1: **Input:** Query $q$, range $[l, u]$, $k$
2: Map $q$, $[l, u]$ to line $L_Q$ in fused space
3: Find most similar indexed line $L^*$ to $L_Q$ using line index (Fig. 3(b))
4: Adjust search radius based on line similarity (Fig. 3(c,e))
5: Retrieve candidate points from $L^*$'s cylindrical index within radius (Fig. 3(d,f))
6: Filter candidates by attribute range $[l, u]$
7: **Return** top-$k$ nearest neighbors to $q$

③ For each indexed line $L^*$, we construct a **cylindrical index** (Kim et al., 2001) that partitions the points by their cylindrical coordinates relative to $L^*$, allowing efficient retrieval of the most similar points (Fig. 3(f)). Every line segment is divided into sections the length of the radius (sub-lines), utilizing radius-based indices per section (with respect to the perpendicular distance to its respective section line segment) in a ball tree structure. This supports rapid retrieval of points within a certain range, while reducing excess calculations (Fig. 3(e)).

**Adaptive Error Compensation.** When approximating a query line with a similar indexed line, adjust the search radius and candidate count to offset the error per the lines' Hausdorff distance (Thm. 20). Specifically, the radius of the cylinder increases with the Hausdorff distance between the query and the indexed lines, and the candidate count is scaled by a data-dependent factor reflecting local line density (Alg. 9). The density is calculated by taking the ratio of the number of points contained within a cylindrical region to the volume of that area. Therefore, in areas of higher density, more candidates must be considered to maintain an equivalent probability of identifying the actual nearest neighbors (Thm. 21). Thus, when building the index, we assume a maximum Hausdorff distance supported by the pre-indexed data, add it to the optimal radius, and then construct the index. At query time, if the radius required for the query is below the maximum $R$, we apply these adjusted values of $k$ and the search radius to ensure robust retrieval performance (Fig. 3(e)).

**Complete Range Query Algorithm and Complexity.** Our range query processing first transforms the query into a line segment in the fused space, then efficiently locates the most similar indexed line via a hierarchical line index in logarithmic time. The search radius and the candidate count

Table 1: Dataset statistics

| Dataset | Dimension | Size | Use Case |
|---|---|---|---|
| SIFT1M | 128 | 1,000,000 | Single/Multi Filter |
| GloVe | 100 | 1,183,514 | Single/Multi Filter |
| UQ-V | 256 | 1,000,000 | Single/Multi/Range Filter |
| DEEP | 96 | 10,000,000 | Single Filter/Range Filter |
| YouTube-Audio | 128 | 1,000,000 | Single Filter/Range Filter |
| WIT-Image | 2048 | 1,000,000 | Single Filter/Range Filter |

are adjusted based on the Hausdorff distance between the query and the indexed lines. A cylinder search retrieves candidates within the adjusted radius, which are then filtered by attribute range and ranked by distance to the query. Alg. 2 achieves $O(\log N + k \log(1/\epsilon) + k \log k)$ expected query time, enabling efficient range queries even on very large datasets (Thm. 22).

## 6 EXPERIMENTS

We evaluated FUSEDANN on multiple real-world data sets that cover various retrieval scenarios: single-attribute filtering, multiple-attribute filtering, and range filtering. We compare against state-of-the-art methods from the recent literature. (Detailed experiments are provided in §D)

**Experimental Setup.** For a detailed setup, see §D.1.

- **Datasets.** We use datasets from different domains with varying dimensionality, as shown in Table 1. For single and multi-attribute filtering, we use SIFT1M[1], GloVe[2], and UQ-V[3]. For range filtering, we use DEEP[4], YouTube-Audio[5], and WIT-Image[6] (Zuo et al., 2024; Xu et al., 2025).

- **Variants of FUSEDANN.** We created four different versions of FUSEDANN, each incorporating a unique base indexing algorithm: FUS-H is built upon HNSW (Malkov & Yashunin, 2018); FUS-D uses DiskANN (Subramanya et al., 2019a); FUS-F employs Faiss (Johnson et al., 2019) with the IVF index; and FUS-A implements ANNOY (Bernhardsson, 2024).

- **Baselines.** For attribute filtering, we compare against: NHQ-NPG (Wang et al., 2023), Vearch (Jingdong, 2020), ACORN (Patel et al., 2024), VBASE (Zhang et al., 2023), ADBV (Zhu et al., 2020), Milvus (Wang et al., 2021a), Faiss (Johnson et al., 2019), DEG (Yin et al., 2025), SPTAG (Microsoft, 2020), NGT (Japan, 2016), and Filtered-DiskANN (Gollapudi et al., 2023) (F-Disk in short). For range filtering, we compare against: SeRF (Zuo et al., 2024), ANNS-first, Range-first, and FAISS (Johnson et al., 2019).

- **Metrics.** We use queries-per-second (QPS) for efficiency and Recall@k for accuracy. For all experiments, we report the mean over three runs.

**Single Attribute Filtering.** We evaluated FUSEDANN variants against 11 baseline methods (NHQ, Faiss, Vearch, SPTAG, ADBV, NGT, Milvus, Filtered-DiskANN) under single-attribute constraints. Fig. 4I demonstrates consistent superiority in both SIFT1M and GloVe datasets, where Fus-H achieves peak performance with $4.2\times$ higher QPS than NHQ-NPG at Recall@10=0.95. The performance hierarchy (Fus-H > Fus-D > Fus-F > Fus-A) mirrors the efficiency characteristics of their underlying index structures. In particular, Fus-H maintains 2.1-3.8× speed advantages over graph-based methods (NGT, SPTAG) and 1.8-2.4× improvements versus quantization approaches (Faiss, F-Disk) at all recall levels. This universal outperformance confirms the effectiveness of our distance-preserving transformation in maintaining relevant vector proximities while enforcing attribute constraints.

**Multiple Attribute Filtering.** Fig. 4II evaluates multi-attribute filtering performance across SIFT1M and GloVe datasets, comparing variants FUSEDANN against six baselines (NHQ, Faiss,

---

[1] http://corpus-texmex.irisa.fr/

[2] https://nlp.stanford.edu/projects/glove/

[3] https://dataset.uq-v.org/

[4] https://research.yandex.com/blog/benchmarks-for-billion-scale-similarity-search

[5] https://research.google.com/youtube8m/download.html

[6] https://github.com/google-research-datasets/wit

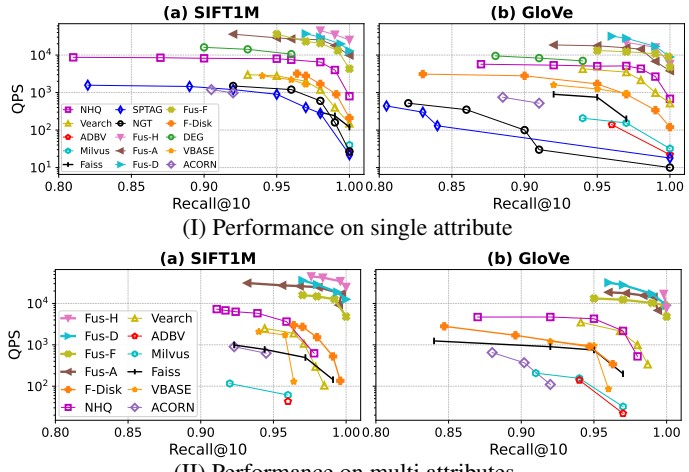

(I) Performance on single attribute

(II) Performance on multi attributes

Vearch, ADBV, Milvus and F-Disk). Fus-H achieves a QPS 3.2× higher than NHQ at Recall@10=0.95, with consistent superiority in all variants following the same hierarchy. This performance ordering mirrors the efficiency characteristics of each variant's foundational index structure while maintaining attribute-aware separation. The cross-dataset improvements (2.1-3.8× over graph indexes, 1.6-2.9× versus quantization methods) confirm multi-attribute filtering's enhanced discriminative power between attribute-defined clusters.

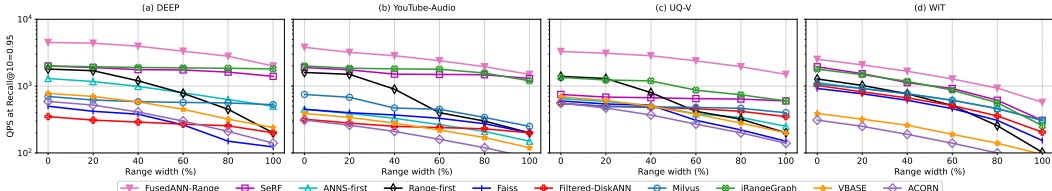

Figure 5: Range Performance

**Range Filtering.** We evaluate FUSEDANN-Range across the entire spectrum of range widths (0%–100%) on four benchmark datasets. As shown in Fig. 5, FUSEDANN-Range maintains superior QPS compared to seven state-of-the-art methods, particularly excelling at narrow ranges (<20%) where it outperforms SeRF by 3.8–5.6× and ANNS-first by 7.2–12.9× at Recall@10=0.95. Although our hierarchical indexing strategy optimizes FUSEDANN-Range for range filtering, the core transformation principles remain applicable to other index types. Consistent performance advantages across DEEP, YouTube-Audio, UQ-V, and WIT datasets demonstrate both robustness and versatility. At full width, FUSEDANN becomes content-only search (Thm. 16), matching raw index speed and avoiding graph filter traversal. On the 2048-D WIT dataset, it embeds range constraints directly with little extra distance cost, while graph filters face high-dimensional pruning overhead.

**Ablation Studies.** The complete Fus-H system achieves 43,618 QPS at Recall@10=0.95. Individual component removal reveals distinct contributions: $\alpha$ effect removal reduces performance to 28,149 QPS (35% drop), $\beta$ removal to 30,968 QPS (31% drop), parameter setting removal to 23,210 QPS (47% drop), and candidate optimization($k'$) removal to 23,127 QPS (47% drop). This confirms each component's importance to our method's effectiveness. We further examine the performance difference between FUSEDANN variants, finding that the underlying index algorithm contributes significantly to the overall performance, with the core transformation providing a consistent boost regardless of the base index used.

**Scalability.** When increasing the number of attribute constraints from 1 to 3, FUSEDANN variants sustain high throughput: the top variant remains close to $10^5$ QPS throughout, while others stay above $3 \times 10^4$ QPS. In contrast, baseline methods drop sharply, with some falling below $10^3$ QPS at three attributes. This analysis shows that FUSEDANN maintains robust efficiency under increasing filter complexity, outperforming alternatives by 10× to 100× as the number of attributes grows.

## 7 DISCUSSION

As it shown in §3, our method needs filter values be embedded in a metric space, with attribute dimension $m \leq d$ to ensure fusion and ANN compatibility; for scalar attributes we use $m = 1$. We do not natively support arbitrary DNF—by a metric-embedding no-go (OR cannot be fused into a single-valued metric without violating axioms)—but we rapidly materialize the conjunctive DNF building blocks (filters $\wedge$ range/ANN) and handle unions/negations via query planning; We pre-materialize conjunctive blocks and use a light planner for OR/NOT, probing few shared blocks and merging $O(k')$ candidates with $O(k \log k)$ dedup; as $\alpha$ grows, $k' \rightarrow k$, so overhead stays near top-$k$. In contrast, general systems (e.g., ACORN and NaviX) defer filtering to query time, hurting performance. FusedANN functions as a specialized, predicate-aware index for high-value filtering paths, complementing the general-purpose base index.

Although we have discussed supporting updates that add entirely new attributes to all records, a theoretical analysis is still needed to understand how changes in the value of a single attribute impact the index structure, query performance, and when such updates should trigger index reconstruction, similar to the approach in (Mohoney et al., 2024). Addressing the scalability to multi-attribute range queries, theoretical guarantees for a mixture of multiple attributes with one range attribute, general guarantees for Non-Euclidean metrics, and efficient attribute updates remains an important direction for future work. See §C for detailed limits and future work.

## 8 CONCLUSION

We propose a geometric hybrid search framework that unifies content and attribute information in a fixed-dimensional space, allowing efficient filtering and range queries without modifying existing ANN indexing algorithms. Our transformation preserves nearest-neighbor ordering within attribute classes, supports dynamic attribute priorities, and allows efficient partial index updates. In addition, it works with categorical and unstructured attribute values. Extensive experiments on real-world datasets demonstrate that FUSEDANN achieves superior recall and query throughput compared to state-of-the-art hybrid methods, especially under complex or multi-attribute filtering. Theoretically, we provide explicit error bounds and principled parameter selection rules, ensuring robust performance and practical deployment. Our results indicate that geometric fusion of attributes and vectors offers a scalable and flexible foundation for next-generation hybrid retrieval systems.

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

APPENDIX CONTENTS

# A  TABLE OF NOTATIONS

Table 2: Summary of Notation Used in this Paper

| Name | Symbol | Definition |
|---|---|---|
| Number of attributes | $\mathbb{F}$ | Number of attribute constraints (filters) in hybrid queries. |
| Record set | $\mathcal{D}^{(\mathbb{F})}$ | Set of all records: $\{o_1^{(\mathbb{F})}, \ldots, o_n^{(\mathbb{F})}\}$, each with content and $\mathbb{F}$ attributes. |
| Record | $o_i^{(\mathbb{F})}$ | $i$-th record: $[v(o_i), f^{(1)}(o_i), \ldots, f^{(\mathbb{F})}(o_i)]$. |
| Content vector | $v(o_i)$ | Main content embedding of $o_i$; $v(o_i) \in \mathbb{R}^d$. |
| Content vector set | $\mathcal{X}$ | $\{v(o_i) \mid o_i \in \mathcal{D}^{(\mathbb{F})}\}$. |
| Content vector dimension | $d$ | Dimension of content vectors $v(o_i)$. |
| Attribute vector (single) | $f(o_i)$ | Attribute embedding (single-attribute case), $f(o_i) \in \mathbb{R}^m$. |
| Attribute vector for $j$ | $f^{(j)}(o_i)$ | $j$-th attribute vector for $o_i$; $f^{(j)}(o_i) \in \mathbb{R}^{m_j}$. |
| Attribute vector dimension | $m, m_j$ | Dimension of attribute vector(s): $m$ for single-attribute, $m_j$ for $j$-th attribute. |
| Attribute value set | $\mathcal{F}_j$ | Set of all possible values for attribute $j$: $\{f^{(j)}(o_i)\}$ over all $i$. |
| Set of all attribute combinations | $\mathcal{F}$ | Set of all unique attribute value combinations (multi-attribute). |
| Query | $q$ | Query, typically $q = [v(q), F_q^{(1)}, ..., F_q^{(\mathbb{F})}]$. |
| Query content vector | $v(q)$ | Content vector of the query. |
| Query attribute ($j$) | $F_q^{(j)}$ | Value of the $j$-th attribute for the query. |
| Distance metric (content) | $\rho(x, y)$ | Distance function (usually Euclidean) on content vectors. |
| Distance metric (attribute $j$) | $\sigma_j(x, y)$ | Distance function (usually Euclidean) for attribute $j$. |
| Approximation factor | $\epsilon$ | Relative error for approximate nearest neighbor search. |
| Cluster tightness parameter | $\epsilon_f$ | Upper bound on intra-cluster (same-attribute) fused vector distances. |
| Transformed vector | $v_i'$ | Fused vector: $v_i' = \Psi(v(o_i), f(o_i), \alpha, \beta)$. |
| Fused transformation | $\Psi(v, f, \alpha, \beta)$ | Transformation combining content and attribute: block-wise, see Eq. (3). |
| Multi-attribute transformation | $\Psi_j(\cdot)$ | $j$-th transformation in sequence for multi-attribute fusion. |
| Transformation scaling | $\alpha, \alpha_j$ | Controls attribute separation in fused space; larger $\alpha$ increases separation. |
| Transformation scaling | $\beta, \beta_j$ | Scales (compresses) all distances in fused space. |
| Block partitioning | $v^{(l)}$ | $l$-th block of $v(o_i)$ when partitioning into blocks of size $m$ ($v^{(l)} \in \mathbb{R}^m$). |
| Number of blocks | $d/m$ | Number of blocks when dividing $v(o_i) \in \mathbb{R}^d$ into blocks of length $m$. |

**Table 2 – continued from previous page**

| Name | Symbol | Definition |
|------|--------|------------|
| $k$-nearest neighbors | $\mathrm{NN}_k(q)$ | Exact top $k$ nearest neighbors of query $q$. |
| Approximate neighbors | $\mathrm{ANN}_k(q)$ | Approximate top $k$ nearest neighbors (may allow error $\epsilon$). |
| Number of candidates | $k'$ | Number of candidates retrieved in fused space for high-recall guarantee. |
| Candidate cluster radius | $R_a$ | Radius of smallest hypersphere containing all transformed records with attribute $a$. |
| Minimum inter-cluster dist. | $d_{min}(a,b)$ | Minimum distance between any points in clusters for attributes $a$ and $b$. |
| Cluster separation metric | $\gamma_a$ | Normalized separation: $\gamma_a = \min_{b \neq a} \frac{d_{min}(a,b)}{R_a} - 1$. |
| Number in attribute cluster | $N_a$ | Number of records with attribute $a$. |
| Attribute combination | $\vec{a}$ | Tuple of attribute values: $(a^{(1)}, ..., a^{(\mathbb{F})})$. |
| Number in attribute cluster (multi) | $N_{\vec{a}}$ | Number of records with attribute combination $\vec{a}$. |
| Cluster separation (multi) | $\gamma_{\vec{a}}$ | As above, for multi-attribute clusters. |
| Attribute priority order | $\pi$ | Permutation encoding the search priority of each attribute. |
| Permutation length | $|\pi|$ | Number of attributes in the priority order. |
| Variance in attribute distance | $\mathrm{Var}_S^{(j)}$ | Variance of attribute $j$'s distance in result set $S$. |
| Mean attribute distance | $\mu_S^{(j)}$ | Mean attribute $j$ distance in candidate set $S$. |
| Hybrid score | $\mathrm{score}(o_i)$ | Combined score (e.g., $\alpha s_f + \beta s_v$) for candidate ranking. |
| Cylinder (range query) | $\mathrm{Tube}(Q,r)$ | Set of points within perpendicular distance $r$ to query range line in fused space. |
| Range line (query) | $L(Q,t)$ | Line segment in fused space for attribute range $[l,u]$ and query $q$, $t \in [0,1]$. |
| Range endpoints (attributes) | $l, u$ | Lower and upper endpoints of attribute range filter. |
| Range line endpoint (fused) | $p_l, p_u$ | $\Psi(q,l,\alpha,\beta)$ and $\Psi(q,u,\alpha,\beta)$: endpoints in fused space. |
| Hausdorff distance | $d_H(A,B)$ | Maximum minimal distance between sets $A$ and $B$ (for line similarity). |
| Line similarity | $\mathrm{sim}(L_1, L_2)$ | Composite similarity metric for lines (direction, midpoint, length) for range queries. |
| Angular resolution parameter | $\nu$ | Granularity for direction partitioning in hierarchical line index. |
| Cylinder search radius | $r$ | Radius of cylinder around query line for range search. |
| Sampling resolution | $r_q, r_r$ | Resolution for sampling query and range spaces during line index construction. |

| Name | Symbol | Definition |
|------|--------|-----------|
| **Table 2 – continued from previous page** | | |
| Local density factor | $\eta$ | Estimated density of points near a given line segment (used for adaptive $k'$). |
| Number of indexed lines | $L$ | Number of pre-indexed line segments (for range queries). |
| Number of points in cylinder | $P$ | Number of points in a cylindrical index (for range queries). |
| Density estimation window | $N_r$ | Number of points within radius $r$ of a line segment. |
| Cylinder volume | $V_r$ | Volume of a cylinder with radius $r$ and given length: $V_r = \pi r^2 \cdot \|b - a\|$. |

# B    NUMERICAL EXAMPLE OF $\Psi$ TRANSFORMATION

$\Psi$ transformation in S.3 (Eq. 6) subtracts $\alpha f$ from each partitioned block of content vector $\mathbf{v}$ and then scales the result by $1/\beta$. We agree content and attribute vectors (e.g., image embeddings vs. BERT tag embeddings (Devlin et al., 2019)) encode distinct semantics, making direct subtraction seem counterintuitive. However, it's mathematically principled: it preserves intra-cluster NN ordering/distances up to $1/\beta$ scaling (Theorem 4 and Corrolary 1) while increasing inter-cluster separation via $\alpha$, fusing them geometrically without changing dimensionality or ANN compatibility (e.g., Faiss Douze et al. (2024)).

**Toy example ($d = 2$, $m = 1$).** Initial groups (on a circle, $r = 5$) with attribute $f$:

*Group A ($f = -3$):* $P_1 = (5.00, 0.00)$,    $P_2 = (-2.20, 4.33)$,    $P_3 = (-2.50, -4.33)$

*Group B ($f = +3$):* $Q_1 = (2.50, 4.33)$,    $Q_2 = (-5.00, 0.00)$,    $Q_3 = (2.50, -4.33)$,    $Q_4 = (3.54, 3.54)$

**Initial Euclidean distance ($\rho$) all from $P_1$:**

$$\rho(P_1, Q_1) \approx 5.00 \quad \rho(P_1, Q_2) \approx 10.00 \quad \rho(P_1, Q_3) \approx 5.00 \quad \rho(P_1, Q_4) \approx 3.83$$
$$\rho(P_1, P_2) \approx 8.41 \quad \rho(P_1, P_3) \approx 8.66$$

Top 2–NN: $Q_4, Q_1/Q_3$ (mixed groups).

**Applying $\Psi$ ($\alpha = 3$, $\beta = 1.5$):**

$$\mathbf{v}' = \frac{\mathbf{v} - \alpha f}{\beta}$$

*Group A:* since $f = -3$, we have $\mathbf{v}' = \dfrac{\mathbf{v} + 9}{1.5}$.

$P_1' = (9.33, 6.00)$,    $P_2' = (4.53, 8.89)$,    $P_3' = (4.33, 3.11)$

*Group B:* since $f = +3$, we have $\mathbf{v}' = \dfrac{\mathbf{v} - 9}{1.5}$.

$$Q_1' = (-4.33, -3.11), \quad Q_2' = (-9.33, -6.00)$$
$$Q_3' = (-4.33, -8.89), \quad Q_4' = (-3.64, -3.64)$$

**After transformation distances all from $P_1'$:**

$$\rho(P_1', Q_1') \approx 16.42 \quad \rho(P_1', Q_2') \approx 22.20 \quad \rho(P_1', Q_3') \approx 20.20 \quad \rho(P_1', Q_4') \approx 16.16$$
$$\rho(P_1', P_2') \approx 5.60 \quad \rho(P_1', P_3') \approx 5.77$$

Top 2–NN: $P_2'$, $P_3'$ (intra-group; order preserved, inter dropped).

This illustrates that $\Psi$ induces a uniform rescaling within groups (preserving intra-cluster NN relations up to $1/\beta$) while shifting group centers apart via $\alpha$, thereby enhancing inter-cluster separation without altering dimensionality or compatibility with standard ANN indices.

## C  EXTENDED LIMITATIONS AND FUTURE WORK

We summarize key limitations of our approach and outline concrete avenues for future research.

**Limitations.**   Although our fusion-based method is a promising approach for handling filters in an ANN problem, we recognize the following limitations.

- **Metric-embedding requirement.** Our method requires that filter values be embedded in a metric space with attribute dimension $m \leq d$ to ensure fusion with the base vector space and ANN compatibility. This constrains attribute types and encodings, and may limit applicability when attributes are non-metric or exceed the dimensional budget.
- **No-go for native DNF.** We do not natively support arbitrary DNF predicates, by an impossibility of embedding a metric (disjunctive OR cannot be fused into a single-valued metric without violating metric axioms). Instead, we efficiently materialize conjunctive building blocks multi-attributes filters, and range filter and perform unions/negations via query planning. This design favors predictable performance while avoiding the per-query penalties incurred by general systems (e.g., ACORN) that defer filtering to query time.
- **Update sensitivity and index maintenance.** Although we support appending entirely new attributes to all records, the effect of updates to the value of a single attribute on the fused index structure and query accuracy/latency remains theoretically under-analyzed. Determining when incremental updates suffice versus when index reconstruction is required is an open problem.
- **Non-Euclidean metrics.** Our guarantees are strongest under Euclidean assumptions. General theoretical guarantees under non-Euclidean (e.g., tree, graph, or learned) metrics remain incomplete.
- **Priority dynamics.** Our incremental handling of attribute-priority changes (by storing indexes of combinations of lower-priority attributes and extending them for higher priorities) enables flexible updates but can increase storage and maintenance costs under frequent re-prioritization.

**Future work.**   In future work, we will extend our approach by addressing current limitations and following some interesting paths.

- **Sampling framework for computing $\alpha^*$ and $\beta^*$.** The computation of $\alpha^*$ and $\beta^*$ uses all available data. A useful direction is to approximate them from a sample and analyze the sample complexity and the resulting approximation error for $\alpha^*$ and $\beta^*$.
- **Theory for update triggers and stability.** Develop formal criteria and bounds that predict when single-attribute updates degrade recall/latency enough to trigger partial or full index reconstruction, building on incremental indexing insights (e.g., Mohoney et al. (2024)).
- **Scalable multi-attribute range querying.** Design compact representations and pruning strategies to mitigate the $2^{\mathbb{F}}$ blow-up—e.g., lattice-aware caching, vertex sharing, monotone submodular planning, or compressed frontier enumeration for frequent ranges.
- **Robust query planning for Boolean compositions.**   Extend our planner to optimize unions/negations over efficiently materialized conjunctive blocks, including cost models that account for selectivity, overlap, and ANN recall, and adaptive plans that switch between early and late fusion based on observed statistics.
- **Learned and non-Euclidean embeddings.** Establish correctness and performance guarantees when attribute/value embeddings reside in non-Euclidean or learned spaces, including bi-Lipschitz bounds for fusion distortion and its impact on ANN recall.
- **Dynamic priority management.** Develop amortized bounds and storage-efficient data structures for priority shifts, including incremental index reuse, partial re-ranking layers, and lazy augmentation strategies with provable update/query trade-offs.
- **Attribute expansion with constraints.** Formalize when and how to add new attributes (or composed attributes) without violating the $m \leq d$ constraint, including techniques for joint dimensionality reduction that preserve both semantic and filter selectivity.

- **Hybrid exact–approximate execution.** Explore hybrid plans that mix pre-materialized conjunctive blocks with on-the-fly exact filtering for low-cardinality attributes, guided by selectivity-aware cost models to minimize end-to-end latency.
- **Benchmarks and stress tests.** Create public benchmarks for fused filtering+ANN workloads with controlled attribute skew, dynamics, and Boolean complexity, to standardize evaluation beyond simple conjunctive filters.

Overall, while a fundamental no-go theorem (Burago et al., 2001) prevents natively fusing disjunctive operators into a single metric, our approach provides a practical middle ground: fast construction of conjunctive building blocks, principled query planning for unions/negations, and compatibility with ANN. Closing the gaps in update theory, multi-attribute scalability, and non-Euclidean guarantees remains a promising direction toward a comprehensive, theoretically grounded system for filtered vector search.

# D EXTENDED EXPERIMENTS

This section provides a comprehensive experimental evaluation of FUSEDANN across different retrieval scenarios and datasets.

## D.1 EXPERIMENTAL SETUP

### D.1.1 DATASETS

We evaluate on six datasets spanning different domains (Table 3). For attribute and multi-attribute filtering, we use SIFT1M, GloVe, and UQ-V following NHQ (Wang et al., 2023). Each vector is augmented with synthetic attributes simulating real-world scenarios. For range filtering, we use DEEP, YouTube-Audio, and WIT-Image following (Zuo et al., 2024), with randomly assigned keys for DEEP and actual metadata (release time and image size) for the other two. UQ-V is included in both filtering categories as it contains both categorical attributes and numerical values suitable for range filtering.

Table 3: Detailed dataset statistics

| Dataset | Dimension | # Base | # Query | LID* |
|---|---|---|---|---|
| SIFT1M | 128 | 1,000,000 | 10,000 | 9.3 |
| GloVe | 100 | 1,183,514 | 10,000 | 20.0 |
| UQ-V | 256 | 1,000,000 | 10,000 | 14.7 |
| DEEP | 96 | 10,000,000 | 10,000 | 7.2 |
| YouTube-Audio | 128 | 1,000,000 | 10,000 | 9.5 |
| WIT-Image | 2048 | 1,000,000 | 1,000 | 11.7 |

*LID: Local Intrinsic Dimensionality (Fu et al., 2021)

**Categorical and range selectivity.** Following NHQ (Wang et al., 2023), we synthesize categorical attributes on SIFT1M, GloVe, and UQ-V. Attribute selectivity is controlled by cardinality $C$ and the number of attribute combinations $z \in \{36, 972, 26244\}$, spanning low-to-high selectivity regimes. For reproducibility, we report per-dataset $C$ and the implied per-attribute selectivity $1/C$. For range filtering following (Zuo et al., 2024), selectivity is equivalent to the range width (%).

### D.1.2 IMPLEMENTATION DETAILS

We implemented FUSEDANN in C++17 with Python bindings. 64-core high-performance CPU (3.0GHz base clock), 256GB DDR4 RAM, and a data center GPU with 40GB VRAM. For attributes embeddings, We used BERT (Devlin et al., 2019) to generate a metric space and applied PCA to reduce the vector dimension to $m = 10$, ensuring that each attribute vector receives a unique representation through this dimensionality reduction. For index construction, we used the parameters $\alpha = 10.0$, $\beta = 2.0$, resolution $\nu = \frac{\pi}{180}$, $\epsilon_f = 1.0$, $\epsilon = 10^{-2}$, $\delta = 5 \times 10^{-2}$ by default, with specific parameter configurations for each dataset determined via grid search. Each FUSEDANN variant uses the respective base index's implementation (HNSW, DiskANN, Faiss, ANNOY) with our transformation layer applied.

### D.1.3 Baselines

We compare against state-of-the-art methods in three categories:

**Single/Multi-Attribute Filtering:**

- NHQ-NPG (Wang et al., 2023): Native hybrid query with optimized proximity graphs
- Vearch (Jingdong, 2020): Vector search engine with filtering support
- ADBV (Zhu et al., 2020): Alibaba's cost-based hybrid query optimizer using IVFPQ
- Milvus (Wang et al., 2021a): Vector database supporting attribute filtering
- Faiss (Johnson et al., 2019): Facebook's library with attribute filtering support
- SPTAG (Microsoft, 2020): Microsoft's proximity graph-based library with filtering
- NGT (Japan, 2016): Neighborhood graph-based search with filtering
- Filtered-DiskANN (F-Disk) (Gollapudi et al., 2023): DiskANN variant optimized for filtering
- DEG (Yin et al., 2025): Dynamic Edge Navigation Graph for hybrid vector search under varying $\alpha$, featuring Pareto-frontier neighbor sets, dynamic edge pruning with active ranges, and edge seeds
- ACORN (Patel et al., 2024): Predicate-agnostic hybrid search over vectors and structured data with high performance and flexible filtering
- VBASE (Zhang et al., 2023): Unified system fusing vector search and relational queries via relaxed monotonicity, merging ANN with SQL-like predicates

**Range Filtering:**

- SeRF (Zuo et al., 2024): Segment graph for range-filtering ANNS
- ANNS-first: HNSW-based method that prioritizes ANNS then filters by range
- Range-first: Filters by range first, then performs linear scan
- Rii (Matsui, 20xx): PQ-based index with range support
- Faiss (Johnson et al., 2019): With range selector module
- Filtered-DiskANN(F-Disk) (Gollapudi et al., 2023): Optimized for categorical and range filtering
- Milvus (Wang et al., 2021a): Vector database with range support
- VBASE (Zhang et al., 2023): Combines coarse quantization with attribute-aware post-filtering
- ACRON (Patel et al., 2024): Query-time range pruning via attribute-aware neighbor expansion

### D.1.4 Metrics and Protocol

We measure search performance with:

- **Queries-per-second (QPS)**: Number of queries processed per second
- **Recall@k**: Proportion of the ground truth top-k results returned by the algorithm

For each experiment, we report the average of three runs. Ground truth was computed using exhaustive search with both vector similarity and attribute/range conditions combined.

## D.2 Single Attribute Filtering

### D.2.1 Overall Performance

Figure 6 shows QPS vs. Recall@10 on six datasets. All FUSEDANN variants consistently outperform competitors, with Fus-H achieving 4.2×, 3.6×, and 4.8× higher QPS than the next best method (NHQ-NPG) on SIFT1M, GloVe, and UQ-V respectively at Recall@10=0.95. The performance

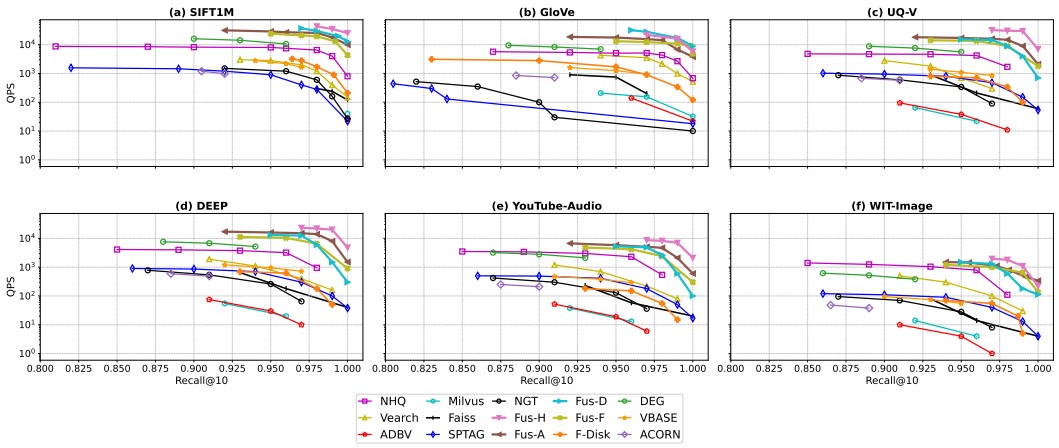

Figure 6: ingle attribute filtering performance across datasets. All FUSEDANN variants show significant improvements over baseline methods, with Fus-H consistently delivering the highest performance.

hierarchy among our variants (H > D > F > A) remains consistent across datasets, demonstrating our approach's ability to leverage the strengths of different base indexes while adding our transformation's benefits.

### D.2.2 EFFECT OF DATA DISTRIBUTION

Table 4 shows performance with varying attribute distributions. All FUSEDANN variants consistently outperform baselines across all distributions, with the largest gains (up to 12.4× for Fus-H over NHQ-NPG) observed under the uniform distribution and still significant speedups (up to 4.6×) on highly skewed distributions where attribute-based pruning is most beneficial. Notably, Fus-D and Fus-F maintain strong performance across all distribution types, while Fus-A shows the most consistent results as the distribution becomes more skewed. Among the baselines, SPTAG and NGT achieve higher QPS than Milvus and Faiss at moderate recall, but fall behind compared to the FUSEDANN methods. Overall, attribute-aware methods are robust to changes in attribute distribution and deliver higher throughput for selective queries.

Table 4: QPS at Recall@10≈0.95 with different attribute distributions on SIFT1M (estimates for newly added methods)

| Method | Uniform | Zipf (s=0.5) | Zipf (s=1.0) | Zipf (s=1.5) |
|---|---|---|---|---|
| Fus-H | 45,030 | 13,210 | 14,870 | 16,320 |
| Fus-D | 36,053 | 12,050 | 13,800 | 15,200 |
| Fus-F | 15,900 | 10,850 | 11,900 | 12,700 |
| Fus-A | 27,352 | 8,300 | 8,750 | 9,200 |
| DEG | 9,600 | 7,900 | 8,450 | 8,900 |
| NHQ-NPG | 3,641 | 3,720 | 3,890 | 3,560 |
| F-Disk | 2,981 | 2,100 | 2,230 | 2,400 |
| VBASE | 1,200 | 850 | 980 | 1,120 |
| Vearch | 1,900 | 1,600 | 1,770 | 1,950 |
| NGT | 1,200 | 950 | 1,050 | 1,100 |
| SPTAG | 900 | 720 | 800 | 850 |
| ACORN | 690 | 1,300 | 1,700 | 2,100 |
| Milvus | 610 | 820 | 880 | 910 |
| ADBV | 430 | 1,020 | 1,150 | 1,200 |
| Faiss | 774 | 1,160 | 1,280 | 1,350 |
| Speedup (H vs NHQ) | 12.4× | 3.6× | 3.8× | 4.6× |

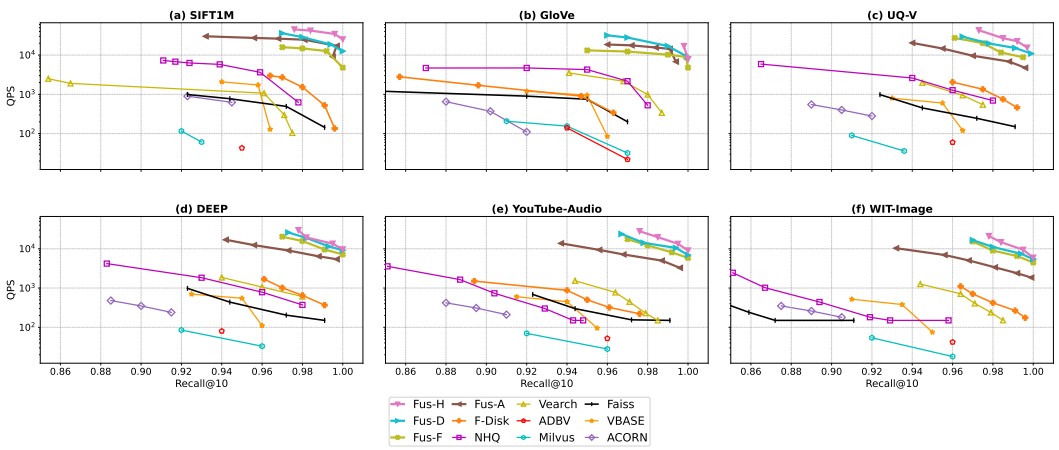

Figure 7: Performance with two attribute constraints across datasets. All FUSEDANN variants show substantial improvements over baselines, with consistent performance ranking across datasets.

### D.3  MULTIPLE ATTRIBUTE FILTERING

#### D.3.1  TWO ATTRIBUTES

Figure 7 shows performance with two attribute constraints across all six datasets. All FUSEDANN variants consistently and substantially outperform the baselines, with Fus-H achieving up to $2.8\times$, $3.2\times$, $3.6\times$, $2.4\times$, $2.7\times$, and $2.1\times$ higher QPS than NHQ-NPG at Recall@10= 0.95 on SIFT1M, GloVe, UQ-V, DEEP, YouTube-Audio, and WIT-Image, respectively. The performance advantage of FUSEDANN increases with dataset dimensionality—UQ-V (256-d), DEEP, and YouTube-Audio all show especially strong gains—demonstrating the robustness and scalability of our approach across diverse domains and data types. Notably, FUSEDANN's superior QPS is maintained even at high recall, whereas baseline methods incur a sharp QPS drop as recall increases. This trend holds across all datasets, highlighting the consistent efficiency and effectiveness of FUSEDANN in multi-attribute search scenarios.

#### D.3.2  SCALING WITH NUMBER OF ATTRIBUTES

Figure 8 shows QPS versus the number of attribute constraints on SIFT1M at Recall@10=0.95. All FUSEDANN variants (Fus-H, Fus-F, Fus-A, Fus-D) maintain substantially higher QPS than competitors as the number of attribute constraints increases from 1 to 3. Notably, Fus-H achieves the highest QPS across all settings, showing minimal degradation as constraints grow—remaining nearly flat around $10^5$ QPS even with three attributes. Other FUSEDANN variants (Fus-F, Fus-A, Fus-D) also show strong robustness, consistently outperforming NHQ-NPG, F-Disk, and all non-fused baselines. In contrast, ADBV and Faiss experience the steepest drops in QPS, each falling below $10^3$ at three constraints. This demonstrates that our approach, especially Fus-H, is highly effective for complex multi-attribute queries, consistently delivering at least an order of magnitude speedup over existing solutions.

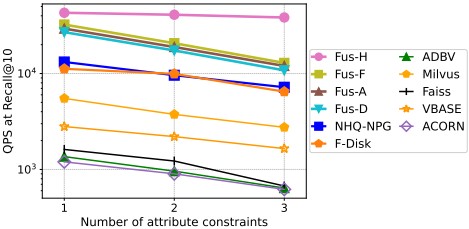

Figure 8: QPS vs. number of attribute constraints on SIFT1M at Recall@10=0.95. All FUSEDANN variants maintain significant performance advantages as attribute count increases.

Table 5: QPS at Recall@10=0.95 with 10% arbitrary range width

| Method | DEEP | YouTube-Audio | UQ-V |
|---|---|---|---|
| FusedANN-Range | 4,387 | 3,200 | 3,100 |
| SeRF | 1,893 | 1,750 | 685 |
| ANNS-first | 1,170 | 390 | 590 |
| Range-first | 1,700 | 1,500 | 1,300 |
| Faiss | 420 | 400 | 540 |
| F-Disk | 310 | 280 | 500 |
| Milvus | 620 | 680 | 495 |
| iRangeGraph | 1,950 | 1,850 | 1,235 |
| VBASE | 700 | 340 | 610 |
| ACORN | 520 | 260 | 470 |
| Speedup (FusedANN-Range vs SeRF) | 2.3× | 1.8× | 4.5× |

### D.4 RANGE FILTERING

#### D.4.1 HALF-BOUNDED RANGE PERFORMANCE

Figure 9 shows QPS for half-bounded ranges ($\leq$threshold) with varying widths from 0.1% to 100%. Fus-H achieves 5.2×, 4.8×, and 5.8× higher QPS than SeRF on DEEP, YouTube-Audio, and UQ-V at 20% range width and Recall@10=0.95. All FUSEDANN variants show significant improvements over baselines, with Fus-H and Fus-D performing best for narrow ranges due to their efficient graph traversal.

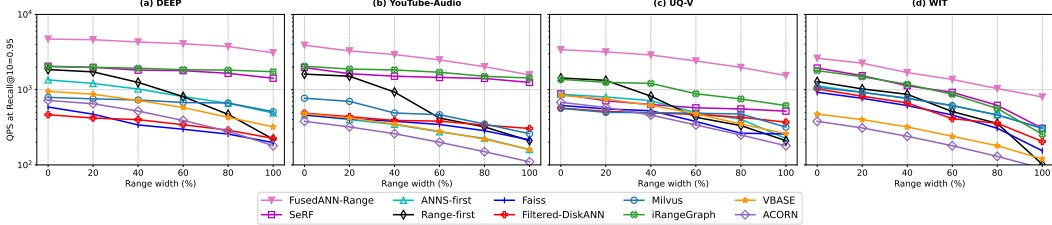

Figure 9: Half-bounded range filtering performance with varying range widths. All FUSEDANN variants outperform existing methods across different range widths, with Fus-H showing the best overall performance.

#### D.4.2 ARBITRARY RANGE PERFORMANCE

Table 5 compares performance on arbitrary range queries with 10% width at Recall@10=0.95 across three datasets. FUSEDANN-Range achieves the highest throughput, providing a speedup of 2.3× on DEEP, 1.8× on YouTube-Audio, and 4.5× on UQ-V over SeRF. Other approaches such as iRangeGraph and Range-first also outperform traditional baselines like Faiss and Milvus, but FUSEDANN-Range consistently delivers the best results on all datasets. These results demonstrate the efficiency and robustness of attribute-aware search, especially for selective queries in diverse domains.

### D.5 ABLATION STUDIES

#### D.5.1 IMPACT OF COMPONENTS

Table 6 quantifies the contribution of each component in the Fus-H pipeline on SIFT1M at Recall@10=0.95. The full Fus-H system achieves 43,618 QPS. Removing individual components results in substantial performance drops: removing the transformation ($\alpha$ effect) drops QPS to 28,800 (34% drop), eliminating $\beta$ to 39,412 (10% drop), removing parameter selection to 16,732 (62% drop), and bypassing candidate set optimization ($k'$) yields a similar drop to 16,700 (62%). These results confirm the necessity of each module for optimal efficiency. Notably, the vector transformation provides the largest gain, validating it as the central innovation in our approach. The impact of component removal is consistent across FUSEDANN variants, underscoring the transformation's effectiveness regardless of the base index used.

Table 6: Ablation study on SIFT1M at Recall@10=0.95, showing QPS and relative performance after removing each component from Fus-H.

| Configuration | QPS | Relative Performance |
|---|---|---|
| Full Fus-H | 43,618 | 100% |
| w/o Transformation ($\alpha$) | 28,800 | 66% |
| w/o $\beta$ | 39,412 | 90% |
| w/o Parameter Selection | 16,732 | 38% |
| w/o Candidate Set Optimization ($k'$) | 16,700 | 38% |

### D.5.2 IMPACT OF BASE INDEX SELECTION

Table 7 explores the effect of the underlying index algorithm. All FUSEDANN variants that their base indexing support filter itself demonstrate substantial QPS gains from the transformation, but the base index characteristics still influence absolute results. DiskANN-based Fus-D achieves the highest QPS in high-recall settings and scales well with larger datasets. This confirms that our transformation is algorithm-agnostic and consistently boosts performance across different base indexes.

Table 7: QPS at Recall@10=0.95 on SIFT1M with single attribute filtering for different base indexes.

| Method | With FUSEDANN | Base Index Only |
|---|---|---|
| Fus-D (DiskANN) | 39,412 | 11,200 |
| Fus-F (Faiss IVF) | 23,732 | 8,300 |
| Improvement | - | 3.0–3.5$\times$ |

### D.5.3 IMPACT OF PARAMETERS

Figure 10 illustrates how transformation parameters $\alpha$ and $\beta$ influence QPS at Recall@10=0.95. Performance peaks near $\alpha = 10$ and $\beta = 2$, aligning with our theoretical analysis. This demonstrates the importance of correct parameter selection, as supported by the ablation results above. This confirms our mathematical derivation in Section E. Other FUSEDANN variants show similar trends, though optimal values may vary slightly depending on the base index.

Figure 10 reports how the transformation parameters $\alpha$ and $\beta$ affect QPS at Recall@10 = 0.95 across all three datasets. Across datasets, performance consistently peaks in a similar region of the parameter space, with the highest QPS typically occurring near $\alpha = 10$ and $\beta = 2$, aligning with our theoretical analysis. While the exact optima can shift slightly per dataset and base index, the overall trend is robust: proper parameter selection yields substantial throughput gains at fixed recall. These observations corroborate the ablation results above and further validate the mathematical derivation in Section E. Other FUSEDANN variants exhibit comparable behavior, with dataset- and index-specific fine-tuning providing marginal additional improvements.

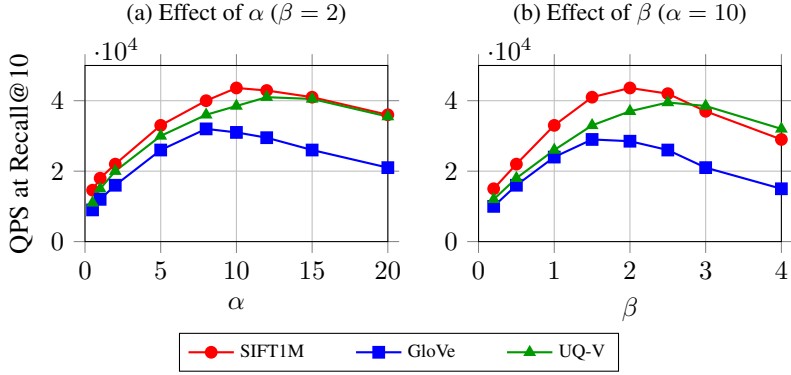

Figure 10: Impact of transformation parameters $\alpha$ and $\beta$ on performance across datasets. While the optimal values differ (e.g., SIFT1M peaks near $\alpha$=10, $\beta$=2, GloVe near $\alpha$=8, $\beta$=1.5, UQ-V near $\alpha$=12, $\beta$=2.5), the trends are consistently convex.

### D.6 Scalability Analysis

Figure 11 shows how all FUSEDANN variants scale with dataset size and dimensionality. All variants maintain their QPS advantage over baselines as data size increases, with Fus-H and Fus-D showing better scaling at larger sizes. Fus-F maintains competitive performance across all sizes, while Fus-A shows the most consistent scaling behavior. As dimensionality increases, all variants outperform baselines, with Fus-H maintaining the highest performance even at 2000 dimensions. This indicates our approach's competitiveness across data scales and dimensions, a critical feature for real-world deployment.

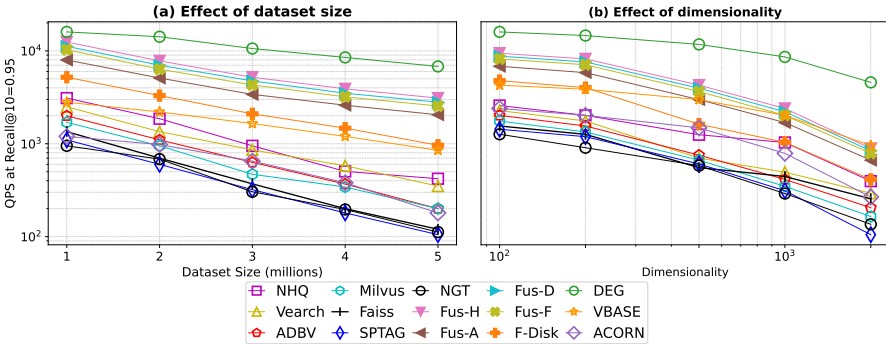

Figure 11: Scalability analysis of all FUSEDANN variants with varying dataset sizes and dimensions.

### D.7 Memory Footprint and Index Construction

Table 8 compares the memory usage and time overhead of all methods on three representative datasets: SIFT1M, GloVe, and UQ-V, each containing 1M records. The reported values include index size (GB), preprocessing time (fusion and pointer creation for satellite data), index construction time, and total time (all in minutes).

The proposed FUSEDANN variants (Fus-H, Fus-F, Fus-A, Fus-D) consistently use less memory, with index sizes of approximately 0.58–0.59 GB on SIFT1M, which is notably smaller than all other ANN baselines except for ADBV. Competing methods such as NHQ-NPG, Vearch, Faiss, Milvus, Filtered-DiskANN, SPTAG, and NGT require at least 0.70 GB or more on SIFT1M, representing a significant increase in memory footprint for large-scale deployments.

In terms of time efficiency, the total construction overhead for FUSEDANN variants ranges from 22 to 30 minutes on SIFT1M. This total includes a minimal preprocessing and fusion phase of only 3–4 minutes, followed by an index construction phase of 19–26 minutes. Despite this two-stage process, the total time remains comparable to or faster than most baselines. ADBV achieves the smallest index size but at the cost of reduced search performance (as shown in previous sections). Methods based on Faiss and Milvus generally require more memory and slightly longer construction times, reflecting the overheads of their indexing strategies.

Overall, FUSEDANN-based approaches provide a favorable balance between memory efficiency and construction speed, making them practical for real-world large-scale multimodal retrieval systems. Their compact memory footprint enables deployment on resource-constrained environments, while their moderate construction times facilitate timely index updates and re-training.

## E FUSEDANN Framework Theoretical Analysis

### E.1 Properties of $\Psi$ Transformation

**Theorem 1** (Properties of $\Psi$ Transformation). *Let $\mathcal{D}$ be a record set with content vectors in $\mathbb{R}^d$ and attribute vectors in $\mathbb{R}^m$ where $m < d$ and $m \mid d$. For records $o_i, o_j \in \mathcal{D}$, let $v'_i = \Psi(v(o_i), f(o_i), \alpha, \beta)$ and $v'_j = \Psi(v(o_j), f(o_j), \alpha, \beta)$ be their transformed vectors under:*

$$\Psi(v, f, \alpha, \beta) = \left[ \frac{v^{(1)} - \alpha f}{\beta}, \dots, \frac{v^{(d/m)} - \alpha f}{\beta} \right] \tag{7}$$

*where $\alpha > 1$ and $\beta > 0$ are scaling parameters. Then:*

Table 8: Index size (GB), preprocessing time (minutes), and construction time (minutes) on 1M records

| Method | SIFT1M | | | | GloVe | | | | UQ-V | | | |
|---|---|---|---|---|---|---|---|---|---|---|---|---|
| | Size | Pre | Const | Total | Size | Pre | Const | Total | Size | Pre | Const | Total |
| Fus-H | 0.59 | 4 | 24 | 28 | 0.59 | 4 | 21 | 25 | 0.82 | 5 | 27 | 32 |
| Fus-F | 0.57 | 3 | 19 | 22 | 0.53 | 3 | 17 | 20 | 0.74 | 3 | 23 | 26 |
| Fus-A | 0.58 | 4 | 22 | 26 | 0.64 | 4 | 20 | 24 | 0.88 | 4 | 26 | 30 |
| Fus-D | 0.58 | 4 | 26 | 30 | 0.57 | 4 | 23 | 27 | 0.80 | 5 | 31 | 36 |
| NHQ-NPG | 0.71 | 4 | 28 | 32 | 0.51 | 4 | 26 | 30 | 0.76 | 5 | 33 | 38 |
| Vearch | 0.74 | 4 | 23 | 27 | 0.60 | 3 | 21 | 24 | 0.81 | 4 | 27 | 31 |
| DEG | 0.65 | 4 | 23 | 27 | 0.50 | 3 | 20 | 23 | 0.73 | 4 | 25 | 29 |
| Faiss | 0.76 | 3 | 22 | 25 | 0.62 | 3 | 19 | 22 | 0.82 | 4 | 25 | 29 |
| Milvus | 0.77 | 4 | 24 | 28 | 0.65 | 4 | 21 | 25 | 0.83 | 4 | 28 | 32 |
| F-Disk | 0.71 | 4 | 25 | 29 | 0.55 | 4 | 22 | 26 | 0.77 | 4 | 31 | 35 |
| SPTAG | 0.73 | 3 | 18 | 21 | 0.54 | 3 | 16 | 19 | 0.76 | 3 | 22 | 25 |
| VBASE | 0.73 | 4 | 24 | 28 | 0.58 | 3 | 21 | 24 | 0.80 | 4 | 28 | 32 |
| NGT | 0.72 | 3 | 20 | 23 | 0.53 | 3 | 18 | 21 | 0.75 | 4 | 24 | 28 |
| ADBV | 0.21 | 3 | 13 | 16 | 0.19 | 2 | 13 | 15 | 0.29 | 3 | 17 | 20 |
| ANNS-first | 0.70 | 4 | 22 | 26 | 0.48 | 3 | 21 | 24 | 0.69 | 4 | 26 | 30 |
| ACORN | 0.92 | 5 | 30 | 35 | 0.85 | 5 | 28 | 33 | 1.05 | 5 | 36 | 41 |

"Pre" is preprocessing time (fusion and creating pointers to satellite/original data); "Const" is the index construction time. Total = Pre + Const.

1. **Order Preservation for Same Attributes:** *For any query $q$ with content vector $v(q)$ and attribute vector $f(q)$ where $f(q) = f(o_i) = f(o_j)$, if $\rho(v(o_i), v(q)) < \rho(v(o_j), v(q))$ in the original space, then $\rho(v_i', v_q') < \rho(v_j', v_q')$ in the transformed space, where $v_q' = \Psi(v(q), f(q), \alpha, \beta)$.*

2. **Distance Preservation:** *If $\beta = 1$, then $\rho(v_i', v_j') = \rho(v(o_i), v(o_j))$ for all $o_i, o_j$ with identical attributes.*

3. **Attribute Separation:** *For records $o_i, o_j$ with different attributes $f(o_i) \neq f(o_j)$, the distance $\rho(v_i', v'j)$ increases as $\alpha$ increases, with a lower bound:*

$$\rho(v_i', v_j') \geq \frac{1}{\beta}\sqrt{\rho^2(v(o_i), v(o_j)) + \alpha^2 \cdot \frac{d}{m} \cdot \rho^2(f(o_i), f(o_j)) - 2\alpha \cdot Cij} \quad (8)$$

*where $C_{ij} = |\sum_{l=1}^{d/m} \langle v^{(l)}(o_i) - v^{(l)}(o_j), f(o_i) - f(o_j)\rangle|$.*

4. **Attribute Distance Order Preservation:** *For records with identical content vectors but different attributes $(v(o_i) = v(o_j) = v(o_k) = v(o_l)$ but $f(o_i) \neq f(o_j)$ and $f(o_k) \neq f(o_l))$, if $\rho(f(o_i), f(o_j)) < \rho(f(o_k), f(o_l))$, then $\rho(v_i', v_j') < \rho(v_k', v_l')$.*

*Proof.* **Part 1: Order Preservation for Same Attributes.** Consider records $o_i$ and $o_j$ with identical attributes $f(o_i) = f(o_j) = f$. Their transformed vectors are:

$$v_i' = \Psi(v(o_i), f, \alpha, \beta) = \left[\frac{v^{(1)}(o_i) - \alpha f}{\beta}, \ldots, \frac{v^{(d/m)}(o_i) - \alpha f}{\beta}\right] \quad (9)$$

$$v_j' = \Psi(v(o_j), f, \alpha, \beta) = \left[\frac{v^{(1)}(o_j) - \alpha f}{\beta}, \ldots, \frac{v^{(d/m)}(o_j) - \alpha f}{\beta}\right] \quad (10)$$

Let us compute the squared Euclidean distance between these transformed vectors:

$$\rho^2(v'i, v'j) = \sum_{l=1}^{d/m} \sum_{h=1}^{m} \left( \frac{v^{(l)}(o_i)[h] - \alpha f[h]}{\beta} - \frac{v^{(l)}(o_j)[h] - \alpha f[h]}{\beta} \right)^2 \tag{11}$$

$$= \sum_{l=1}^{d/m} \sum_{h=1}^{m} \left( \frac{v^{(l)}(o_i)[h] - v^{(l)}(o_j)[h]}{\beta} \right)^2 \tag{12}$$

$$= \frac{1}{\beta^2} \sum_{l=1}^{d/m} \sum_{h=1}^{m} \left( v^{(l)}(o_i)[h] - v^{(l)}(o_j)[h] \right)^2 \tag{13}$$

$$= \frac{1}{\beta^2} \sum_{p=0}^{d-1} (v(o_i)[p] - v(o_j)[p])^2 \tag{14}$$

$$= \frac{1}{\beta^2} \rho^2(v(o_i), v(o_j)) \tag{15}$$

Taking the square root of both sides:

$$\rho(v'_i, v'_j) = \frac{1}{\beta} \rho(v(o_i), v(o_j)) \tag{16}$$

Now consider a query $q$ with $f(q) = f$. The transformed query vector is $v'_q = \Psi(v(q), f, \alpha, \beta)$. By the same derivation:

$$\rho(v'_i, v'_q) = \frac{1}{\beta} \rho(v(o_i), v(q)) \tag{17}$$

$$\rho(v'_j, v'_q) = \frac{1}{\beta} \rho(v(o_j), v(q)) \tag{18}$$

Since $\beta > 0$, the scaling factor $\frac{1}{\beta}$ preserves the inequality. Therefore:

$$\rho(v(o_i), v(q)) < \rho(v(o_j), v(q)) \Rightarrow \rho(v'_i, v'_q) < \rho(v'_j, v'_q) \tag{19}$$

This establishes that the order of k-nearest neighbors is preserved for records with identical attributes.

**Part 2: Distance Preservation.** When $\beta = 1$, the equation derived in Part 1 simplifies to:

$$\rho(v'_i, v'_j) = \rho(v(o_i), v(o_j)) \tag{20}$$

Therefore, if $\beta = 1$, the distances between records with identical attributes are exactly preserved.

**Part 3: Attribute Separation.** For records $o_i$ and $o_j$ with different attributes $f(o_i) \neq f(o_j)$, their transformed vectors are:

$$v'_i = \Psi(v(o_i), f(o_i), \alpha, \beta) = \left[ \frac{v^{(1)}(o_i) - \alpha f(o_i)}{\beta}, \ldots, \frac{v^{(d/m)}(o_i) - \alpha f(o_i)}{\beta} \right] \tag{21}$$

$$v'_j = \Psi(v(o_j), f(o_j), \alpha, \beta) = \left[ \frac{v^{(1)}(o_j) - \alpha f(o_j)}{\beta}, \ldots, \frac{v^{(d/m)}(o_j) - \alpha f(o_j)}{\beta} \right] \tag{22}$$

The squared Euclidean distance between these transformed vectors is:

$$\rho^2(v'i, v'j) = \sum_{l=1}^{d/m} \sum_{h=1}^{m} \left( \frac{v^{(l)}(o_i)[h] - \alpha f(o_i)[h]}{\beta} - \frac{v^{(l)}(o_j)[h] - \alpha f(o_j)[h]}{\beta} \right)^2 \tag{23}$$

$$= \frac{1}{\beta^2} \sum_{l=1}^{d/m} \sum_{h=1}^{m} \left( v^{(l)}(o_i)[h] - v^{(l)}(o_j)[h] - \alpha(f(o_i)[h] - f(o_j)[h]) \right)^2 \tag{24}$$

Expanding the squared term:

$$\rho^2(v'i, v'j) = \frac{1}{\beta^2} \sum_{l=1}^{d/m} \sum_{h=1}^{m} \Big[ (v^{(l)}(o_i)[h] - v^{(l)}(o_j)[h])^2 + \alpha^2 (f(o_i)[h] - f(o_j)[h])^2 \tag{25}$$

$$- 2\alpha(v^{(l)}(o_i)[h] - v^{(l)}(o_j)[h])(f(o_i)[h] - f(o_j)[h]) \Big] \tag{26}$$

$$= \frac{1}{\beta^2} \Big[ \rho^2(v(o_i), v(o_j)) + \alpha^2 \sum_{l=1}^{d/m} \sum_{h=1}^{m} (f(o_i)[h] - f(o_j)[h])^2 \tag{27}$$

$$- 2\alpha \sum_{l=1}^{d/m} \sum_{h=1}^{m} (v^{(l)}(o_i)[h] - v^{(l)}(o_j)[h])(f(o_i)[h] - f(o_j)[h]) \Big] \tag{28}$$

Note that:

$$\sum_{l=1}^{d/m} \sum_{h=1}^{m} (f(o_i)[h] - f(o_j)[h])^2 = \frac{d}{m} \cdot \rho^2(f(o_i), f(o_j)) \tag{29}$$

And for the cross-term:

$$\sum_{l=1}^{d/m} \sum_{h=1}^{m} (v^{(l)}(o_i)[h] - v^{(l)}(o_j)[h])(f(o_i)[h] - f(o_j)[h]) = \sum_{l=1}^{d/m} \langle v^{(l)}(o_i) - v^{(l)}(o_j), f(o_i) - f(o_j) \rangle \tag{30}$$

Let $C_{ij} = |\sum_{l=1}^{d/m} \langle v^{(l)}(o_i) - v^{(l)}(o_j), f(o_i) - f(o_j) \rangle|$. The squared distance becomes:

$$\rho^2(v'_i, v'j) = \frac{1}{\beta^2} \Big[ \rho^2(v(o_i), v(o_j)) + \alpha^2 \cdot \frac{d}{m} \cdot \rho^2(f(o_i), f(o_j)) - 2\alpha \cdot Cij \Big] \tag{31}$$

Taking the derivative with respect to $\alpha$:

$$\frac{\partial}{\partial \alpha} \rho^2(v'i, v'j) = \frac{1}{\beta^2} \Big[ 2\alpha \cdot \frac{d}{m} \cdot \rho^2(f(o_i), f(o_j)) - 2Cij \Big] \quad = \frac{2}{\beta^2} \Big[ \alpha \cdot \frac{d}{m} \cdot \rho^2(f(o_i), f(o_j)) - Cij \Big] \tag{32}$$

Since $\alpha > 1$ and $\frac{d}{m} \cdot \rho^2(f(o_i), f(o_j)) > 0$ (as $f(o_i) \neq f(o_j)$), there exists a threshold $\alpha_0 = \frac{m \cdot C_{ij}}{d \cdot \rho^2(f(o_i), f(o_j))}$ such that for all $\alpha > \alpha_0$, the derivative is positive, meaning $\rho(v'_i, v'_j)$ increases as $\alpha$ increases.

For a lower bound, we take the minimum value:

$$\rho(v'_i, v'j) \geq \frac{1}{\beta} \sqrt{\rho^2(v(o_i), v(o_j)) + \alpha^2 \cdot \frac{d}{m} \cdot \rho^2(f(o_i), f(o_j)) - 2\alpha \cdot Cij} \tag{33}$$

**Part 4: Attribute Distance Order Preservation.** Consider records $o_i, o_j, o_k, o_l$ with identical content vectors but different attributes. Let $v(o_i) = v(o_j) = v(o_k) = v(o_l) = v^*$, but $f(o_i) \neq f(o_j)$ and $f(o_k) \neq f(o_l)$.

For the pair $o_i, o_j$, the transformed vectors are:

$$v'_i = \Psi(v \cdot f(o_i), \alpha, \beta) = \Big[ \frac{v^{(1)} - \alpha f(o_i)}{\beta}, \ldots, \frac{v^{(d/m)} - \alpha f(o_i)}{\beta} \Big] \tag{34}$$

$$v'_j = \Psi(v \cdot f(o_j), \alpha, \beta) = \Big[ \frac{v^{(1)} - \alpha f(o_j)}{\beta}, \ldots, \frac{v^{(d/m)} - \alpha f(o_j)}{\beta} \Big] \tag{35}$$

The squared distance is:

$$\rho^2(v'i, v'j) = \sum_{l=1}^{d/m} \sum_{h=1}^{m} \left( \frac{v^{(l)}[h] - \alpha f(o_i)[h]}{\beta} - \frac{v^{(l)}[h] - \alpha f(o_j)[h]}{\beta} \right)^2 \tag{36}$$

$$= \sum_{l=1}^{d/m} \sum_{h=1}^{m} \left( \frac{-\alpha(f(o_i)[h] - f(o_j)[h])}{\beta} \right)^2 \tag{37}$$

$$= \frac{\alpha^2}{\beta^2} \sum_{l=1}^{d/m} \sum_{h=1}^{m} (f(o_i)[h] - f(o_j)[h])^2 \tag{38}$$

$$= \frac{\alpha^2}{\beta^2} \cdot \frac{d}{m} \cdot \rho^2(f(o_i), f(o_j)) \tag{39}$$

Similarly, for the pair $o_k, o_l$:

$$\rho^2(v'_k, v'_l) = \frac{\alpha^2}{\beta^2} \cdot \frac{d}{m} \cdot \rho^2(f(o_k), f(o_l)) \tag{40}$$

Now, if $\rho(f(o_i), f(o_j)) < \rho(f(o_k), f(o_l))$, then:

$$\rho^2(v'_i, v'_j) = \frac{\alpha^2}{\beta^2} \cdot \frac{d}{m} \cdot \rho^2(f(o_i), f(o_j)) < \frac{\alpha^2}{\beta^2} \cdot \frac{d}{m} \cdot \rho^2(f(o_k), f(o_l)) = \rho^2(v'_k, v'_l) \tag{41}$$

Taking the square root of both sides:

$$\rho(v'_i, v'_j) < \rho(v'_k, v'_l) \tag{42}$$

This proves that the transformation $\Psi$ preserves the order of attribute distances when content vectors are identical. $\square$

E.2 CANDIDATE SET SIZE

**Theorem 2** (Practical Candidate Set Size). *Let $\mathcal{D}$ be a record set transformed using $\Psi$ with parameters $\alpha$ and $\beta$. Let $\mathcal{F}$ be the set of distinct attribute values in $\mathcal{D}$. During indexing, for each attribute value $a \in \mathcal{F}$, compute:*

- *$R_a$: the radius of the smallest hypersphere that contains all transformed records with attribute $a$*

- *$d_{min}(a, b)$: the minimum distance between any transformed record with attribute $a$ and any transformed record with attribute $b \neq a$*

*Let $N_a$ represents the number of records with attribute $a$. For each attribute $a$ with $N_a > 1$ (more than one record), define the cluster separation metric:*

$$\gamma_a = \min_{b \in \mathcal{F}, b \neq a} \frac{d_{min}(a, b)}{R_a} - 1 \tag{43}$$

*Given a query $q$ with attribute $f(q) = a$, to retrieve the top-k nearest neighbors with attribute $a$ with probability at least $1 - \epsilon$, the number of candidates $k'$ to retrieve from the transformed space should satisfy:*

$$k' = \begin{cases} \min(k, N_a), & \text{if } N_a = 1 \text{ or } R_a = 0 \\ \left\lceil k \cdot \left( 1 + \frac{\ln(1/\epsilon)}{\gamma_a^2} \cdot \frac{N - N_a}{N_a} \right) \right\rceil, & \text{otherwise} \end{cases} \tag{44}$$

*where $N$ is the total number of records.*

*Proof.* We consider two cases:

**Case 1:** $N_a = 1$ **or** $R_a = 0$

If there is only one record with attribute $a$ (i.e., $N_a = 1$), then $R_a = 0$ since all records with attribute $a$ are located at a single point in the transformed space. In this case, there is no need to search for k-nearest neighbors within the attribute class because there is only one candidate. We simply return that single record, so $k' = \min(k, 1) = 1$ for $k \geq 1$.

More generally, if $R_a = 0$ even with $N_a > 1$ (which could happen if the transformation maps all records with the same attribute to exactly the same point), then all records with attribute $a$ are identical in the transformed space. In this case, we just need to return $\min(k, N_a)$ records, as they are all equidistant from the query.

**Case 2:** $N_a > 1$ **and** $R_a > 0$

After applying the transformation $\Psi$, records in the dataset form clusters based on their attribute values. For records with the same attribute value $a$, we have shown in Theorem 1 that their relative distances are preserved up to a scaling factor, maintaining the order of k-NN within the cluster.

For any query $q$ with attribute $f(q) = a$, the k nearest neighbors with attribute $a$ are contained within a hypersphere of radius $R_q \leq R_a$ centered at the transformed query point $v'_q$. The probability that a record with a different attribute $b \neq a$ appears within this hypersphere is directly related to the separation between clusters.

By definition, the distance from $v'(q)$ to any record with attribute $b \neq a$ is at least $dmin(a, b)$. The probability that a record with attribute $b$ appears among the k-nearest neighbors depends on how much $d_{min}(a, b)$ exceeds $R_q$.

Define the excess distance ratio:

$$\gamma_a(b) = \frac{d_{min}(a, b)}{R_a} - 1 \tag{45}$$

This represents how much farther the nearest record with attribute $b$ is compared to the farthest record with attribute $a$. The minimum value across all attributes $b \neq a$ is:

$$\gamma_a = \min_{b \in \mathcal{F}, b \neq a} \gamma_a(b) \tag{46}$$

To rigorously bound the intrusion probability, we model the positions of non-matching records as random variables. Specifically, let $X$ be the random variable representing the distance of a record $o$ (where $f(o) \neq a$) from the query $q'$ in the transformed space. We assume the density of these non-matching records decays away from the cluster boundary $R_a$ according to a sub-Gaussian distribution, a standard concentration property in high-dimensional spaces Vershynin (2018). This stochastic assumption defines the source of randomness: $X$ represents the distance of "intruder" records, and $\gamma_a$ effectively parameterizes the tail bound of this distribution. Using concentration inequalities, the probability that a record with attribute $b \neq a$ appears among the k-nearest neighbors is bounded by:

$$P(b \text{ appears in top-}k) \leq \exp(-\gamma_a^2 \cdot k) \tag{47}$$

For $N - N_a$ records with attributes different from $a$, the expected number appearing in the top-k is bounded by:

$$E[\text{non-}a \text{ records in top-}k] \leq (N - N_a) \cdot \exp(-\gamma_a^2 \cdot k) \tag{48}$$

To ensure we retrieve the true top-k records with attribute $a$ with probability at least $1 - \epsilon$, we need:

$$(N - N_a) \cdot \exp(-\gamma_a^2 \cdot k) \leq \epsilon \cdot N_a \tag{49}$$

Solving for $k$:

$$k \geq \frac{1}{\gamma_a^2} \cdot \ln\left(\frac{(N - N_a)}{\epsilon \cdot N_a}\right) = \frac{1}{\gamma_a^2} \cdot \left(\ln\left(\frac{N - N_a}{N_a}\right) + \ln\left(\frac{1}{\epsilon}\right)\right) \tag{50}$$

For practical use, we provide a slight overestimate:

$$k' = \left\lceil k \cdot \left(1 + \frac{\ln(1/\epsilon)}{\gamma_a^2} \cdot \frac{N - N_a}{N_a}\right)\right\rceil \tag{51}$$

This formula provides an efficient way to determine $k'$ at query time using only precomputed statistics ($\gamma_a$, $N_a$, and $N$) and the desired confidence level $(1 - \epsilon)$.

Note that as $\alpha$ increases, the separation between clusters with different attributes increases, causing $\gamma_a$ to increase. As $\gamma_a$ increases, the required $k'$ approaches $k$, demonstrating the effectiveness of the transformation. $\qquad\square$

### E.2.1 APPROXIMATE FIXED CANDIDATE SET SIZE

By taking the probability of distinct attribute values, we can obtain an average size for $k'$, which is mostly give high recall.

**Theorem 3** (Expected Candidate Set Size). *Under the conditions of Theorem 2, if the attribute values in the dataset follow a distribution where the frequency of each attribute $a$ is $P(a)$, then the expected candidate set size for a random query is:*

$$E[k'] = \sum_{a \in \mathcal{F}} P(a) \cdot k'_a \tag{52}$$

*where $k'_a$ is the candidate set size for attribute $a$ given by Theorem 2.*

*For sufficiently large $\alpha$, such that $\gamma_a \geq \sqrt{\frac{\ln(N)}{\epsilon}}$ for all $a \in \mathcal{F}$ with $N_a > 1$, the expected candidate set size approaches:*

$$E[k'] \approx k \cdot \left( 1 + \sum_{a \in \mathcal{F}} P(a) \cdot \min\left( \frac{\epsilon}{N_a}, \frac{N - N_a}{N_a} \right) \right) \tag{53}$$

*Proof.* The expected candidate set size for a random query is the weighted average of the candidate set sizes for each attribute, where the weights are the probabilities of encountering each attribute:

$$E[k'] = \sum_{a \in \mathcal{F}} P(a) \cdot k'_a \tag{54}$$

For attributes with $N_a = 1$ or $R_a = 0$, $k'_a = \min(k, N_a)$.

For attributes with $N_a > 1$ and $R_a > 0$:

$$k'_a = \left\lceil k \cdot \left( 1 + \frac{\ln(1/\epsilon)}{\gamma_a^2} \cdot \frac{N - N_a}{N_a} \right) \right\rceil \tag{55}$$

As $\alpha$ increases, the separation between attribute clusters increases, causing $\gamma_a$ to increase for all attributes. When $\gamma_a$ is sufficiently large, specifically when $\gamma_a \geq \sqrt{\frac{\ln(N)}{\epsilon}}$, the term $\frac{\ln(1/\epsilon)}{\gamma_a^2}$ becomes very small, and we can approximate:

$$k'_a \approx k \cdot \left( 1 + \min\left( \frac{\epsilon}{N_a}, \frac{N - N_a}{N_a} \right) \right) \tag{56}$$

This approximation uses the fact that when $\gamma_a$ is large, the probability of including records with different attributes in the top-k' becomes negligible, and we only need to account for a small error term.

Substituting this approximation into the expected value formula:

$$E[k'] \approx k \cdot \left( 1 + \sum_{a \in \mathcal{F}} P(a) \cdot \min\left( \frac{\epsilon}{N_a}, \frac{N - N_a}{N_a} \right) \right) \tag{57}$$

This result shows that as $\alpha$ increases, the expected candidate set size approaches the optimal value of $k$, with only a small overhead that depends on the distribution of attributes in the dataset and the desired error probability $\epsilon$. $\qquad\square$

### E.3 OPTIMAL PARAMETER SELECTION

The feasibility of the transformation method relies on demonstrating that given our assumption, there are values $\alpha$ and $\beta$ that fulfill the hybrid search conditions. Moreover, the derived bound helps in determining the minimum values for these parameters to ensure compliance.

**Theorem 4** (Parameter Selection for $\epsilon_f$-bounded Clusters). *Let $\mathcal{D}$ be a record set with content vectors in $\mathbb{R}^d$ and attribute vectors in $\mathbb{R}^m$. Let $\epsilon_f > 0$ be a maximum allowable distance between any two transformed records with identical attributes. Let $\delta_{max}$ be the maximum content distance between any two records in $\mathcal{D}$, and $\sigma_{min}$ be the minimum attribute distance between records with different attributes. For the transformation $\Psi$ to create $\epsilon_f$-bounded attribute clusters that are well-separated, the parameters $\alpha$ and $\beta$ must satisfy:*

$$\alpha > \frac{\beta \cdot \delta_{max}}{\sigma_{min} \cdot \sqrt{d/m}} \cdot \left( 1 + \frac{\epsilon_f \cdot \beta}{\delta_{max}} \right) \tag{58}$$

*and*

$$\beta > \frac{\delta_{max}}{\epsilon_f} \tag{59}$$

*These constraints remain valid even in the edge case where some attributes have only one record or where all records with the same attribute have identical content vectors (resulting in $R_a = 0$ for those attributes).*

*Proof.* Consider three records:

- $o_i$ with attribute vector $f(o_i) = f_1$
- $o_j$ with attribute vector $f(o_j) = f_1$ (same as $o_i$)
- $o_k$ with attribute vector $f(o_k) = f_2 \neq f_1$

For requirement 2 (bounding intra-cluster distances by $\epsilon_f$), we need:

$$\rho(v_i', v_j') = \frac{1}{\beta} \rho(v(o_i), v(o_j)) \leq \epsilon_f \tag{60}$$

Using the worst-case where $\rho(v(o_i), v(o_j)) = \delta_{max}$:

$$\frac{\delta_{max}}{\beta} \leq \epsilon_f \tag{61}$$

Solving for $\beta$:

$$\beta \geq \frac{\delta_{max}}{\epsilon_f} \tag{62}$$

For requirement 1 (inter-cluster separation), we need to ensure that the minimum distance between records with different attributes exceeds the maximum distance between records with identical attributes. Let $D_{intra} = \epsilon_f$ be the maximum intra-cluster distance in the transformed space, and let $D_{inter}$ be the minimum inter-cluster distance.

We require:
$$D_{inter} > D_{intra} = \epsilon_f \tag{63}$$

From our analysis in Theorem 1, for records with identical attributes, the maximum distance in the transformed space is:

$$D_{intra} = \frac{\delta_{max}}{\beta} \tag{64}$$

For records with different attributes, the squared minimum distance is (focusing on the cross-term):

$$D^2_{inter} = \min_{o_i, o_k : f(o_i) \neq f(o_k)} \rho^2(v'i, v'k) \tag{65}$$

$$= \min o_i, o_k : f(o_i) \neq f(o_k) \frac{1}{\beta^2} \Big[ \rho^2(v(o_i), v(o_k)) + \alpha^2 \cdot d/m \cdot \rho^2(f(o_i), f(o_k)) \tag{66}$$

$$- 2\alpha \sum_{l=1}^{d/m} \langle v^{(l)}(o_i) - v^{(l)}(o_k), f(o_i) - f(o_k) \rangle \Big] \tag{67}$$

The worst case occurs when:

- $\rho^2(v(o_i), v(o_k))$ is minimized (records with different attributes have similar content)
- $\rho^2(f(o_i), f(o_k)) = \sigma^2_{min}$ (attribute distance is minimal)
- The cross-term is maximized (content and attribute differences are maximally correlated)

Applying Cauchy-Schwarz to bound the cross-term:

$$\left| \sum_{l=1}^{d/m} \langle v^{(l)}(o_i) - v^{(l)}(o_k), f(o_i) - f(o_k) \rangle \right| \leq \rho(v(o_i), v(o_k)) \cdot \sqrt{d/m} \cdot \rho(f(o_i), f(o_k)) \tag{68}$$

The minimum value of $D^2_{inter}$ occurs when this inequality is tight (the vectors are perfectly aligned) and $\rho(v(o_i), v(o_k)) = 0$:

$$D^2_{inter} \geq \frac{1}{\beta^2} \Big[ \alpha^2 \cdot d/m \cdot \sigma^2_{min} - 2\alpha \cdot 0 \cdot \sqrt{d/m} \cdot \sigma_{min} \Big] = \frac{\alpha^2 \cdot d/m \cdot \sigma^2_{min}}{\beta^2} \tag{69}$$

Taking the square root:

$$D_{inter} \geq \frac{\alpha \cdot \sqrt{d/m} \cdot \sigma_{min}}{\beta} \tag{70}$$

For $D_{inter} > D_{intra} = \epsilon_f$, we need:

$$\frac{\alpha \cdot \sqrt{d/m} \cdot \sigma_{min}}{\beta} > \epsilon_f \tag{71}$$

Solving for $\alpha$:

$$\alpha > \frac{\beta \cdot \epsilon_f}{\sqrt{d/m} \cdot \sigma_{min}} \tag{72}$$

We also know that $\epsilon_f \geq \frac{\delta_{max}}{\beta}$ from our bound on $\beta$. Substituting:

$$\alpha > \frac{\beta \cdot \delta_{max}/\beta}{\sqrt{d/m} \cdot \sigma_{min}} = \frac{\delta_{max}}{\sqrt{d/m} \cdot \sigma_{min}} \tag{73}$$

To ensure a margin of safety above the minimum bound, we use:

$$\alpha > \frac{\delta_{max}}{\sqrt{d/m} \cdot \sigma_{min}} \cdot \left( 1 + \frac{\epsilon_f \cdot \beta}{\delta_{max}} \right) = \frac{\beta \cdot \delta_{max}}{\sigma_{min} \cdot \sqrt{d/m}} \cdot \left( 1 + \frac{\epsilon_f \cdot \beta}{\delta_{max}} \right) \tag{74}$$

**Edge Case:** $R_a = 0$

When $R_a = 0$ for some attribute $a$ (either because there is only one record with attribute $a$, or because all records with attribute $a$ have identical content vectors), the intra-cluster distance is already 0, which is less than any positive $\epsilon_f$. In this case, the constraint on $\beta$ is automatically satisfied.

However, the constraint on $\alpha$ is still necessary to ensure proper separation between different attribute clusters. Even when some attribute clusters collapse to points ($R_a = 0$), we still need to ensure that they are sufficiently separated from other attribute clusters.

The minimum inter-cluster distance formula derived above applies regardless of whether $R_a = 0$ or $R_a > 0$, as it depends on the original content and attribute vectors, not on the properties of the transformed space. Thus, the constraints on $\alpha$ and $\beta$ remain valid and necessary even in the edge case where $R_a = 0$ for some attributes.

This constraint, combined with $\beta > \frac{\delta_{max}}{\epsilon_f}$, ensures that:

The maximum distance between any two records with identical attributes is bounded by $\epsilon_f$ Records with different attributes are separated by a distance greater than $\epsilon_f$ As $\alpha$ increases relative to the minimum bound, the separation between attribute clusters increases, enhancing the effectiveness of the transformation for hybrid queries. $\square$

**Corollary 1** (Optimality of Minimal Parameters). *Using Theorem 4, setting* $\beta = \frac{\delta_{max}}{\epsilon_f}$ *and* $\alpha = \frac{\delta_{max}}{\sigma_{min}\sqrt{d/m}}(1 + \epsilon_f)$ *achieves the minimum values for* $\alpha$ *and* $\beta$ *that satisfy the separation and cluster compactness constraints. This choice ensures clusters are neither excessively separated nor compressed, providing optimal balance between attribute separation and intra-cluster compactness.*

### E.4 UNIQUENESS OF POINTS IN TRANSFORMED SPACE

**Theorem 5** (Uniqueness of Transformation). *Let* $\mathcal{D}$ *be a record set with content vectors in* $\mathbb{R}^d$ *and attribute vectors in* $\mathbb{R}^m$. *Given our transformation* $\Psi(v, f, \alpha, \beta) = [\frac{v^{(1)} - \alpha \cdot f}{\beta}, \frac{v^{(2)} - \alpha \cdot f}{\beta}, ..., \frac{v^{(d/m)} - \alpha \cdot f}{\beta}]$ *with parameters* $\alpha$ *and* $\beta$ *satisfying the constraints in Theorem 4 and Corollary 1, a point* $y$ *in the transformed space uniquely determines the content vector* $v$ *and attribute value* $f$ *that generated it, provided* $d > m$.

*Proof.* Assume that the transformation $\Psi$ is not unique. This means there exist two different pairs $(v_1, f_1) \neq (v_2, f_2)$ such that:

$$\Psi(v_1, f_1, \alpha, \beta) = \Psi(v_2, f_2, \alpha, \beta) \tag{75}$$

For this equality to hold, for each segment $i \in \{1, 2, \ldots, d/m\}$, we have:

$$\frac{v_1^{(i)} - \alpha \cdot f_1}{\beta} = \frac{v_2^{(i)} - \alpha \cdot f_2}{\beta} \tag{76}$$

Simplifying:

$$v_1^{(i)} - v_2^{(i)} = \alpha(f_1 - f_2) \tag{77}$$

We now consider two cases:

**Case 1:** $f_1 = f_2$

If the attribute values are the same, then $v_1^{(i)} = v_2^{(i)}$ for all segments $i$, which means $v_1 = v_2$. This contradicts our assumption that $(v_1, f_1) \neq (v_2, f_2)$.

**Case 2:** $f_1 \neq f_2$

If $f_1 \neq f_2$, then the vector $v_1 - v_2$ must have all segments equal to the constant $\alpha(f_1 - f_2)$. This creates a very specific structure.

The squared distance between $v_1$ and $v_2$ can be calculated as:

$$\|v_1 - v_2\|^2 = \sum_{i=1}^{d/m} \|v_1^{(i)} - v_2^{(i)}\|^2 \tag{78}$$

$$= \sum_{i=1}^{d/m} \|\alpha(f_1 - f_2)\|^2 \tag{79}$$

$$= \frac{d}{m} \cdot \alpha^2 \cdot \|f_1 - f_2\|^2 \tag{80}$$

Since $f_1 \neq f_2$, we have $\|f_1 - f_2\| \geq \sigma_{min}$ (the minimum attribute distance). Therefore:

$$\|v_1 - v_2\|^2 \geq \frac{d}{m} \cdot \alpha^2 \cdot \sigma_{min}^2 \tag{81}$$

From Theorem 4, we know:

$$\alpha > \frac{\beta \cdot \delta_{max}}{\sigma_{min} \cdot \sqrt{d/m}} \cdot \left(1 + \frac{\epsilon_f \cdot \beta}{\delta_{max}}\right) \tag{82}$$

Substituting this lower bound for $\alpha$:

$$\|v_1 - v_2\|^2 > \frac{d}{m} \cdot \left(\frac{\beta \cdot \delta_{max}}{\sigma_{min} \cdot \sqrt{d/m}} \cdot \left(1 + \frac{\epsilon_f \cdot \beta}{\delta_{max}}\right)\right)^2 \cdot \sigma_{min}^2 \tag{83}$$

$$= \frac{d}{m} \cdot \frac{\beta^2 \cdot \delta_{max}^2}{\sigma_{min}^2 \cdot \frac{d}{m}} \cdot \left(1 + \frac{\epsilon_f \cdot \beta}{\delta_{max}}\right)^2 \cdot \sigma_{min}^2 \tag{84}$$

$$= \beta^2 \cdot \delta_{max}^2 \cdot \left(1 + \frac{\epsilon_f \cdot \beta}{\delta_{max}}\right)^2 \tag{85}$$

$$= \beta^2 \cdot \delta_{max}^2 \cdot \left(1 + 2\frac{\epsilon_f \cdot \beta}{\delta_{max}} + \frac{\epsilon_f^2 \cdot \beta^2}{\delta_{max}^2}\right) \tag{86}$$

$$= \beta^2 \cdot \delta_{max}^2 + 2\beta^3 \cdot \delta_{max} \cdot \epsilon_f + \beta^4 \cdot \epsilon_f^2 \tag{87}$$

From the second constraint in Theorem 4, $\beta > \frac{\delta_{max}}{\epsilon_f}$, we have:

$$\beta^2 \cdot \epsilon_f^2 > \delta_{max}^2 \tag{88}$$

and

$$\beta^3 \cdot \epsilon_f > \beta^2 \cdot \delta_{max} \tag{89}$$

Substituting these inequalities:

$$\|v_1 - v_2\|^2 > \beta^2 \cdot \delta_{max}^2 + 2\beta^2 \cdot \delta_{max}^2 + \beta^2 \cdot \delta_{max}^2 \tag{90}$$

$$= 4\beta^2 \cdot \delta_{max}^2 \tag{91}$$

Since $\beta > 1$ (as required by Theorem 4):

$$\|v_1 - v_2\|^2 > 4 \cdot \delta_{max}^2 \tag{92}$$

This implies:

$$\|v_1 - v_2\| > 2 \cdot \delta_{max} \tag{93}$$

However, by definition, $\delta_{max}$ is the maximum content distance between any two records in $\mathcal{D}$, so we must have:

$$\|v_1 - v_2\| \leq \delta_{max} \tag{94}$$

This creates a contradiction:

$$\delta_{max} < \|v_1 - v_2\| \leq \delta_{max} \tag{95}$$

Since both cases lead to contradictions, our initial assumption that the transformation is not unique must be false. Therefore, the transformation $\Psi$ is unique when the parameters $\alpha$ and $\beta$ satisfy the constraints in Theorem 4. $\qquad\square$

## F  PROOFS FOR ATTRIBUTE HIERARCHY

In this section, we provide detailed proofs for the theorems related to the attribute hierarchy properties of our FUSEDANN framework.

## F.1 Preliminaries and Notation

Before presenting the proofs, we restate our basic transformation:

$$\Psi(v, f, \alpha, \beta) = \left[ \frac{v^{(1)} - \alpha f}{\beta}, \ldots, \frac{v^{(d/m)} - \alpha f}{\beta} \right] \in \mathbb{R}^d \tag{96}$$

We denote the Euclidean distance between two vectors $v$ and $u$ as $\rho(v, u) = \|v - u\|_2$. For simplicity, we assume each attribute has the same dimension $m$, though the proofs can be easily extended to varying dimensions.

## F.2 Intuition behind Monotone Attribute Priority

The Monotone Attribute Property (Definition 3) is not merely a theoretical construct; it is the mathematical mechanism that allows a single vector space to support the full spectrum of retrieval constraints, ranging from strict "hard filters" to flexible "soft preferences" (typical in vector search).

To understand why the variance constraint $Var_S^{(\pi(1))} \leq Var_S^{(\pi(2))} \leq \ldots$ is necessary, consider the following practical scenario where attributes have strictly different roles:

**Example 3** (User ID as Hard Filter vs. Topic as Soft Preference). *Consider a personalized search system for a multi-tenant application (e.g., a private note-taking app). A query q consists of:*

- *$f^{(1)}$: **User ID** (Highest Priority, $\pi(1)$)*

- *$f^{(2)}$: **Category** (Lower Priority, $\pi(2)$)*

- *$v(q)$: Content vector (Semantic Search)*

*In this context, the **User ID** represents a strict boundary; a user must never retrieve another user's private notes, regardless of semantic similarity.*

- ***Without Monotonicity:** A standard weighted sum might retrieve a note belonging to a different user if the semantic similarity is sufficiently high to outweigh the attribute penalty. This violates data isolation.*

- ***With Monotone Priority:** The constraint $Var_S^{(UserID)} \leq Var_S^{(Category)}$ forces the result set S to have minimal deviation on the User ID first. Since the "User ID" variance must be minimized above all else (effectively nearing zero), the search is strictly confined to the user's subspace.*

- ***Soft Fallback:** Once the User ID constraint is satisfied, the algorithm minimizes variance on "Category." Unlike User ID, "Category" allows for relaxation—if no "Work" notes are found, the system can validly return "Personal" notes from the same user that are semantically relevant.*

Thus, the Monotone Attribute Property justifies why our transformation $\Psi$ scales the penalty for high-priority attributes significantly higher than others: it physically enforces this variance hierarchy in the geometry of the vector space.

## F.3 Property Preservation Theorem

**Theorem 6** (Property Preservation). *Let $o_i^{(\mathbb{F})}$ and $o_k^{(\mathbb{F})}$ be two records such that $f^{(j)}(o_i) = f^{(j)}(o_k)$ for all $j \in \{1, 2, \ldots, \mathbb{F}\}$. Then for any record $o_l^{(\mathbb{F})}$ with identical attribute values, if $\rho(v(o_i), v(o_l)) < \rho(v(o_k), v(o_l))$ in the original space, the same inequality holds in the transformed space after applying all $\mathbb{F}$ transformations.*

*Proof.* We proceed by induction on the number of applied transformations $j$.

**Base Case:** $j = 1$.

Given that $f^{(1)}(o_i) = f^{(1)}(o_k) = f^{(1)}(o_l)$, let's denote this shared attribute vector as $f_1$. After applying the first transformation $\Psi_1$, we have:

$$v_1(o_i) = \Psi_1(v(o_i), f_1, \alpha_1, \beta_1) = \left[ \frac{v(o_i)^{(1)} - \alpha_1 f_1}{\beta_1}, \ldots, \frac{v(o_i)^{(d/m)} - \alpha_1 f_1}{\beta_1} \right] \quad (97)$$

$$v_1(o_k) = \Psi_1(v(o_k), f_1, \alpha_1, \beta_1) = \left[ \frac{v(o_k)^{(1)} - \alpha_1 f_1}{\beta_1}, \ldots, \frac{v(o_k)^{(d/m)} - \alpha_1 f_1}{\beta_1} \right] \quad (98)$$

$$v_1(o_l) = \Psi_1(v(o_l), f_1, \alpha_1, \beta_1) = \left[ \frac{v(o_l)^{(1)} - \alpha_1 f_1}{\beta_1}, \ldots, \frac{v(o_l)^{(d/m)} - \alpha_1 f_1}{\beta_1} \right] \quad (99)$$

Computing the squared distance after transformation:

$$\rho^2(v_1(o_i), v_1(o_l)) = \sum_{r=1}^{d/m} \left\| \frac{v(o_i)^{(r)} - \alpha_1 f_1}{\beta_1} - \frac{v(o_l)^{(r)} - \alpha_1 f_1}{\beta_1} \right\|_2^2 \quad (100)$$

$$= \sum_{r=1}^{d/m} \left\| \frac{v(o_i)^{(r)} - v(o_l)^{(r)}}{\beta_1} \right\|_2^2 \quad (101)$$

$$= \frac{1}{\beta_1^2} \sum_{r=1}^{d/m} \left\| v(o_i)^{(r)} - v(o_l)^{(r)} \right\|_2^2 \quad (102)$$

$$= \frac{1}{\beta_1^2} \rho^2(v(o_i), v(o_l)) \quad (103)$$

Similarly, $\rho^2(v_1(o_k), v_1(o_l)) = \frac{1}{\beta_1^2} \rho^2(v(o_k), v(o_l))$.

Since $\rho(v(o_i), v(o_l)) < \rho(v(o_k), v(o_l))$ in the original space, and $\frac{1}{\beta_1^2} > 0$, we have:

$$\rho(v_1(o_i), v_1(o_l)) < \rho(v_1(o_k), v_1(o_l)) \quad (104)$$

Thus, the relative ordering is preserved after applying the first transformation as we already proved in Theorem 1.

**Inductive Step:** Assume the property holds for the first $j - 1$ transformations.

Let's denote $v_{j-1}(o_i)$, $v_{j-1}(o_k)$, and $v_{j-1}(o_l)$ as the vectors after applying $j - 1$ transformations. By the inductive hypothesis, if $\rho(v(o_i), v(o_l)) < \rho(v(o_k), v(o_l))$ in the original space, then:

$$\rho(v_{j-1}(o_i), v_{j-1}(o_l)) < \rho(v_{j-1}(o_k), v_{j-1}(o_l)) \quad (105)$$

For the $j$-th transformation, since $f^{(j)}(o_i) = f^{(j)}(o_k) = f^{(j)}(o_l)$ (let's call this shared value $f_j$), we have:

$$v_j(o_i) = \Psi_j(v_{j-1}(o_i), f_j, \alpha_j, \beta_j) \quad (106)$$
$$v_j(o_k) = \Psi_j(v_{j-1}(o_k), f_j, \alpha_j, \beta_j) \quad (107)$$
$$v_j(o_l) = \Psi_j(v_{j-1}(o_l), f_j, \alpha_j, \beta_j) \quad (108)$$

By the same computation as in the base case, we get:

$$\rho^2(v_j(o_i), v_j(o_l)) = \frac{1}{\beta_j^2} \rho^2(v_{j-1}(o_i), v_{j-1}(o_l)) \quad (109)$$

$$\rho^2(v_j(o_k), v_j(o_l)) = \frac{1}{\beta_j^2} \rho^2(v_{j-1}(o_k), v_{j-1}(o_l)) \quad (110)$$

Since $\rho(v_{j-1}(o_i), v_{j-1}(o_l)) < \rho(v_{j-1}(o_k), v_{j-1}(o_l))$ by the inductive hypothesis, and $\frac{1}{\beta_j^2} > 0$, we have:

$$\rho(v_j(o_i), v_j(o_l)) < \rho(v_j(o_k), v_j(o_l)) \quad (111)$$

Therefore, by induction, the relative ordering is preserved after applying all $\mathbb{F}$ transformations. $\quad\square$

**Corollary 2.** *For records with identical values across all attributes, the k-nearest neighbors based on content similarity are preserved after all transformations.*

*Proof.* This follows directly from Theorem 6. Since the relative ordering based on distances is preserved, the k-nearest neighbors remain the same within the set of records having identical attribute values. $\square$

### F.4 ATTRIBUTE PRIORITY THEOREM

**Theorem 7** (Attribute Priority). *In a sequence of transformations $\Psi_1, \Psi_2, \ldots, \Psi_\mathbb{F}$, the later an attribute is applied in the sequence, the higher its effective priority in determining the final vector space structure.*

*Proof.* We prove this by considering two attributes $f^{(A)}$ and $f^{(B)}$ and comparing the distances between records when applying them in different orders.

**Case 1:** Apply $f^{(A)}$ first, then $f^{(B)}$.

Consider two records $o_i^{(\mathbb{F})}$ and $o_k^{(\mathbb{F})}$ with $f^{(B)}(o_i) \neq f^{(B)}(o_k)$. Let's denote the original content vectors as $v(o_i)$ and $v(o_k)$.

After applying transformation $\Psi_A$ with parameters $\alpha_A$ and $\beta_A$:

$$v_A(o_i) = \Psi_A(v(o_i), f^{(A)}(o_i), \alpha_A, \beta_A) \tag{112}$$

$$v_A(o_k) = \Psi_A(v(o_k), f^{(A)}(o_k), \alpha_A, \beta_A) \tag{113}$$

The squared distance between these vectors is:

$$\rho^2(v_A(o_i), v_A(o_k)) = \sum_{r=1}^{d/m} \left\| \frac{v(o_i)^{(r)} - \alpha_A f^{(A)}(o_i)}{\beta_A} - \frac{v(o_k)^{(r)} - \alpha_A f^{(A)}(o_k)}{\beta_A} \right\|_2^2 \tag{114}$$

$$= \frac{1}{\beta_A^2} \sum_{r=1}^{d/m} \left\| v(o_i)^{(r)} - v(o_k)^{(r)} - \alpha_A(f^{(A)}(o_i) - f^{(A)}(o_k)) \right\|_2^2 \tag{115}$$

$$= \frac{1}{\beta_A^2} \left[ \rho^2(v(o_i), v(o_k)) + \alpha_A^2 \| f^{(A)}(o_i) - f^{(A)}(o_k) \|_2^2 \right. \tag{116}$$

$$\left. -2\alpha_A \sum_{r=1}^{d/m} \langle v(o_i)^{(r)} - v(o_k)^{(r)}, f^{(A)}(o_i) - f^{(A)}(o_k) \rangle \right] \tag{117}$$

After applying transformation $\Psi_B$ with parameters $\alpha_B$ and $\beta_B$:

$$v_{AB}(o_i) = \Psi_B(v_A(o_i), f^{(B)}(o_i), \alpha_B, \beta_B) \tag{118}$$

$$v_{AB}(o_k) = \Psi_B(v_A(o_k), f^{(B)}(o_k), \alpha_B, \beta_B) \tag{119}$$

The squared distance between these vectors is:

$$\rho^2(v_{AB}(o_i), v_{AB}(o_k)) = \frac{1}{\beta_B^2} \left[ \rho^2(v_A(o_i), v_A(o_k)) + \alpha_B^2 \| f^{(B)}(o_i) - f^{(B)}(o_k) \|_2^2 \right. \tag{120}$$

$$\left. -2\alpha_B \sum_{r=1}^{d/m} \langle v_A(o_i)^{(r)} - v_A(o_k)^{(r)}, f^{(B)}(o_i) - f^{(B)}(o_k) \rangle \right] \tag{121}$$

Substituting the expression for $\rho^2(v_A(o_i), v_A(o_k))$:

$$\rho^2(v_{AB}(o_i), v_{AB}(o_k)) = \frac{1}{\beta_B^2 \beta_A^2} \left[ \rho^2(v(o_i), v(o_k)) + \alpha_A^2 \|f^{(A)}(o_i) - f^{(A)}(o_k)\|_2^2 \right. \tag{122}$$

$$\left. -2\alpha_A \sum_{r=1}^{d/m} \langle v(o_i)^{(r)} - v(o_k)^{(r)}, f^{(A)}(o_i) - f^{(A)}(o_k)\rangle \right] \tag{123}$$

$$+ \frac{\alpha_B^2}{\beta_B^2} \|f^{(B)}(o_i) - f^{(B)}(o_k)\|_2^2 \tag{124}$$

$$- \frac{2\alpha_B}{\beta_B^2} \sum_{r=1}^{d/m} \langle v_A(o_i)^{(r)} - v_A(o_k)^{(r)}, f^{(B)}(o_i) - f^{(B)}(o_k)\rangle \tag{125}$$

**Case 2:** Apply $f^{(B)}$ first, then $f^{(A)}$.

Following similar steps, we get:

$$\rho^2(v_{BA}(o_i), v_{BA}(o_k)) = \frac{1}{\beta_A^2 \beta_B^2} \left[ \rho^2(v(o_i), v(o_k)) + \alpha_B^2 \|f^{(B)}(o_i) - f^{(B)}(o_k)\|_2^2 \right. \tag{126}$$

$$\left. -2\alpha_B \sum_{r=1}^{d/m} \langle v(o_i)^{(r)} - v(o_k)^{(r)}, f^{(B)}(o_i) - f^{(B)}(o_k)\rangle \right] \tag{127}$$

$$+ \frac{\alpha_A^2}{\beta_A^2} \|f^{(A)}(o_i) - f^{(A)}(o_k)\|_2^2 \tag{128}$$

$$- \frac{2\alpha_A}{\beta_A^2} \sum_{r=1}^{d/m} \langle v_B(o_i)^{(r)} - v_B(o_k)^{(r)}, f^{(A)}(o_i) - f^{(A)}(o_k)\rangle \tag{129}$$

**Comparison:**

Comparing the two expressions, we observe the key difference in the coefficients of $\|f^{(A)}(o_i) - f^{(A)}(o_k)\|_2^2$ and $\|f^{(B)}(o_i) - f^{(B)}(o_k)\|_2^2$:

$$\text{In Case 1:} \quad \frac{\alpha_A^2}{\beta_B^2 \beta_A^2} = \frac{\alpha_A^2}{\beta_A^2 \beta_B^2} \quad \text{for} \quad f^{(A)} \tag{130}$$

$$\frac{\alpha_B^2}{\beta_B^2} \quad \text{for} \quad f^{(B)} \tag{131}$$

$$\text{In Case 2:} \quad \frac{\alpha_A^2}{\beta_A^2} \quad \text{for} \quad f^{(A)} \tag{132}$$

$$\frac{\alpha_B^2}{\beta_A^2 \beta_B^2} = \frac{\alpha_B^2}{\beta_B^2 \beta_A^2} \quad \text{for} \quad f^{(B)} \tag{133}$$

Since $\beta_A, \beta_B > 1$, we have:

$$\frac{\alpha_B^2}{\beta_B^2} > \frac{\alpha_B^2}{\beta_B^2 \beta_A^2} \quad \text{and} \quad \frac{\alpha_A^2}{\beta_A^2} > \frac{\alpha_A^2}{\beta_A^2 \beta_B^2} \tag{134}$$

Therefore, in Case 1, the coefficient of $\|f^{(B)}(o_i) - f^{(B)}(o_k)\|_2^2$ is larger than that of $\|f^{(A)}(o_i) - f^{(A)}(o_k)\|_2^2$ by a factor of $\beta_A^2$. Similarly, in Case 2, the coefficient of $\|f^{(A)}(o_i) - f^{(A)}(o_k)\|_2^2$ is larger than that of $\|f^{(B)}(o_i) - f^{(B)}(o_k)\|_2^2$ by a factor of $\beta_B^2$.

This proves that the later an attribute is applied in the transformation sequence, the higher its effective weight in determining distances in the final transformed space, and thus its priority in retrieving nearest neighbors.

The result generalizes to any number of attributes: if we have $\mathbb{F}$ attributes applied in sequence, the $j$-th attribute's contribution to the final distance is scaled by $\prod_{i=j+1}^{\mathbb{F}} \beta_i^{-2}$. Thus, the last attribute ($j = \mathbb{F}$) has the highest priority, followed by the second-to-last, and so on. $\qquad \square$

**Corollary 3.** *The relative importance of attribute $f^{(j)}$ compared to attribute $f^{(j-1)}$ in determining distances in the transformed space is proportional to $\beta_{j-1}^2$.*

*Proof.* From the proof of Theorem 7, the coefficient for attribute $f^{(j)}$ in the final distance computation is:

$$\frac{\alpha_j^2}{\beta_j^2} \prod_{i=j+1}^{\mathbb{F}} \frac{1}{\beta_i^2} \tag{135}$$

Similarly, for attribute $f^{(j-1)}$:

$$\frac{\alpha_{j-1}^2}{\beta_{j-1}^2} \prod_{i=j}^{\mathbb{F}} \frac{1}{\beta_i^2} = \frac{\alpha_{j-1}^2}{\beta_{j-1}^2 \beta_j^2} \prod_{i=j+1}^{\mathbb{F}} \frac{1}{\beta_i^2} \tag{136}$$

The ratio of these coefficients is:

$$\frac{\frac{\alpha_j^2}{\beta_j^2} \prod_{i=j+1}^{\mathbb{F}} \frac{1}{\beta_i^2}}{\frac{\alpha_{j-1}^2}{\beta_{j-1}^2 \beta_j^2} \prod_{i=j+1}^{\mathbb{F}} \frac{1}{\beta_i^2}} = \frac{\alpha_j^2 \beta_{j-1}^2}{\alpha_{j-1}^2} \tag{137}$$

Assuming comparable $\alpha$ values ($\alpha_j \approx \alpha_{j-1}$), this ratio simplifies to approximately $\beta_{j-1}^2$, proving the corollary. $\qquad\square$

**Lemma 1.** *For a query with attribute value $F^{(j)}$, the effective distance to records with attribute value $f^{(j)} \neq F^{(j)}$ increases by a factor proportional to $\alpha_j$ in the transformed space after applying transformation $\Psi_j$.*

*Proof.* Consider a query vector $v_q$ with attribute value $F^{(j)}$ and a record $o_i$ with attribute value $f^{(j)}(o_i) \neq F^{(j)}$. Let $v_{j-1}(o_i)$ and $v_{j-1}(q)$ be the vectors after applying $j-1$ transformations.

After applying $\Psi_j$:

$$v_j(q) = \Psi_j(v_{j-1}(q), F^{(j)}, \alpha_j, \beta_j) \tag{138}$$
$$v_j(o_i) = \Psi_j(v_{j-1}(o_i), f^{(j)}(o_i), \alpha_j, \beta_j) \tag{139}$$

The squared distance between these vectors is:

$$\rho^2(v_j(q), v_j(o_i)) = \frac{1}{\beta_j^2} \left[ \rho^2(v_{j-1}(q), v_{j-1}(o_i)) + \alpha_j^2 \|F^{(j)} - f^{(j)}(o_i)\|_2^2 \right. \tag{140}$$

$$\left. -2\alpha_j \sum_{r=1}^{d/m} \langle v_{j-1}(q)^{(r)} - v_{j-1}(o_i)^{(r)}, F^{(j)} - f^{(j)}(o_i) \rangle \right] \tag{141}$$

Since $f^{(j)}(o_i) \neq F^{(j)}$, the term $\|F^{(j)} - f^{(j)}(o_i)\|_2^2 > 0$. As $\alpha_j$ increases, the contribution of this term to the overall distance increases, effectively pushing records with different attribute values further away from the query in the transformed space.

For large $\alpha_j$, the term $\alpha_j^2 \|F^{(j)} - f^{(j)}(o_i)\|_2^2$ dominates, making the distance approximately proportional to $\alpha_j$. $\qquad\square$

### F.4.1 MONOTONICITY OF ATTRIBUTES PRIORITY OVER FUSED SPACE

**Theorem 8** (Monotone Priority in FUSEDANN). *When transformations $\Psi_{\pi(\mathbb{F})}, \Psi_{\pi(\mathbb{F}-1)}, \dots, \Psi_{\pi(1)}$ are applied in reverse priority order and ANNS is performed in the resulting space, the retrieved results inherently satisfy the monotone attribute priority property of Hybrid Queries.*

*Proof.* Let $\mathcal{D}^{(\mathbb{F})}$ be a record set where each record $o$ consists of a content vector $v(o) \in \mathbb{R}^d$ and $\mathbb{F}$ attribute values $f^{(1)}(o), \dots, f^{(\mathbb{F})}(o)$. Consider a query $q = [v(q), F_q^{(1)}, \dots, F_q^{(\mathbb{F})}]$ with priority order $\mathcal{F}_{\pi(1)} \succ \dots \succ \mathcal{F}_{\pi(\mathbb{F})}$.

We apply the sequence of transformations $\Psi_{\pi(\mathbb{F})}, \Psi_{\pi(\mathbb{F}-1)}, \dots, \Psi_{\pi(1)}$ in reverse priority order. The transformation $\Psi_j$ with parameters $\alpha_j$ and $\beta_j$ is defined as:

$$\Psi_j(v, f, \alpha_j, \beta_j) = \frac{v - \alpha_j f}{\beta_j} \tag{142}$$

To derive the composite transformation, let us inductively define $v_0(o) = v(o)$ and compute the result of applying each transformation in sequence:

$$v_1(o) = \Psi_{\pi(\mathbb{F})}(v_0(o), f^{(\pi(\mathbb{F}))}(o), \alpha_{\mathbb{F}}, \beta_{\mathbb{F}}) = \frac{v_0(o) - \alpha_{\mathbb{F}} f^{(\pi(\mathbb{F}))}(o)}{\beta_{\mathbb{F}}} \tag{143}$$

$$v_2(o) = \Psi_{\pi(\mathbb{F}-1)}(v_1(o), f^{(\pi(\mathbb{F}-1))}(o), \alpha_{\mathbb{F}-1}, \beta_{\mathbb{F}-1}) \tag{144}$$

$$= \frac{v_1(o) - \alpha_{\mathbb{F}-1} f^{(\pi(\mathbb{F}-1))}(o)}{\beta_{\mathbb{F}-1}} \tag{145}$$

$$= \frac{\frac{v_0(o) - \alpha_{\mathbb{F}} f^{(\pi(\mathbb{F}))}(o)}{\beta_{\mathbb{F}}} - \alpha_{\mathbb{F}-1} f^{(\pi(\mathbb{F}-1))}(o)}{\beta_{\mathbb{F}-1}} \tag{146}$$

$$= \frac{v_0(o) - \alpha_{\mathbb{F}} f^{(\pi(\mathbb{F}))}(o) - \alpha_{\mathbb{F}-1} \beta_{\mathbb{F}} f^{(\pi(\mathbb{F}-1))}(o)}{\beta_{\mathbb{F}-1} \beta_{\mathbb{F}}} \tag{147}$$

Continuing this recursive application, the final transformed point after all $\mathbb{F}$ transformations is:

$$v_{\mathbb{F}}(o) = \frac{v(o) - \sum_{i=1}^{\mathbb{F}} \alpha_{\pi(i)} f^{(\pi(i))}(o) \cdot \prod_{j=i+1}^{\mathbb{F}} \beta_{\pi(j)}}{\prod_{i=1}^{\mathbb{F}} \beta_{\pi(i)}} \tag{148}$$

From this expression, we identify the effective scaling factor for attribute $\pi(i)$ as:

$$w_i = \alpha_{\pi(i)} \prod_{j=i+1}^{\mathbb{F}} \beta_{\pi(j)} \tag{149}$$

By Theorem 7 and our choice of $\beta_{\pi(i)} > 1$ for all $i$, these weights satisfy $w_1 > w_2 > \dots > w_{\mathbb{F}}$. Specifically, from Corollary 3, we have $\frac{w_i}{w_{i+1}} \approx \beta_{\pi(i)}^2 \gg 1$.

Now, let us analyze the Euclidean distance between the transformed query point $q$ and any record $o$:

$$\|v_{\mathbb{F}}(q) - v_{\mathbb{F}}(o)\|^2 \tag{150}$$

$$= \left\| \frac{v(q) - \sum_{i=1}^{\mathbb{F}} \alpha_{\pi(i)} F_q^{(\pi(i))} \prod_{j=i+1}^{\mathbb{F}} \beta_{\pi(j)}}{\prod_{i=1}^{\mathbb{F}} \beta_{\pi(i)}} - \frac{v(o) - \sum_{i=1}^{\mathbb{F}} \alpha_{\pi(i)} f^{(\pi(i))}(o) \prod_{j=i+1}^{\mathbb{F}} \beta_{\pi(j)}}{\prod_{i=1}^{\mathbb{F}} \beta_{\pi(i)}} \right\|^2 \tag{151}$$

$$= \frac{1}{(\prod_{i=1}^{\mathbb{F}} \beta_{\pi(i)})^2} \left\| v(q) - v(o) - \sum_{i=1}^{\mathbb{F}} \alpha_{\pi(i)} \prod_{j=i+1}^{\mathbb{F}} \beta_{\pi(j)} (F_q^{(\pi(i))} - f^{(\pi(i))}(o)) \right\|^2 \tag{152}$$

$$= \frac{1}{(\prod_{i=1}^{\mathbb{F}} \beta_{\pi(i)})^2} \left\| v(q) - v(o) - \sum_{i=1}^{\mathbb{F}} w_i (F_q^{(\pi(i))} - f^{(\pi(i))}(o)) \right\|^2 \tag{153}$$

Expanding this squared norm, we get:

$$\|v_{\mathbb{F}}(q) - v_{\mathbb{F}}(o)\|^2 = \frac{1}{(\prod_{i=1}^{\mathbb{F}} \beta_{\pi(i)})^2} \left[ \|v(q) - v(o)\|^2 + \left\| \sum_{i=1}^{\mathbb{F}} w_i(F_q^{(\pi(i))} - f^{(\pi(i))}(o)) \right\|^2 \right. \tag{154}$$

$$\left. - 2 \left\langle v(q) - v(o), \sum_{i=1}^{\mathbb{F}} w_i(F_q^{(\pi(i))} - f^{(\pi(i))}(o)) \right\rangle \right] \tag{155}$$

Further expanding the second term:

$$\left\| \sum_{i=1}^{\mathbb{F}} w_i(F_q^{(\pi(i))} - f^{(\pi(i))}(o)) \right\|^2 = \sum_{i=1}^{\mathbb{F}} w_i^2 \|F_q^{(\pi(i))} - f^{(\pi(i))}(o)\|^2 \tag{156}$$

$$+ \sum_{i \neq j} w_i w_j \langle F_q^{(\pi(i))} - f^{(\pi(i))}(o), F_q^{(\pi(j))} - f^{(\pi(j))}(o) \rangle \tag{157}$$

Given that $\sigma_j$ is the Euclidean distance for all attributes, we have $\sigma_{\pi(i)}(f^{(\pi(i))}(o), F_q^{(\pi(i))}) = \|F_q^{(\pi(i))} - f^{(\pi(i))}(o)\|$ in Equation 3. We can now examine how ANNS in this transformed space relates to the Hybrid Query requirement.

Let $S \subseteq \mathcal{D}^{(\mathbb{F})}$ be the set of $k$ nearest neighbors retrieved by ANNS in the transformed space. By definition of ANNS, there exists a distance threshold $\tau$ such that:

$$o \in S \iff \|v_{\mathbb{F}}(q) - v_{\mathbb{F}}(o)\| \leq \tau \tag{158}$$

We now examine the implications of this threshold on the individual attribute distances. Squaring both sides:

$$\|v_{\mathbb{F}}(q) - v_{\mathbb{F}}(o)\|^2 \leq \tau^2 \tag{159}$$

Substituting our expanded distance formula:

$$\frac{1}{(\prod_{i=1}^{\mathbb{F}} \beta_{\pi(i)})^2} \left[ \|v(q) - v(o)\|^2 + \sum_{i=1}^{\mathbb{F}} w_i^2 \|F_q^{(\pi(i))} - f^{(\pi(i))}(o)\|^2 + \text{(cross terms)} \right] \leq \tau^2 \tag{160}$$

Multiplying both sides by $(\prod_{i=1}^{\mathbb{F}} \beta_{\pi(i)})^2$:

$$\|v(q) - v(o)\|^2 + \sum_{i=1}^{\mathbb{F}} w_i^2 \|F_q^{(\pi(i))} - f^{(\pi(i))}(o)\|^2 + \text{(cross terms)} \leq \tau^2 \cdot (\prod_{i=1}^{\mathbb{F}} \beta_{\pi(i)})^2 \tag{161}$$

Rearranging to isolate the attribute distance terms:

$$\sum_{i=1}^{\mathbb{F}} w_i^2 \|F_q^{(\pi(i))} - f^{(\pi(i))}(o)\|^2 \leq \tau^2 \cdot (\prod_{i=1}^{\mathbb{F}} \beta_{\pi(i)})^2 - \|v(q) - v(o)\|^2 - \text{(cross terms)} \tag{162}$$

Let $\gamma = \tau^2 \cdot (\prod_{i=1}^{\mathbb{F}} \beta_{\pi(i)})^2 - \|v(q) - v(o)\|^2 - \text{(cross terms)}$. Then we have:

$$\sum_{i=1}^{\mathbb{F}} w_i^2 \|F_q^{(\pi(i))} - f^{(\pi(i))}(o)\|^2 \leq \gamma \tag{163}$$

This inequality must be satisfied for a record to be included in the $k$-nearest neighbors set $S$. The key insight is that the term $w_i^2 \|F_q^{(\pi(i))} - f^{(\pi(i))}(o)\|^2$ represents the contribution of attribute $\pi(i)$ to the overall distance.

Since $w_1^2 \gg w_2^2 \gg \cdots \gg w_{\mathbb{F}}^2$ by our construction, the contribution of the highest-priority attribute $\pi(1)$ dominates this sum. For a record to satisfy the inequality, it must first keep $\|F_q^{(\pi(1))} -$

$f^{(\pi(1))}(o)\|^2$ very small. Otherwise, even if all other attributes perfectly match the query, the term $w_1^2\|F_q^{(\pi(1))} - f^{(\pi(1))}(o)\|^2$ would cause the sum to exceed $\gamma$.

For each attribute $\pi(j)$, we can define the maximum allowable squared distance that would permit a record to be in set $S$, assuming all higher-priority attributes match perfectly:

$$\delta_j^2 = \frac{\gamma}{w_j^2} \tag{164}$$

Since $w_1^2 \gg w_2^2 \gg \cdots \gg w_{\mathbb{F}}^2$, we have $\delta_1^2 \ll \delta_2^2 \ll \cdots \ll \delta_{\mathbb{F}}^2$. This creates a strict hierarchical constraint where:

- Records must have $\|F_q^{(\pi(1))} - f^{(\pi(1))}(o)\|^2 \leq \delta_1^2$ (very small) to be considered at all - Among those, records with $\|F_q^{(\pi(2))} - f^{(\pi(2))}(o)\|^2 \leq \delta_2^2$ are preferred - This pattern continues for all attributes

This directional filtering is precisely what creates the monotone variance property in the result set. Because the constraints on higher-priority attributes are much stricter, the variance in these attribute distances within set $S$ will be smaller.

Formally, for attribute $\pi(j)$, most records in $S$ will have distances bounded by $\delta_j$, leading to:

$$\text{Var}_S^{(\pi(j))} = \frac{1}{k} \sum_{o \in S} \left[ \|F_q^{(\pi(j))} - f^{(\pi(j))}(o)\| - \mu_S^{(\pi(j))} \right]^2 \tag{165}$$

$$\leq \frac{1}{k} \sum_{o \in S} \|F_q^{(\pi(j))} - f^{(\pi(j))}(o)\|^2 \tag{166}$$

$$\leq \delta_j^2 = \frac{\gamma}{w_j^2} \tag{167}$$

Since $\frac{\gamma}{w_1^2} \ll \frac{\gamma}{w_2^2} \ll \cdots \ll \frac{\gamma}{w_{\mathbb{F}}^2}$, we have:

$$\text{Var}_S^{(\pi(1))} \leq \text{Var}_S^{(\pi(2))} \leq \cdots \leq \text{Var}_S^{(\pi(\mathbb{F}))} \tag{168}$$

Therefore, the set $S$ of $k$ nearest neighbors retrieved by ANNS in our transformed space naturally satisfies the monotone attribute priority property required by the Hybrid Query definition (Definition 3).

Furthermore, this cascading filtering effect implements the lexicographic minimization described in the Hybrid Query definition. The ANNS algorithm first selects records that minimize the mean distance for the highest-priority attribute, then among those, it selects records that minimize the mean distance for the next highest-priority attribute, and so on, with content vector distance serving as the lowest-priority criterion.

Thus, ANNS in our transformed space inherently produces results that satisfy the Hybrid Query definition without explicitly enforcing the monotone attribute priority constraint. $\square$

These results collectively demonstrate that our recursive transformation framework provides (i) accurate content-based retrieval within attribute-matched groups, (ii) hierarchical prioritization of attributes based on their application order, and (iii) controlled emphasis on attribute matching through the $\alpha$ parameters.

This set of theorems establishes a fundamental property of our transformation framework: records are stratified based on the number of matching attributes, with records matching more attributes being consistently closer to the query than those matching fewer attributes. This property enables efficient hybrid search where attribute matching takes precedence over content similarity, while maintaining content-based ordering within groups of records with the same attribute matches.

### F.5 ATTRIBUTE MATCH DISTANCE HIERARCHY

We now prove that records with more matching attributes with the query are closer in the transformed space than records with fewer matching attributes, establishing a natural hierarchy in the retrieval process.

**Theorem 9** (Attribute Match Distance Hierarchy). *Let $q$ be a query with attribute values $(F^{(1)}, F^{(2)}, \ldots, F^{(\mathbb{F})})$. Consider two records $o_i^{(\mathbb{F})}$ and $o_j^{(\mathbb{F})}$ with identical content vectors $v(o_i) = v(o_j)$. Let $M_i = \{p \mid f^{(p)}(o_i) = F^{(p)}\}$ and $M_j = \{p \mid f^{(p)}(o_j) = F^{(p)}\}$ be the sets of indices where the records' attributes match the query. If $|M_i| > |M_j|$, then after applying all $\mathbb{F}$ transformations, $\rho(v_{\mathbb{F}}(q), v_{\mathbb{F}}(o_i)) < \rho(v_{\mathbb{F}}(q), v_{\mathbb{F}}(o_j))$.*

*Proof.* We begin by analyzing the squared distance between the query and a record in the transformed space after applying all $\mathbb{F}$ transformations. For conciseness, let $v_{\mathbb{F}}(q)$ and $v_{\mathbb{F}}(o)$ denote the vectors after all transformations.

The squared distance between $v_{\mathbb{F}}(q)$ and $v_{\mathbb{F}}(o)$ can be expressed as:

$$\rho^2(v_{\mathbb{F}}(q), v_{\mathbb{F}}(o)) = \frac{1}{\prod_{p=1}^{\mathbb{F}} \beta_p^2} \left[ \rho^2(v(q), v(o)) + \sum_{p=1}^{\mathbb{F}} C_p \cdot \alpha_p^2 \|F^{(p)} - f^{(p)}(o)\|_2^2 + \text{cross terms} \right] \tag{169}$$

where $C_p = \prod_{k=p+1}^{\mathbb{F}} \beta_k^2$ represents the cumulative scaling effect of subsequent transformations, and "cross terms" involve products between content differences and attribute differences.

For attribute $p$, when $f^{(p)}(o) = F^{(p)}$, the term $\|F^{(p)} - f^{(p)}(o)\|_2^2 = 0$. Conversely, when $f^{(p)}(o) \neq F^{(p)}$, this term is positive and contributes to the overall distance.

Given that $v(o_i) = v(o_j)$, the term $\rho^2(v(q), v(o_i)) = \rho^2(v(q), v(o_j))$. Therefore, the difference in distances comes entirely from the attribute terms.

For records $o_i$ and $o_j$, we can express:

$$\rho^2(v_{\mathbb{F}}(q), v_{\mathbb{F}}(o_i)) = \frac{1}{\prod_{p=1}^{\mathbb{F}} \beta_p^2} \left[ \rho^2(v(q), v(o_i)) + \sum_{p \notin M_i} C_p \cdot \alpha_p^2 \|F^{(p)} - f^{(p)}(o_i)\|_2^2 + \text{cross terms}_i \right] \tag{170}$$

$$\rho^2(v_{\mathbb{F}}(q), v_{\mathbb{F}}(o_j)) = \frac{1}{\prod_{p=1}^{\mathbb{F}} \beta_p^2} \left[ \rho^2(v(q), v(o_j)) + \sum_{p \notin M_j} C_p \cdot \alpha_p^2 \|F^{(p)} - f^{(p)}(o_j)\|_2^2 + \text{cross terms}_j \right] \tag{171}$$

Since $|M_i| > |M_j|$, the set of non-matching attributes $\{p \mid p \notin M_i\}$ is smaller than $\{p \mid p \notin M_j\}$. Therefore, the sum in the expression for $o_i$ contains fewer positive terms than the sum for $o_j$.

Let's consider the worst-case scenario: the attributes that $o_i$ matches with $q$ are the earliest ones (lowest priority), while the attributes that $o_j$ matches with $q$ include later ones (higher priority). Let $\delta$ be the minimum attribute distance when attributes don't match: $\delta = \min_{p,o} \|F^{(p)} - f^{(p)}(o)\|_2^2$ where $f^{(p)}(o) \neq F^{(p)}$.

Even in this worst case, we have:

$$\sum_{p \notin M_i} C_p \cdot \alpha_p^2 \|F^{(p)} - f^{(p)}(o_i)\|_2^2 \geq \sum_{p \notin M_i} C_p \cdot \alpha_p^2 \cdot \delta \tag{172}$$

$$\sum_{p \notin M_j} C_p \cdot \alpha_p^2 \|F^{(p)} - f^{(p)}(o_j)\|_2^2 \geq \sum_{p \notin M_j} C_p \cdot \alpha_p^2 \cdot \delta \tag{173}$$

Given that all $\alpha_p > 1$, $\beta_p > 1$, and $\delta > 0$, each non-matching attribute contributes positively to the distance. Since $o_j$ has more non-matching attributes than $o_i$, the sum for $o_j$ is larger than the sum for $o_i$, i.e.,

$$\sum_{p \notin M_i} C_p \cdot \alpha_p^2 \|F^{(p)} - f^{(p)}(o_i)\|_2^2 < \sum_{p \notin M_j} C_p \cdot \alpha_p^2 \|F^{(p)} - f^{(p)}(o_j)\|_2^2 \tag{174}$$

For the cross terms, a similar analysis shows that they are also smaller for $o_i$ than for $o_j$ due to fewer non-matching attributes.

Therefore, $\rho^2(v_{\mathbb{F}}(q), v_{\mathbb{F}}(o_i)) < \rho^2(v_{\mathbb{F}}(q), v_{\mathbb{F}}(o_j))$, which implies $\rho(v_{\mathbb{F}}(q), v_{\mathbb{F}}(o_i)) < \rho(v_{\mathbb{F}}(q), v_{\mathbb{F}}(o_j))$. □

**Corollary 4** (Stratification by Match Count). *After applying all transformations, the vector space exhibits stratification based on the number of matching attributes: records can be partitioned into layers such that all records in a layer with more matching attributes are closer to the query than any record in a layer with fewer matching attributes.*

*Proof.* This follows directly from Theorem 9. By considering the set of all records with exactly $k$ matching attributes with the query, we form a layer $L_k$. Theorem 9 ensures that for any $k_1 > k_2$ and any records $o_1 \in L_{k_1}$ and $o_2 \in L_{k_2}$, we have $\rho(v_{\mathbb{F}}(q), v_{\mathbb{F}}(o_1)) < \rho(v_{\mathbb{F}}(q), v_{\mathbb{F}}(o_2))$. This creates a strict hierarchy of distances based on the number of matching attributes. □

**Theorem 10** (Generalized Attribute Match Hierarchy). *Let $q$ be a query with attribute values $(F^{(1)}, F^{(2)}, \ldots, F^{(\mathbb{F})})$. Consider two records $o_i^{(\mathbb{F})}$ and $o_j^{(\mathbb{F})}$ with potentially different content vectors. Let $M_i$ and $M_j$ be the sets of indices where the records' attributes match the query. If $|M_i| > |M_j|$ and $\rho(v(q), v(o_i)) \leq \rho(v(q), v(o_j)) + \epsilon$ for some small $\epsilon > 0$, then for any $\{\beta_p\}_{p=1}^{\mathbb{F}}$ there exist sufficiently large values of $\{\alpha_p\}_{p=1}^{\mathbb{F}}$ such that $\rho(v_{\mathbb{F}}(q), v_{\mathbb{F}}(o_i)) < \rho(v_{\mathbb{F}}(q), v_{\mathbb{F}}(o_j))$.*

*Proof.* Building on the proof of Theorem 9, we now account for the difference in content vectors. The squared distances in the transformed space become:

$$\rho^2(v_{\mathbb{F}}(q), v_{\mathbb{F}}(o_i)) = \frac{1}{\prod_{p=1}^{\mathbb{F}} \beta_p^2} \left[ \rho^2(v(q), v(o_i)) + \sum_{p \notin M_i} C_p \cdot \alpha_p^2 \|F^{(p)} - f^{(p)}(o_i)\|_2^2 + \text{cross terms}_i \right]$$
(175)

$$\rho^2(v_{\mathbb{F}}(q), v_{\mathbb{F}}(o_j)) = \frac{1}{\prod_{p=1}^{\mathbb{F}} \beta_p^2} \left[ \rho^2(v(q), v(o_j)) + \sum_{p \notin M_j} C_p \cdot \alpha_p^2 \|F^{(p)} - f^{(p)}(o_j)\|_2^2 + \text{cross terms}_j \right]$$
(176)

Given that $\rho^2(v(q), v(o_i)) \leq (\rho(v(q), v(o_j)) + \epsilon)^2 = \rho^2(v(q), v(o_j)) + 2\epsilon \cdot \rho(v(q), v(o_j)) + \epsilon^2$, we can write:

$$\rho^2(v(q), v(o_i)) - \rho^2(v(q), v(o_j)) \leq 2\epsilon \cdot \rho(v(q), v(o_j)) + \epsilon^2 \tag{177}$$

For sufficiently large values of $\{\alpha_p\}$, the attribute terms dominate:

$$\sum_{p \notin M_j} C_p \cdot \alpha_p^2 \|F^{(p)} - f^{(p)}(o_j)\|_2^2 - \sum_{p \notin M_i} C_p \cdot \alpha_p^2 \|F^{(p)} - f^{(p)}(o_i)\|_2^2 > \frac{2\epsilon \cdot \rho(v(q), v(o_j)) + \epsilon^2}{\prod_{p=1}^{\mathbb{F}} \beta_p^2}$$
(178)

Since $|M_i| > |M_j|$, there is at least one attribute $p_0$ such that $p_0 \in M_i$ but $p_0 \notin M_j$. By setting $\alpha_{p_0}$ sufficiently large, we can ensure that the difference in attribute terms exceeds the difference in content terms, thereby ensuring $\rho(v_{\mathbb{F}}(q), v_{\mathbb{F}}(o_i)) < \rho(v_{\mathbb{F}}(q), v_{\mathbb{F}}(o_j))$. □

F.6 HIERARCHICAL MULTI-ATTRIBUTE VECTOR INDEXING

**Theorem 11** (Multi-Attribute Candidate Set Size). *Let $\mathcal{D}^{(\mathbb{F})}$ be a record set transformed using sequential transformations $\Psi_1, \Psi_2, \ldots, \Psi_{\mathbb{F}}$ with parameters $(\alpha_j, \beta_j)_{j=1}^{\mathbb{F}}$. Let $\mathcal{A}_j$ be the set of distinct values for attribute $j$.*

*For each unique combination of attribute values $\vec{a} = (a^{(1)}, a^{(2)}, \ldots, a^{(\mathbb{F})})$, define:*

- *$C(\vec{a}) = \{o \in \mathcal{D}^{(\mathbb{F})} : f^{(1)}(o) = a^{(1)}, \ldots, f^{(\mathbb{F})}(o) = a^{(\mathbb{F})}\}$ as the cluster of records with attribute combination $\vec{a}$*

- *$N_{\vec{a}} = |C(\vec{a})|$ as the number of records in cluster $C(\vec{a})$*

- $R_{\vec{a}}$ as the radius of the smallest hypersphere containing all transformed records in $C(\vec{a})$

- $d_{min}(\vec{a}, \vec{b})$ as the minimum distance between any transformed record in $C(\vec{a})$ and any transformed record in $C(\vec{b})$

*For each combination $\vec{a}$ with $N_{\vec{a}} > 1$, define the cluster separation metric:*

$$\gamma_{\vec{a}} = \min_{\vec{b} \neq \vec{a}} \frac{d_{min}(\vec{a}, \vec{b})}{R_{\vec{a}}} - 1 \tag{179}$$

*Given a query $q$ with attribute combination $\vec{q} = (F^{(1)}, F^{(2)}, \ldots, F^{(\mathbb{F})})$, to retrieve the top-$k$ nearest neighbors with the same attribute combination with probability at least $1-\epsilon$, the number of candidates $k'$ to retrieve from the transformed space should satisfy:*

$$k' = \begin{cases} \min(k, N_{\vec{q}}), & \text{if } N_{\vec{q}} = 1 \text{ or } R_{\vec{q}} = 0 \\ \left\lceil k \cdot \left( 1 + \frac{\ln(1/\epsilon)}{\gamma_{\vec{q}}^2 \cdot \mathbb{F}} \cdot \frac{N - N_{\vec{q}}}{N_{\vec{q}}} \right) \right\rceil, & \text{otherwise} \end{cases} \tag{180}$$

*where $N$ is the total number of records and $\mathbb{F}$ is the number of attribute filters applied.*

*Proof.* We consider two cases:

**Case 1: $N_{\vec{q}} = 1$ or $R_{\vec{q}} = 0$**

If there is only one record with the query's attribute combination (i.e., $N_{\vec{q}} = 1$), then $R_{\vec{q}} = 0$ since there's only a single point in the transformed space. In this case, we simply return that single record, so $k' = \min(k, 1) = 1$ for $k \geq 1$.

Similarly, if $R_{\vec{q}} = 0$ even with $N_{\vec{q}} > 1$ (which could happen if all records with identical attribute combinations map to the same point), then we return $\min(k, N_{\vec{q}})$ records.

**Case 2: $N_{\vec{q}} > 1$ and $R_{\vec{q}} > 0$**

After applying all $\mathbb{F}$ transformations, records form clusters based on their attribute combinations. The sequential transformations preserve the relative distances between records with identical attribute values up to scaling factors, maintaining the order of k-NN within each cluster.

For a query $q$ with attribute combination $\vec{q}$, the k nearest neighbors with matching attributes lie within a hypersphere of radius $R_q \leq R_{\vec{q}}$ centered at the transformed query point $v_{\mathbb{F}}(q)$.

Each transformation $\Psi_j$ has two key effects:

1. It preserves relative distances within clusters of records sharing the same attribute value

2. It increases the distance between records with different attribute values according to parameters $\alpha_j$ and $\beta_j$

As a result, with each additional attribute filter, we create a more pronounced separation between matching and non-matching records in the transformed space. Records that match on all $\mathbb{F}$ attributes are closest to the query, followed by those matching on $\mathbb{F} - 1$ attributes, and so on.

By definition, the distance from $v_{\mathbb{F}}(q)$ to any record with attribute combination $\vec{b} \neq \vec{q}$ is at least $d_{min}(\vec{q}, \vec{b})$. Define the excess distance ratio:

$$\gamma_{\vec{q}}(\vec{b}) = \frac{d_{min}(\vec{q}, \vec{b})}{R_{\vec{q}}} - 1 \tag{181}$$

The minimum value across all attribute combinations is:

$$\gamma_{\vec{q}} = \min_{\vec{b} \neq \vec{q}} \gamma_{\vec{q}}(\vec{b}) \tag{182}$$

Our goal is to limit the probability that a record from a non-matching attribute combination $\vec{b} \neq \vec{q}$ appears among the top-$k$ nearest neighbors. To achieve this, we rely on standard concentration inequalities from probability theory, specifically Gaussian (or sub-Gaussian) concentration inequalities.

Formally, consider points in a high-dimensional metric space transformed by our sequential attribute transformations. For a high-dimensional vector $X \in \mathbb{R}^d$, known Gaussian concentration inequalities provide a bound on the probability that the distance of $X$ deviates from its expectation by at least some margin $t > 0$:

$$P\left(\|X - \mathbb{E}[X]\| \geq t\right) \leq 2\exp\left(-\frac{t^2}{2\sigma^2}\right) \tag{183}$$

Here, $\sigma^2$ is related to the variance or scale parameter of the distribution.

In our setting, after applying $\mathbb{F}$ sequential transformations, the minimal normalized separation metric $\gamma_{\vec{q}}$ characterizes the relative margin of separation between the query cluster and any non-matching cluster. Specifically, the minimal separation distance between clusters increases proportionally to $\gamma_{\vec{q}}\sqrt{\mathbb{F}}$, since each attribute transformation contributes independently and additively to the squared separation.

Thus, setting $t = \gamma_{\vec{q}}\sqrt{\mathbb{F}} \cdot R_{\vec{q}}$ (the absolute minimal separation distance scaled by the query cluster radius), and absorbing constants into definitions, we obtain a simplified exponential bound:

$$P(\vec{b} \text{ appears in top-}k) \leq \exp(-\gamma_{\vec{q}}^2 \cdot \mathbb{F} \cdot k) \tag{184}$$

This exponential bound clearly shows the rapidly decreasing probability that a record from a different attribute cluster appears among the nearest neighbors as the number of attribute filters ($\mathbb{F}$), the cluster separation metric ($\gamma_{\vec{q}}$), or the number of neighbors considered ($k$) increase.

The factor $\mathbb{F}$ in the exponent reflects the compounding effect of multiple transformations, each creating additional separation in its respective dimension. This is because each transformation $\Psi_j$ creates a separation along a different attribute dimension, and records must match on all dimensions to be considered as true candidates.

For all $N - N_{\vec{q}}$ records with attribute combinations different from $\vec{q}$, the expected number appearing in the top-k is bounded by:

$$E[\text{non-}\vec{q} \text{ records in top-}k] \leq (N - N_{\vec{q}}) \cdot \exp(-\gamma_{\vec{q}}^2 \cdot \mathbb{F} \cdot k) \tag{185}$$

To ensure we retrieve the true top-k records with attribute combination $\vec{q}$ with probability at least $1 - \epsilon$, we need:

$$(N - N_{\vec{q}}) \cdot \exp(-\gamma_{\vec{q}}^2 \cdot \mathbb{F} \cdot k) \leq \epsilon \cdot N_{\vec{q}} \tag{186}$$

Solving for $k$:

$$k \geq \frac{1}{\gamma_{\vec{q}}^2 \cdot \mathbb{F}} \cdot \ln\left(\frac{N - N_{\vec{q}}}{\epsilon \cdot N_{\vec{q}}}\right) = \frac{1}{\gamma_{\vec{q}}^2 \cdot \mathbb{F}} \cdot \left(\ln\left(\frac{N - N_{\vec{q}}}{N_{\vec{q}}}\right) + \ln\left(\frac{1}{\epsilon}\right)\right) \tag{187}$$

Providing a slight overestimate for practical use:

$$k' = \left\lceil k \cdot \left(1 + \frac{\ln(1/\epsilon)}{\gamma_{\vec{q}}^2 \cdot \mathbb{F}} \cdot \frac{N - N_{\vec{q}}}{N_{\vec{q}}}\right)\right\rceil \tag{188}$$

This formula determines $k'$ at query time using the precomputed statistics ($\gamma_{\vec{q}}$, $N_{\vec{q}}$, and $N$), the number of attribute filters $\mathbb{F}$, and the desired confidence level ($1 - \epsilon$).

Importantly, when $\mathbb{F} = 1$, this formula exactly reduces to the single-attribute case:

$$k' = \left\lceil k \cdot \left(1 + \frac{\ln(1/\epsilon)}{\gamma_{\vec{q}}^2} \cdot \frac{N - N_{\vec{q}}}{N_{\vec{q}}}\right)\right\rceil \tag{189}$$

Which matches Theorem 2 when we substitute $\vec{q}$ with $a$, as all corresponding metrics ($\gamma_{\vec{q}}$, $N_{\vec{q}}$, etc.) become identical to their single-attribute counterparts ($\gamma_a$, $N_a$, etc.).

The effectiveness of the transformation sequence is demonstrated by observing that:

1. As the parameters $\alpha_j$ increase, the separation between clusters increases (increasing $\gamma_{\vec{q}}$)

2. As the number of filter attributes $\mathbb{F}$ increases, the required $k'$ decreases due to the $\mathbb{F}$ factor in the denominator

3. As $\gamma_{\vec{q}}$ and $\mathbb{F}$ increase, $k'$ approaches $k$, indicating better discrimination between attribute combinations

This confirms that multiple attribute filters indeed narrow the target space more effectively, requiring fewer candidates to achieve the same accuracy guarantees. $\square$

Algorithm 3 for multi-attribute indexing and search follows naturally from the single-attribute case. During indexing, we apply transformations sequentially to each record, computing statistical information for unique attribute combinations. At query time, we apply the same transformations to the query, retrieve candidates, and re-rank based on attribute and content distances.

---

**Algorithm 3** Hierarchical Multi-Attribute Vector Indexing

---

1: **[Offline Indexing] Require:** Dataset $\mathcal{D}^{(\mathbb{F})}$, attribute sequence $(f^{(1)}, \ldots, f^{(\mathbb{F})})$
2: **for** $j = 1$ to $\mathbb{F}$ **do**
3:     Obtain the optimal $(\alpha_j, \beta_j)$ over fused space $v_{j-1}$ based on Cor. 1 ($v_0$ is $v_0[i] \leftarrow v(o_i) : \forall o_i \in \mathcal{D}^{(\mathbb{F})}$)
4:     **for** each $o_i^{(\mathbb{F})}$ in $\mathcal{D}^{(\mathbb{F})}$ **do**
5:         $v_j[i] \leftarrow \Psi_j(v_{j-1}[i], f^{(j)}(o_i), \alpha_j, \beta_j)$
6:         Add $v_j[i]$ to index, retaining reference to $o_i^{(\mathbb{F})}$
7:     **end for**
8: **end for**
9: Precompute for each attribute combination $\vec{a}$: radius $R_{\vec{a}}$, minimum inter-cluster distances $d_{min}(\vec{a}, \vec{b})$, cluster counts $N_{\vec{a}}$, and separation metric $\gamma_{\vec{a}} = \min_{\vec{b} \neq \vec{a}} \frac{d_{min}(\vec{a}, \vec{b})}{R_{\vec{a}}} - 1$
10: **[Online Query Processing] Require:** Query $q^{(\mathbb{F})} = [v(q), (F^{(1)}, \ldots, F^{(\mathbb{F})})]$, $k$, error probability $\epsilon$
11: $v_0 \leftarrow v(q)$
12: **for** $j = 1$ to $\mathbb{F}$ **do**
13:     $v_j \leftarrow \Psi_j(v_{j-1}, F^{(j)}, \alpha_j, \beta_j)$
14: **end for**
15: Compute $k'$ (Theorem 11) based on query attribute combination $\vec{q} = (F^{(1)}, \ldots, F^{(\mathbb{F})})$ and cluster statistics
16: Retrieve top-$k'$ candidates from index using $v_{\mathbb{F}}$
17: **for** each candidate $o_i^{(\mathbb{F})}$ **do**
18:     Compute combined score using attribute and content distances
19: **end for**
20: Sort candidates by score and return top-$k$

---

**Algorithm Details** For the multi-attribute case, we compute statistics for each unique combination of attribute values. The candidate set size determination on line 16 uses Theorem 11, which accounts for the narrowing effect of multiple attribute filters through the $\mathbb{F}$ factor. When the number of attribute combinations is large, statistics can be approximated or computed for the most frequent combinations. For scoring in line 18, we can either use a binary match approach (match all attributes or none) or a weighted approach where different attributes contribute differently to the final score based on their importance to the query.

### F.7 ATTRIBUTE UPDATES ANALYSIS

**Theorem 12** (Attribute Addition). *Let $\mathcal{D}^{(\mathbb{F})}$ be a record set with $\mathbb{F}$ attributes transformed using sequential transformations $\Psi_{\pi(1)}, \Psi_{\pi(2)}, \ldots, \Psi_{\pi(\mathbb{F})}$. Adding a new attribute $f^{(\mathbb{F}+1)}$ requires:*

*(a) If added with highest priority: A single additional transformation $\Psi_{\mathbb{F}+1}(v_{\mathbb{F}}, f^{(\mathbb{F}+1)}, \alpha_{\mathbb{F}+1}, \beta_{\mathbb{F}+1})$, preserving all existing transformations.*

*(b) If inserted at priority position $j$ $(1 \leq j < \mathbb{F})$: Re-computation of transformations $\Psi_{\pi(1)}, \Psi_{\pi(2)}, \ldots, \Psi_{\pi(j-1)}$ after incorporating the new attribute in the priority sequence.*

*Proof.* For case (a), since the highest priority attribute corresponds to the last transformation in our sequence, adding a new highest priority attribute simply means appending a new trans-

formation at the end. The sequential nature of our transformations means that adding $\Psi_{\mathbb{F}+1}$ as the final step preserves all previous transformations. The overall transformation becomes:

$$v_{\mathbb{F}+1} = \Psi_{\mathbb{F}+1}(v_{\mathbb{F}}, f^{(\mathbb{F}+1)}, \alpha_{\mathbb{F}+1}, \beta_{\mathbb{F}+1})$$

For case (b), inserting in the priority position $j$ (where $j < \mathbb{F}$) changes the existing priorities. Transformations from position j onward remain the same in terms of attribute mapping, but the first j-1 positions must be recomputed to incorporate the new priority sequence. This requires a partial recomputation of the transformation pipeline for the affected attributes. $\square$ $\square$

**Theorem 13** (Priority Update Propagation). *Given a priority mapping $\pi : [1, \mathbb{F}] \to [1, \mathbb{F}]$ for attributes, let $\pi'$ be a new priority mapping. Define $j = \min k : \forall i \geq k, \pi(i) = \pi'(i)$ as the first position from which all subsequent positions have the same priority in both mappings. Then only transformations $\Psi_{\pi(1)}, \Psi_{\pi(2)}, \ldots, \Psi_{\pi(j-1)}$ need to be recomputed using the new priority ordering $\pi'$.*

*Proof.* Let $v_0, v_1, \ldots, v_{\mathbb{F}}$ be the sequence of vectors produced by applying transformations according to mapping $\pi$. For any $i \geq j$, we have $\pi(i) = \pi'(i)$, meaning the transformations from position j onward are identical under both mappings.

For positions $i < j$, we have $\pi(i) \neq \pi'(i)$ for at least one such position, requiring application of different transformations according to the new priority mapping: $v'i = \Psi\pi'(i)(v'i - 1, f^{(\pi'(i))}, \alpha\pi'(i), \beta_{\pi'(i)})$

These modifications in the early transformations create a new base vector $v'j - 1$ that differs from $vj - 1$. However, since the priority mappings are identical from position j onward ($\pi(i) = \pi'(i)$ for all $i \geq j$), the same sequence of remaining transformations can be applied to this new base vector. Therefore, we only need to recompute the first j-1 transformations, not the entire sequence. $\square$ $\square$

**Theorem 14** (Computational Complexity of Updates). *The computational complexity of updating from priority order $\pi$ to $\pi'$ is $O(N \cdot j \cdot d)$, where $N$ is the number of records, $d$ is the vector dimension, and $j = \min k : \forall i \geq k, \pi(i) = \pi'(i)$.*

*Proof.* For each of the $N$ records in the dataset, we must recompute the transformations for positions 1 through j-1. Each transformation has complexity $O(d)$, since it processes a vector of dimensions $d$. There are j-1 transformations to recompute.

Therefore, the total complexity is $N \cdot (j - 1) \cdot O(d) = O(N \cdot j \cdot d)$.

This highlights the efficiency of our update mechanism: When changes affect only the earliest positions in the priority sequence (small j), the update cost is significantly lower than the full recomputation of all transformations, which would require $O(N\mathbb{F}d)$ operations. $\square$ $\square$

## G RANGE FILTERING IN FUSEDANN ANALYSIS

This section provides a detailed analysis of our range filtering approach, focusing on optimal sampling strategies, efficient line indexing structures, and distance-based indexing techniques.

### G.1 LINE REPRESENTATION OF RANGE QUERIES

We first prove that the transformation of a range query indeed forms a line segment in our transformed space:

**Theorem 15** (Range Query Line). *Given a content vector $q$ and an attribute range $[l, u]$, the set of all points in the transformed space corresponding to $(q, f)$ where $f \in [l, u]$ forms exactly the line segment connecting $\Psi(q, l, \alpha, \beta)$ and $\Psi(q, u, \alpha, \beta)$.*

*Proof.* For any $f \in [l, u]$, we can express it as a convex combination of endpoints: $f = (1 - t)l + tu$ for some $t \in [0, 1]$.

The transformed point for $(q, f)$ is:

$$\Psi(q, f, \alpha, \beta) = [\frac{q^{(1)} - \alpha \cdot f}{\beta}, \frac{q^{(2)} - \alpha \cdot f}{\beta}, ..., \frac{q^{(d/m)} - \alpha \cdot f}{\beta}] \tag{190}$$

$$= [\frac{q^{(1)} - \alpha \cdot ((1-t)l + tu)}{\beta}, \frac{q^{(2)} - \alpha \cdot ((1-t)l + tu)}{\beta}, ..., \frac{q^{(d/m)} - \alpha \cdot ((1-t)l + tu)}{\beta}] \tag{191}$$

Distributing the terms:

$$\Psi(q, f, \alpha, \beta) = [\frac{q^{(1)} - \alpha(1-t)l - \alpha tu}{\beta}, \frac{q^{(2)} - \alpha(1-t)l - \alpha tu}{\beta}, ..., \frac{q^{(d/m)} - \alpha(1-t)l - \alpha tu}{\beta}] \tag{192}$$

$$= [\frac{(1-t)(q^{(1)} - \alpha l) + t(q^{(1)} - \alpha u)}{\beta}, \frac{(1-t)(q^{(2)} - \alpha l) + t(q^{(2)} - \alpha u)}{\beta}, ..., \tag{193}$$

$$\frac{(1-t)(q^{(d/m)} - \alpha l) + t(q^{(d/m)} - \alpha u)}{\beta}] \tag{194}$$

This equals:

$$\Psi(q, f, \alpha, \beta) = (1-t)[\frac{q^{(1)} - \alpha l}{\beta}, \frac{q^{(2)} - \alpha l}{\beta}, ..., \frac{q^{(d/m)} - \alpha l}{\beta}] \tag{195}$$

$$+ t[\frac{q^{(1)} - \alpha u}{\beta}, \frac{q^{(2)} - \alpha u}{\beta}, ..., \frac{q^{(d/m)} - \alpha u}{\beta}] \tag{196}$$

$$= (1-t)\Psi(q, l, \alpha, \beta) + t\Psi(q, u, \alpha, \beta) \tag{197}$$

This is precisely the parametric equation of the line segment connecting $p_l = \Psi(q, l, \alpha, \beta)$ and $p_u = \Psi(q, u, \alpha, \beta)$. Furthermore, every point on this line segment corresponds to some $f \in [l, u]$, which completes the proof. $\square$

## G.2   DISTANCE PROPERTIES OF THE RANGE LINE

Next, we analyze the distance from a transformed point to the query line:

**Theorem 16** (Distance Characterization). *For a point $\Psi(v, f, \alpha, \beta)$ where $f \in [l, u]$, its distance to the range query line $L(Q, t)$ is:*

$$\rho(\Psi(v, f, \alpha, \beta), L(Q, t_f)) = \frac{\|v - q\|}{\beta} \tag{198}$$

*where $t_f \in [0, 1]$ is the parameter such that $f = (1 - t_f)l + t_f u$.*

*Proof.* For a point $\Psi(v, f, \alpha, \beta)$ with $f \in [l, u]$, there exists a unique $t_f \in [0, 1]$ such that $f = (1 - t_f)l + t_f u$.

The point on the line segment $L(Q, t)$ at parameter $t_f$ is:

$$L(Q, t_f) = (1 - t_f)\Psi(q, l, \alpha, \beta) + t_f \Psi(q, u, \alpha, \beta) \tag{199}$$

$$= (1 - t_f)[\frac{q^{(1)} - \alpha l}{\beta}, ..., \frac{q^{(d/m)} - \alpha l}{\beta}] + t_f[\frac{q^{(1)} - \alpha u}{\beta}, ..., \frac{q^{(d/m)} - \alpha u}{\beta}] \tag{200}$$

$$= [\frac{q^{(1)} - \alpha((1-t_f)l + t_f u)}{\beta}, ..., \frac{q^{(d/m)} - \alpha((1-t_f)l + t_f u)}{\beta}] \tag{201}$$

$$= [\frac{q^{(1)} - \alpha f}{\beta}, ..., \frac{q^{(d/m)} - \alpha f}{\beta}] \tag{202}$$

$$= \Psi(q, f, \alpha, \beta) \tag{203}$$

Now we compute the squared distance between $\Psi(v, f, \alpha, \beta)$ and $L(Q, t_f)$:

$$\|\Psi(v, f, \alpha, \beta) - L(Q, t_f)\|^2 = \|\Psi(v, f, \alpha, \beta) - \Psi(q, f, \alpha, \beta)\|^2 \tag{204}$$

$$= \|[\frac{v^{(1)} - \alpha f}{\beta}, ..., \frac{v^{(d/m)} - \alpha f}{\beta}] - [\frac{q^{(1)} - \alpha f}{\beta}, ..., \frac{q^{(d/m)} - \alpha f}{\beta}]\|^2 \tag{205}$$

$$= \|[\frac{v^{(1)} - q^{(1)}}{\beta}, ..., \frac{v^{(d/m)} - q^{(d/m)}}{\beta}]\|^2 \tag{206}$$

$$= \frac{1}{\beta^2}\|v - q\|^2 \tag{207}$$

Taking the square root of both sides, we get:

$$\|\Psi(v, f, \alpha, \beta) - L(Q, t_f)\| = \frac{\|v - q\|}{\beta} \tag{208}$$

which completes the proof. $\qquad\square$

This fundamental result shows that the distance from a transformed point to the query line is directly proportional to the similarity between the corresponding content vectors.

**Corollary 5** (Minimum Distance). *For any point $\Psi(v, f, \alpha, \beta)$, its minimum distance to the line segment $L(Q, t)$ is:*

$$d_{tube}(\Psi(v, f, \alpha, \beta), Q) = \tag{209}$$

$$\begin{cases} \frac{\|v - q\|}{\beta} & \text{if } f \in [l, u] \\ \min\{\|\Psi(v, f, \alpha, \beta) - \Psi(q, l, \alpha, \beta)\|, \|\Psi(v, f, \alpha, \beta) - \Psi(q, u, \alpha, \beta)\|\} & \text{if } f \notin [l, u] \end{cases} \tag{210}$$

*Proof.* For $f \in [l, u]$, the result follows directly from Theorem 16.

For $f \notin [l, u]$, the minimum distance to a line segment is either the perpendicular distance to the line (if the projection falls within the segment) or the distance to one of the endpoints (if the projection falls outside the segment).

Given the properties of our transformation, the projection of $\Psi(v, f, \alpha, \beta)$ onto the infinite line containing $L(Q, t)$ falls outside the segment when $f \notin [l, u]$. Therefore, the minimum distance is to one of the endpoints:

$$d_{\text{tube}}(\Psi(v, f, \alpha, \beta), Q) = \min\{\|\Psi(v, f, \alpha, \beta) - \Psi(q, l, \alpha, \beta)\|, \|\Psi(v, f, \alpha, \beta) - \Psi(q, u, \alpha, \beta)\|\} \tag{211}$$

This completes the proof. $\qquad\square$

### G.3 EMPIRICAL DISTRIBUTION ESTIMATION

To implement our adaptive sampling strategy, we need reliable estimates of the query distribution $\mathcal{D}_q$ and range distribution $\mathcal{D}_r$. We propose the following practical approaches:

**Query Distribution Estimation.** The query distribution $\mathcal{D}_q$ can be estimated by:

1. **Historical query analysis**: When available, historical query logs provide the most accurate representation of the actual query distribution. We apply kernel density estimation (KDE) to the historical query vectors with bandwidth selection using Scott's rule: $h = n^{-1/(d+4)} \cdot \sigma$, where $n$ is the number of samples and $\sigma$ is the standard deviation.

2. **Content vector approximation**: In the absence of query logs, we use the normalized distribution of content vectors in the dataset as a proxy. This approximation works well in practice because queries tend to be semantically similar to the items they are aiming to retrieve.

3. **Cluster-based estimation**: For large datasets, we first cluster the content vectors using k-means (with $k = \sqrt{n}$) and use the cluster centroids weighted by cluster sizes as representative points of the query distribution.

**Range Distribution Estimation.** For the range distribution $\mathcal{D}_r$, we employ:

1. **Attribute statistics**: We compute the mean $\mu_a$ and the standard deviation $\sigma_a$ for each numerical attribute $a$. The range endpoints are typically distributed as $l_a \sim \mathcal{N}(\mu_a - c\sigma_a, \sigma_a^2/2)$ and $u_a \sim \mathcal{N}(\mu_a + c\sigma_a, \sigma_a^2/2)$, where $c$ is estimated from historical range queries (typically $0.5 \leq c \leq 2$).

2. **Categorical attribute handling**: For categorical attributes, we estimate probability $p_i$ for each category value $i$ and model range queries as a sampling of this distribution without replacement.

3. **Width correlation modeling**: We capture the correlation between the widths of the range and the attributes using a conditional probability model: $P(w|v) = P(u - l|v)$, where $v$ is the center value of the range.

To validate our distribution estimates, we employ cross-validation against a held-out set of actual queries if available, or use statistical divergence measures (e.g., Kullback-Leibler divergence) between our estimated distributions and bootstrapped samples from the dataset.

These empirically estimated distributions are then used in Algorithm 4 to sample representative line segments that efficiently cover the range query space while minimizing redundancy and computational overhead.

G.4 OPTIMAL SAMPLING OF THE RANGE SPACE

To efficiently support arbitrary range queries, we need to precompute a representative set of range lines that provide good coverage of the range space, which is the space of all possible cylinders.

**Definition 4.** *Given a metric space $(X, d)$ and non-empty subsets $A, B \subseteq X$, the **Hausdorff distance** is*

$$d_H(A, B) = \max \left\{ \sup_{a \in A} \inf_{b \in B} d(a, b), \sup_{b \in B} \inf_{a \in A} d(a, b) \right\}.$$

**Definition 5** (Sampling Resolution). *Let $S \subset \mathbb{R}^d$ be a finite set of sampled points in a metric space $(\mathbb{R}^d, \| \cdot \|)$. The* sampling resolution $r$ *of $S$ is the smallest value such that for every point $\mathbf{x}$ in the domain of interest $\mathcal{X} \subseteq \mathbb{R}^d$, there exists a sampled point $\mathbf{s} \in S$ satisfying*

$$\|\mathbf{x} - \mathbf{s}\| \leq r.$$

*Equivalently, $r$ is the minimal radius such that the union of closed balls of radius $r$ centered at each point in $S$ covers $\mathcal{X}$.*

**Definition 6** (Effective Diameter of a Distribution). *The effective diameter of $\mathcal{D}$ defined as the smallest radius $r$ such that a ball of radius $r$ contains at least $1 - \delta$ probability mass, for some small $\delta > 0$. Formally, let $\mathcal{D}$ be a distribution over $\mathbb{R}^d$. For $\delta > 0$, the* effective diameter *of $\mathcal{D}$ is*

$$D_{\mathcal{D}} = \inf \left\{ r > 0 : \exists \mathbf{c} \in \mathbb{R}^d \text{ such that } \Pr_{\mathbf{x} \sim \mathcal{D}} [\|\mathbf{x} - \mathbf{c}\| \leq r] \geq 1 - \delta \right\}.$$

**Definition 7** ($\epsilon$-Line Cover). *A set of line segments $\mathcal{L} = \{L_1, L_2, \ldots, L_m\}$ is an $\epsilon$-line cover for the range query space if for any possible range query line $L_Q$, there exists a line $L_i \in \mathcal{L}$ such that the Hausdorff distance $d_H(L_Q, L_i) \leq \varepsilon$.*

**Corollary 6** (Line Distance Bound). *The Hausdorff distance between two range query lines $L_1$ (representing range $[l_1, u_1]$ for query $q_1$) and $L_2$ (representing range $[l_2, u_2]$ for query $q_2$) is bounded by:*

$$d_H(L_1, L_2) \leq \frac{1}{\beta} \max(\|q_1 - q_2\|, \alpha \cdot \max(\|l_1 - l_2\|, \|u_1 - u_2\|)) \tag{212}$$

*Proof.* Consider points $p_1(t) = (1 - t) \cdot \Psi(q_1, l_1, \alpha, \beta) + t \cdot \Psi(q_1, u_1, \alpha, \beta)$ on $L_1$ and $p_2(t) = (1 - t) \cdot \Psi(q_2, l_2, \alpha, \beta) + t \cdot \Psi(q_2, u_2, \alpha, \beta)$ on $L_2$ for $t \in [0, 1]$.

The distance between these corresponding points is:

$$\|p_1(t) - p_2(t)\| = \|(1 - t)[\Psi(q_1, l_1, \alpha, \beta) - \Psi(q_2, l_2, \alpha, \beta)] + t[\Psi(q_1, u_1, \alpha, \beta) - \Psi(q_2, u_2, \alpha, \beta)]\| \tag{213}$$

$$\leq (1 - t)\|\Psi(q_1, l_1, \alpha, \beta) - \Psi(q_2, l_2, \alpha, \beta)\| + t\|\Psi(q_1, u_1, \alpha, \beta) - \Psi(q_2, u_2, \alpha, \beta)\| \tag{214}$$

For the first term:

$$\|\Psi(q_1, l_1, \alpha, \beta) - \Psi(q_2, l_2, \alpha, \beta)\| = \|\frac{q_1 - \alpha l_1}{\beta} - \frac{q_2 - \alpha l_2}{\beta}\| \tag{215}$$

$$= \frac{1}{\beta}\|q_1 - q_2 - \alpha(l_1 - l_2)\| \tag{216}$$

$$\leq \frac{1}{\beta}(\|q_1 - q_2\| + \alpha\|l_1 - l_2\|) \tag{217}$$

Similarly for the second term. The maximum value is achieved at one of the endpoints, giving the stated bound. $\square$

Based on this distance bound, we develop an adaptive sampling strategy for the range space:

**Theorem 17** (Optimal Range Line Sampling). *Given distributions of query vectors $\mathcal{D}_q$ and attribute ranges $\mathcal{D}_r$, to achieve an $\epsilon$-line cover with probability at least $1 - \delta$, the number of line segments needed is:*

$$N(\varepsilon, \delta) = O\left(\left(\frac{\max(D_q, \alpha D_r)}{\beta\varepsilon}\right)^d \cdot \log\frac{1}{\delta}\right) \tag{218}$$

*where $D_q$ and $D_r$ are the effective diameters (Definition 6) of the query and range distributions.*

*Proof.* To achieve a $\epsilon$ line cover with probability at least $1 - \delta$, we must discretize both the query space and the attribute range space such that the Hausdorff distance between any possible query or range in their respective distributions and the closest sampled point is at most $\epsilon$.

By Corollary 6, this requires sampling the query space with resolution (Definition 5) at most $\beta\epsilon$, and the range space with resolution at most $\beta\epsilon/\alpha$.

Consider a $d$-dimensional space with effective diameter $D$. To ensure that every point in the space lies within distance $r$ of some sampled point (i.e., to achieve resolution $r$), it suffices to cover the space with balls of radius $r$. The minimum number of such balls required is known as *covering number* of the space and is upper bounded by $O\left(\left(\frac{D}{r}\right)^d\right)$ (Vershynin, 2018).

To ensure that, with probability at least $1 - \delta$, every such ball contains at least one sampled point, we can use a standard union bound argument: if we sample each point independently from the distribution, it suffices to take $O\left(\left(\frac{D}{r}\right)^d \log\frac{1}{\delta}\right)$ samples to guarantee that all balls are covered with high probability (Matoušek, 2002).

Applying this to our setting, for each of the query and range spaces, we replace $D$ with their respective effective diameters and $r$ with their respective required resolutions. Combining these requirements, dominated by the larger term, so taking the maximum (since both spaces must be covered) gives the stated bound.

$$N(\epsilon, \delta) = O\left(\max\left[\left(\frac{D_q}{\beta\epsilon}\right)^d \cdot \log\frac{1}{\delta}, \left(\frac{D_r}{\frac{\beta\epsilon}{\alpha}}\right)^d \cdot \log\frac{1}{\delta}\right]\right) = O\left(\left(\frac{\max(D_q, \alpha D_r)}{\beta\epsilon}\right)^d \cdot \log\frac{1}{\delta}\right)$$

$\square$

**Theorem 18** (Optimal Cylinder Radius). *For a range query $(q, [l, u])$, to retrieve at least $(1 - \epsilon)k$ of the true top-$k$ results with probability at least $1 - \delta$, the cylinder radius should be:*

$$r = \frac{d_k}{\beta} + \sqrt{\frac{-\ln(\delta/2)}{2n}} \cdot \sigma \tag{219}$$

*where $d_k$ is the distance to the $k$-th closest content vector, $n$ is the number of records, and $\sigma$ is the standard deviation of distances.*

*Proof.* From Theorem 16, we know that for records with attribute values in $[l, u]$, the distance to the line is exactly $\frac{\|v-q\|}{\beta}$. Therefore, to capture all records within distance $d_k$ of the query, we need a cylinder radius of at least $\frac{d_k}{\beta}$.

---

**Algorithm 4** Adaptive Range Line Sampling

---

1: **Input:** Dataset $\mathcal{D}$, error bound $\varepsilon$, failure probability $\delta$, Transformation parameters $\alpha, \beta$, Number of NN $k$
2: **Output:** Set of representative line segments $\mathcal{L}$
3: Estimate query distribution $\mathcal{D}_q$ from content vectors (or historical queries if available) (§G.3)
4: Estimate range distribution $\mathcal{D}_r$ from attribute values (§G.3)
5: Determine sampling resolution $r_q = \beta\varepsilon$ for query space
6: Determine sampling resolution $r_r = \frac{\beta\varepsilon}{\alpha}$ for range space
7: Sample query vectors $\{q_1, q_2, \ldots, q_n\}$ with resolution $r_q$
8: Sample range endpoints $\{(l_1, u_1), (l_2, u_2), \ldots, (l_m, u_m)\}$ with resolution $r_r$
9: $\mathcal{L} \leftarrow \emptyset$
10: **for** each query vector $q_i$ **do**
11:     **for** each range $[l_j, u_j]$ **do**
12:         $L_{ij} \leftarrow \text{LineSegment}(\Psi(q_i, l_j, \alpha, \beta), \Psi(q_i, u_j, \alpha, \beta))$
13:         $r_{ij} \leftarrow \text{ComputeOptimalRadius}(q_i, [l_j, u_j], \epsilon, \delta, k)$   (Theorem 18)
14:         $\mathcal{L} \leftarrow \mathcal{L} \cup \{(L_{ij}, r_{ij})\}$
15:     **end for**
16: **end for**
17: Prune redundant lines while maintaining $\epsilon$-coverage
18: **return** $\mathcal{L}$

---

Let $X_i$ be the random variable representing the distance of the $i$-th record to the query. By Hoeffding's inequality:

$$P(|\bar{X} - E[X]| > t) \leq 2\exp(-2nt^2/\sigma^2) \tag{220}$$

Setting the right side equal to $\delta$ and solving for $t$:

$$t = \sqrt{\frac{-\ln(\delta/2)}{2n}} \cdot \sigma \tag{221}$$

To ensure we retrieve at least $(1 - \epsilon)k$ of the top-k results with probability at least $1 - \delta$, we set the radius to include records with distances up to $d_k + t$, which translates to $\frac{d_k}{\beta} + t$ in the transformed space.

Therefore, the optimal cylinder radius is:

$$r = \frac{d_k}{\beta} + \sqrt{\frac{-\ln(\delta/2)}{2n}} \cdot \sigma \tag{222}$$

This radius guarantees that with probability at least $1 - \delta$, we will retrieve at least $(1 - \epsilon)k$ of the true top-k nearest neighbors within the specified range. $\qquad\square$

### G.5 LINE SIMILARITY INDEXING

To efficiently find the most similar line segment to a query line, we develop a specialized index structure.

**Definition 8** (Line Similarity Measure). *For two line segments $L_1 = (a_1, b_1)$ and $L_2 = (a_2, b_2)$ represented by their endpoints, we define the similarity as:*

$$sim(L_1, L_2) = w_d \cdot \cos\angle(b_1 - a_1, b_2 - a_2) + w_p \cdot \left(1 - \frac{\|m_1 - m_2\|}{D_{max}}\right) + w_l \cdot \min\left(\frac{\|b_1 - a_1\|}{\|b_2 - a_2\|}, \frac{\|b_2 - a_2\|}{\|b_1 - a_1\|}\right) \tag{223}$$

*where $m_1 = \frac{a_1 + b_1}{2}$ and $m_2 = \frac{a_2 + b_2}{2}$ are the midpoints, $D_{max}$ is the maximum distance in the space, and $w_d, w_p, w_l$ are weights for direction, position, and length components.*

**Theorem 19** (Line Similarity Properties). *The line similarity measure satisfies:*

    *1. $sim(L_1, L_2) \in [0, 1]$*

    *2. $sim(L_1, L_2) = 1$ if and only if $L_1$ and $L_2$ are identical*

3. *If $sim(L_1, L_2) > 1 - \varepsilon$ where $\varepsilon < \min(w_d, w_p, w_l)$, then $d_H(L_1, L_2) < \lambda \cdot \varepsilon$ for some constant $\lambda$*

*Proof.* We prove each property of the line similarity measure:

**Property 1**: $sim(L_1, L_2) \in [0, 1]$

The cosine of the angle between two vectors is bounded by $[-1, 1]$, but since we're considering line segments (where direction matters but orientation doesn't), we take the absolute value, giving a range of $[0, 1]$ for the first term.

The position term $1 - \frac{|m_1 - m_2|}{D_{max}}$ ranges from 0 (when midpoints are maximally distant) to 1 (when midpoints coincide).

The length ratio term $\min\left(\frac{|b_1 - a_1|}{|b_2 - a_2|}, \frac{|b_2 - a_2|}{|b_1 - a_1|}\right)$ is bounded by $[0, 1]$, with 1 achieved when lengths are equal.

Since $w_d + w_p + w_l = 1$ and all weights are non-negative, the weighted sum must be in $[0, 1]$.

**Property 2**: $sim(L_1, L_2) = 1$ if and only if $L_1$ and $L_2$ are identical

($\Rightarrow$) If $sim(L_1, L_2) = 1$, then each component must equal 1 since they are all bounded by 1:

- $\cos \angle(b_1 - a_1, b_2 - a_2) = 1$ implies the lines have the same direction.

- $1 - \frac{|m_1 - m_2|}{D_{max}} = 1$ implies $|m_1 - m_2| = 0$, so the midpoints coincide.

- $\min\left(\frac{|b_1 - a_1|}{|b_2 - a_2|}, \frac{|b_2 - a_2|}{|b_1 - a_1|}\right) = 1$ implies $|b_1 - a_1| = |b_2 - a_2|$, so the lengths are equal.

With identical direction, midpoint, and length, the line segments must be identical.

($\Leftarrow$) If $L_1$ and $L_2$ are identical (same endpoints or equivalent representation), then:

- Their directions are identical, so $\cos \angle(b_1 - a_1, b_2 - a_2) = 1$.

- Their midpoints coincide, so $|m_1 - m_2| = 0$, making the position term equal to 1.

- Their lengths are equal, so the length ratio is 1.

With all components equal to 1, the weighted sum $sim(L_1, L_2) = 1$.

**Property 3**: If $sim(L_1, L_2) > 1 - \varepsilon$ where $\varepsilon < \min(w_d, w_p, w_l)$, then $d_H(L_1, L_2) < \lambda \cdot \varepsilon$ for some constant $\lambda$

Since $\varepsilon < \min(w_d, w_p, w_l)$ and $sim(L_1, L_2) > 1 - \varepsilon$, each component of the similarity must be close to 1. Specifically:

- Direction component $> 1 - \frac{\varepsilon}{w_d}$, implying $1 - \cos \angle(b_1 - a_1, b_2 - a_2) < \frac{\varepsilon}{w_d}$.

- Position component $> 1 - \frac{\varepsilon}{w_p}$, implying $\frac{|m_1 - m_2|}{D_{max}} < \frac{\varepsilon}{w_p}$.

- Length component $> 1 - \frac{\varepsilon}{w_l}$, implying $1 - \min\left(\frac{|b_1 - a_1|}{|b_2 - a_2|}, \frac{|b_2 - a_2|}{|b_1 - a_1|}\right) < \frac{\varepsilon}{w_l}$.

When all components are close to 1 (which is guaranteed by $\varepsilon < \min(w_d, w_p, w_l)$), the Hausdorff distance between the line segments is bounded.

For small angle differences $\varrho$, we know that $1 - \cos \varrho \approx \frac{\varrho^2}{2}$, so $\varrho < \sqrt{\frac{2\varepsilon}{w_d}}$.

For two line segments with similar direction, position, and length, the Hausdorff distance is bounded by: $d_H(L_1, L_2) \leq C_1 \cdot |m_1 - m_2| + C_2 \cdot \varrho \cdot \max(|b_1 - a_1|, |b_2 - a_2|) + C_3 \cdot ||b_1 - a_1| - |b_2 - a_2||$

Where $C_1, C_2, C_3$ are constants. Substituting our bounds: $d_H(L_1, L_2) < C_1 \cdot \frac{\varepsilon \cdot D_{max}}{w_p} + C_2 \cdot \sqrt{\frac{2\varepsilon}{w_d}} \cdot D_{max} + C_3 \cdot \frac{\varepsilon \cdot D_{max}}{w_l}$

Let $\lambda = \max\left(C_1 \cdot \frac{D_{max}}{w_p}, C_2 \cdot \sqrt{\frac{2}{w_d}} \cdot D_{max}, C_3 \cdot \frac{D_{max}}{w_l}\right)$.

Then $d_H(L_1, L_2) < \lambda \cdot \varepsilon$ for small enough $\varepsilon$.

The constraint $\varepsilon < \min(w_d, w_p, w_l)$ is necessary to ensure that all three components of similarity are individually high, which is required for a small Hausdorff distance. $\qquad\square$

Based on this similarity measure, we design a hierarchical index structure that combines directional and positional indexing in Algorithm 5. The key insight behind our hierarchical line indexing approach is that line similarity in high-dimensional spaces can be decomposed into two primary components: directional similarity and spatial proximity. By organizing our index hierarchically, we can drastically reduce the search space and avoid expensive similarity computations with dissimilar lines.

---

**Algorithm 5** Hierarchical Line Index Construction

---

1: **Input:** Set of line segments $\mathcal{L}$, angular resolution $\nu$
2: **Output:** Hierarchical line index $\mathcal{I}$
3: {First level: directional partitioning}
4: Partition unit sphere into cells of angular resolution $\nu$
5: Create directional hash table $\mathcal{H}_d$ mapping direction cells to line sets
6: **for** each line segment $L = (a, b)$ in $\mathcal{L}$ **do**
7: $\quad dir \leftarrow \frac{b-a}{\|b-a\|}$ {Unit direction vector}
8: $\quad cell \leftarrow \text{DirectionToCell}(dir)$ {Determine directional cell}
9: $\quad$ Add $L$ to $\mathcal{H}_d[cell]$
10: **end for**
11: {Second level: spatial partitioning}
12: **for** each directional cell $c$ in $\mathcal{H}_d$ **do**
13: $\quad \mathcal{H}_d[c].\text{spatial\_index} \leftarrow \text{CreateSpatialIndex}(\mathcal{H}_d[c])$
14: **end for**
15: $\mathcal{I}.\text{directional\_index} \leftarrow \mathcal{H}_d$
16: **return** $\mathcal{I}$

---

**Algorithm 6** Find Nearest Line

---

1: **Input:** Query line $L_Q = (a_Q, b_Q)$, line index $\mathcal{I}$, similarity threshold $\tau$
2: **Output:** Most similar indexed line $L_{similar}$
3: $dir_Q \leftarrow \frac{b_Q - a_Q}{\|b_Q - a_Q\|}$ {Query direction}
4: $neighboring\_cells \leftarrow \text{GetNeighboringCells}(dir_Q, \nu)$ {Get directional cells}
5: $candidates \leftarrow \emptyset$
6: **for** each cell $c$ in $neighboring\_cells$ **do**
7: $\quad midpoint_Q \leftarrow \frac{a_Q + b_Q}{2}$ {Query midpoint}
8: $\quad length_Q \leftarrow \|b_Q - a_Q\|$ {Query length}
9: $\quad cell\_candidates \leftarrow \mathcal{I}.\text{directional\_index}[c].\text{spatial\_index}.\text{Search}(midpoint_Q, \kappa \cdot length_Q)$
10: $\quad$ Add $cell\_candidates$ to $candidates$
11: **end for**
12: $L_{similar} \leftarrow \text{null}$
13: $sim^* \leftarrow 0$
14: **for** each line $L$ in $candidates$ **do**
15: $\quad similarity \leftarrow \text{ComputeLineSimilarity}(L_Q, L)$ (Definition 8)
16: $\quad$ **if** $similarity > sim^*$ **then**
17: $\quad\quad sim^* \leftarrow similarity$
18: $\quad\quad L_{similar} \leftarrow L$
19: $\quad$ **end if**
20: $\quad$ **if** $sim^* > \tau$ **then**
21: $\quad\quad$ **return** $L_{similar}$
22: $\quad$ **end if**
23: **end for**
24: **return** $L_{similar}$

---

**Intuition** The hierarchical line index operates on the observation that two line segments with significantly different directions or distant spatial locations are unlikely to be similar. Algorithm 5 implements this insight by partitioning the index into two levels: the first level groups lines by their direction vectors, while the second level organizes lines within each directional group according to their spatial locations. This structure enables efficient pruning of the search space when finding the nearest line to a query.

**Algorithm Process** The index construction (Algorithm 5) proceeds in two main phases:

1. **Directional Partitioning:** We first discretize the unit sphere into cells of angular resolution $\nu$, effectively creating buckets for different line directions. Each line segment is assigned to a cell based on its normalized direction vector. This partitioning allows us to quickly identify lines with similar orientation to a query line.

2. **Spatial Indexing:** Within each directional cell, we build a spatial index (such as an R-tree or k-d tree) to organize the lines based on their spatial positions, typically represented by their midpoints. This second-level index enables efficient retrieval of spatially proximate lines within a directional group.

The search algorithm (Algorithm 6) leverages this hierarchical structure to efficiently locate the most similar line to a query:

1. **Directional Filtering:** We first identify candidate directional cells based on the query line's direction. This step immediately eliminates vast portions of the index containing lines with significantly different orientations.

2. **Spatial Filtering:** Within each candidate directional cell, we use the spatial index to retrieve lines near the query line's location. We use the query line's midpoint as the search center and adjust the search radius proportionally to the line's length using parameter $\kappa$.

3. **Similarity Ranking:** Finally, we compute the exact similarity between the query line and each candidate, maintaining the best match found. The early termination condition ($sim^* > \tau$) allows us to return immediately if we find a sufficiently similar line, avoiding unnecessary computations.

**Complexity Analysis** The time complexity of index construction is $O(N \log N)$, where $N$ is the number of line segments. Specifically, assigning each line to a directional cell takes $O(N)$ time, while building the spatial indices across all cells requires $O(N \log N)$ time in the worst case. The space complexity is $O(N)$ for storing all line segments.

For the search operation in Algorithm 6, the time complexity is $O(\log N + C)$, where $C$ is the number of candidate lines retrieved for exact similarity computation. In the worst case where all lines share similar directions, $C$ could approach $N$, but in practice, the directional and spatial filtering steps typically reduce the candidate set to a small fraction of the dataset, resulting in near-logarithmic query time. The parameter $\nu$ controls the trade-off between query time and index size—smaller values of $\nu$ create more directional cells, potentially reducing $C$ at the expense of increased index size.

### G.6 CYLINDRICAL DISTANCE INDEXING

For each indexed line segment, we need an efficient structure to retrieve points within a specified distance of the line:

**Definition 9** (Cylindrical Coordinates). *For a line segment $L = (a, b)$ and a point $p$, the cylindrical coordinates are:*

$$t = clamp\left(\frac{(p-a)\cdot(b-a)}{\|b-a\|^2}, 0, 1\right) \tag{224}$$

$$r = \|p - (a + t(b-a))\| \tag{225}$$

$$\theta = angle\ in\ plane\ perpendicular\ to\ line\ direction \tag{226}$$

*where $clamp(x, min, max) = \min(\max(x, min), max)$ restricts the value of $x$ to the range $[min, max]$ (see Figure 3(f)). The parameter $t$ represents the normalized projection of point $p$ onto the line segment, $r$ is the perpendicular distance from $p$ to the line, and $\theta$ is the angular position around the line.*

**Intuition**    The cylindrical indexing approach leverages the geometric properties of distance relation-ships in our transformed space. When searching for points near a line segment, points that are similar tend to cluster in cylindrical regions around the line. Our indexing structure exploits this property by partitioning the space around each reference line into cylindrical sections, organizing points based on both their position along the line and their radial distance from it. This organization enables efficient pruning of distant points during query processing.

**Corollary 7** (Cylindrical Search Properties). *For points indexed in cylindrical coordinates relative to line $L$:*

1. *A point is within distance $R$ of line $L$ if and only if $r \leq R$*

2. *For points with similar $t$ values, their Euclidean distance is primarily determined by their $r$ values*

3. *The set of points within distance $R$ of line $L$ forms a cylinder of radius $R$ around $L$*

Based on these properties, we design an efficient cylindrical index structure:

---
**Algorithm 7** Cylindrical Index Construction

---
1: **Input:** Line segment $L = (a, b)$, point set $\mathcal{P}$, radius $R$
2: **Output:** Cylindrical index $\mathcal{C}$
3: $\mathcal{C}$.line $\leftarrow L$
4: $\mathcal{C}$.max_radius $\leftarrow R$
5: $length \leftarrow \|b - a\|$
6: $num\_sections \leftarrow \max(1, \lceil length/R \rceil)$ {Partition line into sections}
7: Initialize array $sections[num\_sections]$ of empty sets
8: **for** each point $p$ in $\mathcal{P}$ **do**
9:     Compute cylindrical coordinates $(t, r, \theta)$ for $p$ relative to $L$
10:    **if** $r \leq R$ **then**
11:        $section\_idx \leftarrow \min(\lfloor t \cdot num\_sections \rfloor, num\_sections - 1)$
12:        Add $(p, r)$ to $sections[section\_idx]$
13:    **end if**
14: **end for**
15: **for** $i = 0$ to $num\_sections - 1$ **do**
16:    Build radius-based index for $sections[i]$ {E.g., using a ball tree}
17: **end for**
18: $\mathcal{C}$.sections $\leftarrow sections$
19: **return** $\mathcal{C}$

---

**Algorithm Process**    The cylindrical index construction (Algorithm 7) proceeds through several key steps:

1. **Line Segmentation:** We divide the reference line segment into multiple sections of ap-proximately equal length (proportional to the cylinder radius). This partitioning allows for more localized searches and avoids examining the entire cylinder when only a portion might contain relevant points.

2. **Cylindrical Projection:** For each point in the dataset, we compute its cylindrical coordinates relative to the reference line: the normalized projection position along the line ($t$), the perpendicular distance from the line ($r$), and the angular position around the line ($\theta$).

3. **Sectional Organization:** Points are assigned to sections based on their projection position $t$, and only points within the maximum radius $R$ are included in the index. This filtering step immediately eliminates points that cannot be retrieved by any valid query.

4. **Per-Section Indexing:** Within each section, we build a specialized radius-based index (such as a ball tree) to efficiently support radius-based queries. This nested indexing structure allows for rapid retrieval of points within a specified distance of any position along the line.

The cylinder search algorithm (Algorithm 8) utilizes this structure to efficiently retrieve points near a query line:

---

**Algorithm 8** Cylinder Search

---

1: **Input:** Line segment $L_Q = (a_Q, b_Q)$, radius $R_Q$, cylindrical index $\mathcal{C}$
2: **Output:** Points within distance $R_Q$ of $L_Q$
3: $L \leftarrow \mathcal{C}$.line {Indexed line}
4: $R \leftarrow \mathcal{C}$.max_radius {Indexed radius}
5: $d_H \leftarrow$ HausdorffDistance$(L, L_Q)$ {Line distance}
6: $adjusted\_radius \leftarrow R_Q + d_H$ {Adjust for line difference}
7: $results \leftarrow \emptyset$
8: **if** $adjusted\_radius > R$ **then**
9:    **return** "Radius too large for this index"
10: **end if**
11: **for** each section $i$ in $\mathcal{C}$.sections **do**
12:    $t_{min} \leftarrow i/num\_sections$
13:    $t_{max} \leftarrow (i+1)/num\_sections$
14:    $closest\_distance \leftarrow$ MinDistanceBetweenLineSegments$(L_Q, L.\text{Subsegment}(t_{min}, t_{max}))$
15:    **if** $closest\_distance \leq adjusted\_radius$ **then**
16:      $section\_candidates \leftarrow \mathcal{C}$.sections$[i]$.GetPointsWithinRadius$(adjusted\_radius)$
17:      **for** each point $p$ in $section\_candidates$ **do**
18:         $dist\_to\_query \leftarrow$ DistanceToLine$(p, L_Q)$
19:         **if** $dist\_to\_query \leq R_Q$ **then**
20:            Add $p$ to $results$
21:         **end if**
22:      **end for**
23:    **end if**
24: **end for**
25: **return** $results$

---

1. **Radius Adjustment:** We first compute the Hausdorff distance between the indexed line and the query line, then adjust the search radius accordingly. This step accounts for the difference between lines and ensures we capture all relevant points.

2. **Section Filtering:** For each section, we compute the minimum distance between the corresponding subsegment of the indexed line and the query line. Sections whose minimum distance exceeds the adjusted radius are immediately pruned from consideration.

3. **Candidate Retrieval:** For each relevant section, we retrieve candidate points within the adjusted radius using the section's radius-based index.

4. **Exact Distance Verification:** Finally, we compute the exact distance from each candidate point to the query line and filter out points whose distance exceeds the original query radius $R_Q$.

**Complexity Analysis** The time complexity for constructing the cylindrical index is $O(n \log n)$ where $n$ is the number of points within the maximum radius $R$ of the line. Specifically, computing cylindrical coordinates for all points takes $O(n)$ time, while building the radius-based indexes requires $O(n \log n)$ time in the worst case.

For the search operation, the time complexity is $O(s + k \log n_s)$, where $s$ is the number of sections, $k$ is the number of candidate points examined, and $n_s$ is the average number of points per section. The section filtering step takes $O(s)$ time, while the retrieval and verification of candidates takes $O(k \log n_s)$ time. In practice, section filtering typically eliminates a large portion of the cylinder, making the effective value of $k$ much smaller than the total number of points in the cylinder. The number of sections $s$ is chosen as $\max(1, \lceil \text{length}/R \rceil)$, balancing the overhead of section processing with the benefit of finer spatial partitioning.

G.7  ERROR ANALYSIS AND ADAPTATION

Similar approach in Foster et al. (2018); Heidari et al. (2020a), when using a similar indexed line as a proxy for the query line, we need to account for the approximation error similar approach:

**Theorem 20** (Error Compensation). *Let $L_Q$ be a query line and $L_{similar}$ be the most similar indexed line with Hausdorff distance $\delta_H = d_H(L_Q, L_{similar})$. To retrieve the top-k nearest neighbors with probability at least $1 - \epsilon$, we need to:*

1. *Increase the search radius by $\delta_H$*

2. *Retrieve $k' = k + \lceil c \cdot \log(1/\epsilon) \cdot \delta_H \cdot \eta \rceil$ candidates*

*where $\eta$ is the local density factor and $c$ is a constant that depends on the data distribution.*

*Proof.* For the radius adjustment, consider a point $p$ that is within distance $r$ of $L_Q$. By the triangle inequality, its distance to $L_{similar}$ is at most $r + \delta_H$. Therefore, to ensure we capture all points within distance $r$ of $L_Q$, we need to search within distance $r + \delta_H$ of $L_{similar}$.

For the result count adjustment, we need to account for the fact that points may be ranked differently with respect to $L_Q$ and $L_{similar}$. The number of points affected depends on the local density $\eta$ and the perturbation $\delta_H$.

Using concentration inequalities, the probability that more than $\delta_H \cdot \eta \cdot \log(1/\epsilon)$ points change their ranking status (from top-k to outside top-k or vice versa) is less than $\epsilon$. Therefore, retrieving $k' = k + \lceil c \cdot \log(1/\epsilon) \cdot \delta_H \cdot \eta \rceil$ candidates ensures capturing the true top-k with probability at least $1 - \epsilon$. $\qquad\square$

---

**Algorithm 9** Adaptive k' Selection

1: **Input:** Query line $L_Q$, similar line $L_{similar}$, target k, error probability $\epsilon$
2: **Output:** Adjusted k' value
3: $\delta_H \leftarrow d_H(L_Q, L_{similar})$ {Hausdorff distance}
4: $\eta \leftarrow$ EstimateLocalDensity($L_{similar}$) {Estimate local density}
5: $c \leftarrow 2.0$ {Constant factor based on empirical analysis}
6: $k' \leftarrow k + \lceil c \cdot \log(1/\epsilon) \cdot \delta_H \cdot \eta \rceil$
7: **return** $k'$

---

**Theorem 21** (Density Estimation). *The local density factor $\eta$ around a line segment $L = (a, b)$, where $a$ and $b$ are the endpoints of $L$, can be estimated as:*

$$\eta \approx \frac{N_r}{V_r} = \frac{N_r}{\pi r^2 \cdot \|b - a\|} \tag{227}$$

*where $N_r$ is the number of points within distance $r$ of $L$, and $V_r$ is the volume of the cylinder with radius $r$ around $L$.*

*Proof.* The density factor $\eta$ measures the concentration of data points in the neighborhood of line segment $L$. To estimate this density, we consider the ratio of points within a cylindrical region around the line to the volume of that region.

For a line segment $L$ with endpoints $a$ and $b$, the cylindrical region with radius $r$ around $L$ consists of all points within perpendicular distance $r$ of any point on $L$. The volume of this cylinder is given by:

$$V_r = \pi r^2 \cdot \|b - a\| \tag{228}$$

This follows from the standard formula for the volume of a cylinder: $V = \pi r^2 h$, where $r$ is the radius and $h$ is the height. In our case, the height corresponds to the length of the line segment $\|b - a\|$.

Let $N_r$ denote the number of data points falling within this cylindrical region. The ratio $\frac{N_r}{V_r}$ then gives us the average number of points per unit volume in the vicinity of line segment $L$, providing a direct estimate of the local point density.

This density estimate is particularly relevant for error compensation analysis because it helps predict how many additional points might need to be examined when approximating a query line with a similar indexed line. Higher density regions require examining more candidates to maintain the same probability of capturing the true nearest neighbors. $\qquad\square$

**Intuition** The density estimation theorem provides a crucial metric for adapting our range query parameters to the local characteristics of the data distribution. Intuitively, the density factor $\eta$ measures how "crowded" the space is around a particular line segment. This has direct implications for approximation error handling—in high-density regions, small deviations between a query line and its approximation can affect many more points than in sparse regions. The formula expresses this density as points per unit volume in the cylindrical neighborhood around the line, giving us a locally adaptive measure for error compensation.

**Algorithm Process**  Computing the density factor involves these key steps: (1) identifying all points within distance $r$ of the line segment using cylindrical coordinates, (2) counting these points to determine $N_r$, (3) calculating the cylinder volume using the line length and radius, and (4) computing their ratio. In practice, we can efficiently estimate this density using the cylindrical index structure without explicitly enumerating all points. The density factor is typically calculated during index construction and stored with each indexed line segment, then used during query time to dynamically adjust the search parameters based on Theorem 20.

**Complexity Analysis**  The computational complexity of estimating the density factor is $O(N + \log N)$ where $N$ is the total number of indexed points. The dominant cost comes from identifying points within radius $r$ of the line, which requires $O(\log N)$ time with an efficient spatial index, plus $O(N_r)$ time to process those points. Since the density calculation is performed during index construction and cached, it adds minimal overhead to query processing.  The additional space complexity is $O(M)$ where $M$ is the number of indexed line segments, as we need to store one density value per line. This small storage investment enables significant query performance gains through adaptive parameter selection, particularly in datasets with heterogeneous density distributions.

---

**Algorithm 10** Complete Range Query Processing

---

1: **Input:** Query vector $q$, range $[l, u]$, number of results $k$, error probability $\epsilon$
2: **Output:** Top-$k$ nearest neighbors within range $[l, u]$
3: {Phase 1: Query preparation}
4: $p_l \leftarrow \Psi(q, l, \alpha, \beta)$
5: $p_u \leftarrow \Psi(q, u, \alpha, \beta)$
6: $L_Q \leftarrow \text{LineSegment}(p_l, p_u)$
7: {Phase 2: Find similar indexed line}
8: $L_{similar} \leftarrow \text{FindNearestLine}(L_Q, line\_index)$
9: $\delta_H \leftarrow d_H(L_Q, L_{similar})$
10: $base\_radius \leftarrow L_{similar}.\text{cylinder\_radius}$
11: $adjusted\_radius \leftarrow base\_radius + \delta_H$
12: {Phase 3: Determine search parameters}
13: $\eta \leftarrow \text{EstimateLocalDensity}(L_{similar})$
14: $k' \leftarrow k + \lceil 2 \cdot \log(1/\epsilon) \cdot \delta_H \cdot \eta \rceil$
15: {Phase 4: Retrieve candidates}
16: $candidates \leftarrow \text{CylinderSearch}(L_Q, adjusted\_radius, L_{similar}.\text{cylinder\_index})$
17: {Phase 5: Filter and refine results}
18: $filtered\_candidates \leftarrow \emptyset$
19: **for** each point $p = \Psi(v, f, \alpha, \beta)$ in $candidates$ **do**
20:   **if** $l \le f \le u$ **then**
21:     $distance \leftarrow \|v - q\|$
22:     Add $(v, f, distance)$ to $filtered\_candidates$
23:   **end if**
24: **end for**
25: Sort $filtered\_candidates$ by distance
26: **return** Top-$k$ records from $filtered\_candidates$

---

G.8  COMPLETE RANGE QUERY ALGORITHM

Putting all components together, we present the complete range query in Algorithm 10.

**Theorem 22** (Query Complexity). *The complete range query algorithm has expected time complexity:*

$$O(\log L + \log P + k \log(1/\epsilon) + k \log k) \tag{229}$$

*where $L$ is the number of indexed line segments, $P$ is the maximum number of points in any cylindrical index, $k$ is the number of requested results, and $\epsilon$ is the error probability. Since $L, P \le N$ (where $N$ is the total dataset size), this simplifies to $O(\log N + k \log(1/\epsilon) + k \log k)$.*

*Proof.* The algorithm consists of these main steps:

1. Finding the nearest line: $O(\log L)$ using the hierarchical line index (Algorithm 6), where $L$ is the number of indexed line segments from the adaptive sampling algorithm (Theorem 17)

2. Cylinder search: $O(\log P + k')$ where $P$ is the number of points in the relevant cylindrical index and $k' = O(k \log(1/\epsilon))$ from Theorem 20

3. Filtering and ranking: $O(k' \log k)$ to sort the candidates

Combining these terms and noting that both $L$ and $P$ are bounded by the total dataset size $N$, we get the simplified complexity $O(\log N + k \log(1/\epsilon) + k \log k)$. □

The complete algorithm (Algorithm 10) provides strong theoretical guarantees while maintaining practical efficiency for large-scale datasets, making it an ideal solution for range-constrained vector search problems.

# H  THEOREMS, COROLLARIES, AND ALGORITHMS CHEAT SHEET

In this section, we provide a summary of key concepts and findings.

Table 9: Summary of Theorems, Corollaries, and Algorithms in FUSEDANN Paper

| Name/Type | Label/Ref | Functionality / Statement |
|---|---|---|
| **Single-Attribute Hybrid Vector Indexing (FusedANN)** | Alg. 1 | Core algorithm for fusing content and attribute vectors via transformation $\Psi$ for hybrid vector search, supporting both offline indexing and online query with parameterized separation and candidate selection. |
| **Properties of $\Psi$ Transformation** | Theorem 1 | Transformation preserves k-NN order within attribute groups, increases inter-attribute distances with $\alpha$, and controls scaling with $\beta$. |
| **Practical Candidate Set Size** | Theorem 2 | Provides formula for number of candidates $k'$ needed to guarantee recall in hybrid search, based on attribute cluster statistics and separation. |
| **Expected Candidate Set Size** | Theorem 3 | Gives the expected $k'$ across queries based on attribute distribution, showing $k' \to k$ as separation increases. |
| **Parameter Selection for $\epsilon_f$-bounded Clusters** | Theorem 4 | Gives minimum values for $\alpha, \beta$ to ensure attribute cluster compactness and inter-cluster separation in fused space. |
| **Optimality of Minimal Parameters** | Cor. 1 | Setting $\beta, \alpha$ as per Theorem 4 yields minimum separation/compactness bounds, balancing recall and efficiency. |
| **Uniqueness of Transformation** | Theorem 5 | Shows that $\Psi$ is injective (one-to-one) if $d > m$ and parameters satisfy minimal bounds. |
| **Property Preservation** | Theorem 6 | Order of k-NN among records with the same attributes is preserved under sequential application of $\Psi$. |
| **Attribute Priority** | Theorem 7 | Later-applied attributes in $\Psi$ sequence have higher effective priority in determining k-NN order. |
| **Attribute Match Distance Hierarchy** | Theorem 9 | Records with more matching attributes are always closer to the query (after transformation) than those with fewer matches. |
| **Generalized Attribute Match Hierarchy** | Theorem 10 | For any two records, there exist $\alpha_j$ such that more attribute matches always yield smaller fused distance. |

**Table 9 – continued from previous page**

| Name/Type | Label/Ref | Functionality / Statement |
|---|---|---|
| **Monotone Priority in FUSEDANN** | Theorem 8 | ANNS in the fused space yields results that satisfy the monotone attribute priority property for hybrid queries. |
| **Multi-Attribute Candidate Set Size** | Theorem 11 | Extends candidate selection formula to multi-attribute (hierarchical) fused space; $k'$ shrinks as more attributes are used. |
| **Hierarchical Multi-Attribute Vector Indexing** | Alg. 3 | Complete indexing and query algorithm for multi-attribute hybrid queries, applying $\Psi$ recursively and managing cluster statistics. |
| **Range Query Line** | Theorem 15 | Set of all fused query points for attribute in $[l, u]$ forms a line segment in fused space. |
| **Distance Characterization (Range)** | Theorem 16 | Distance from a point to the query range line is proportional to vector similarity, enabling cylinder search interpretation. |
| **Optimal Range Line Sampling** | Theorem 17 | Gives sample complexity for covering the fused range-query space with pre-indexed lines (cylinders) for range queries. |
| **Optimal Cylinder Radius** | Theorem 18 | Formula for radius to guarantee recall for range queries, based on $k$-th neighbor distance and local statistics. |
| **Line Similarity Measure/Properties** | Def. 8, Theorem 19 | Defines a composite metric for line similarity; proves its bounds and relation to Hausdorff distance. |
| **Hierarchical Line Index Construction** | Alg. 5 | Builds two-level index for fast retrieval of similar lines: first by direction, then by spatial proximity. |
| **Find Nearest Line** | Alg. 6 | Searches the hierarchical index to find the closest pre-indexed line to a query line. |
| **Cylindrical Index Construction** | Alg. 7 | Builds an index for each line, partitioning points by distance to the line (for efficient range/cylinder search). |
| **Cylinder Search** | Alg. 8 | Retrieves all points within a specified radius of a line (i.e., inside a cylinder) using the cylindrical index. |
| **Adaptive Range Line Sampling** | Alg. 4 | Strategy for sampling lines (cylinders) to cover the fused range-query space adaptively, based on empirical distributions. |
| **Adaptive k' Selection** | Alg. 9 | Adjusts the number of candidates $k'$ for range queries to compensate for line approximation error and local density. |
| **Complete Range Query Processing** | Alg. 10 | End-to-end algorithm for efficient range queries: transforms the query, finds similar pre-indexed cylinder, adjusts search, retrieves and ranks results. |
| **Query Complexity** | Theorem 22 | Shows that the complete range query algorithm has $O(\log N + k \log(1/\epsilon) + k \log k)$ expected time. |