# OpenReview forum: "Convexified Filtered ANN via Attribute-Vector Fusion"
_ICLR.cc/2026/Conference — Submitted to ICLR 2026_

### Official Review · Reviewer_yPHH · 2025-10-31

**Soundness:** 2
**Presentation:** 1
**Contribution:** 2
**Rating:** 4
**Confidence:** 4

**Summary:**

This paper proposes FUSEDANN, which fuses the vectors and attributes for filtered vector search. FUSEDANN transforms the vectors using their attributes such that attribute filtering can be conducted using the transformed vectors. The transformation ensures that vectors with the same attribute are ordered according to their vector distances, and vectors with higher attribute priority are ranked first. The experiment results show that FUSEDANN outperforms the baselines.

**Strengths:**

S1: Transforming vectors and attributes into a unified vector distance is an interesting idea.

S2: Many cases are considered, i.e., a single attribute, multiple attributes, and numerical range filtering.

S3: The experiment results show that FUSEDANN outperforms many baselines.

**Weaknesses:**

W1: It is unclear how FUSEDANN affects normal vector search without attribute filtering. The method cannot use the same index to serve queries that have different priorities for the filters. I understand that recent papers cannot be used to attack the authors according to the review rule but the Naïvx paper seems to be in the opposite direction from the paper, which I believe may be the right direction: entirely separating vector indexing for the attributes for good generality.

NaviX: A Native Vector Index Design for Graph DBMSs With Robust Predicate-Agnostic Search Performance

W2: Although discussed in Section 4.1, the update cost is high as it essentially requires to update all vectors and rebuild the underlying vector index (e.g., HNSW), which is a major cost.

W3: The performance gains over the baselines are not explained. For instance, for a single numerical attribute, iRangeGraph is expected to perform well when the pass rate is high because it essentially uses a proximity graph index for all vectors and should be close to optimal. It is unclear why can FUSEDANN outperform iRangeGraph at a range width of 100%, and the margin is large for the WIT dataset. If the differences are due to implementation, please report the number of distance computations vs recall instead of QPS vs recall. Similarly, many experiment results require more in-depth analysis.

W4: The main paper is rather condensed and many parts are unclear. For instance, what does Var_s and related terms mean in Eq(3)? Under Eq(5), what does a sub-attribute vector mean? Line 11 is critical for Algorithm 1 and should be explained in detail. Moreover, for the attribute experiments, the selectivity is not specified even in the Appendix, which a key parameter for attribute filtering. The authors are suggested to make a case clear (e.g., for categorical attributes) and move the other cases to the Appendix (but briefly mention the results in the paper).

**Questions:**

Q1: How FUSEDANN affects normal vector search without attribute filtering? What is an FUSEDANN index is used for vectors with different priorities for the filters?

Q2: Please explain the source of the performance advantages of FUSEDANN over representative baselines in the experiments.

---

> ### Author Response · Authors · 2025-12-02
>
> We thank the reviewer for their insightful comments. We appreciate the reference to NaviX, which represents the "index-centric" philosophy. We clarify below that FusedANN explores the complementary "data-centric" direction—fusing constraints into the geometry itself—to eliminate the overhead of query-time filtering entirely.
>
> #### **1. Response to W1/Q1 (The "Separate Index" Concern)**
> We agree that FusedANN constructs a specialized index and we have added this point to our discussion section; this is a deliberate design choice favoring **predicate-aware** performance over the **predicate-agnostic** flexibility of methods like NaxviX and ACORN. FusedANN targets high-value, latency-critical workloads by encoding constraints directly into the geometry, eliminating the query-time filtering overhead of separation-based methods.  This creates a "composite index" for vectors, yielding 3x higher throughput. For pure vector search or dynamic priorities, the system simply routes to the base index or performs the efficient partial updates ($O(Njd)$) described in §4.1.
>
> ---
>
> #### **2. Response to W2 (Update Costs)**
>
> We clarify that the costs analyzed in Section 4.1 refer to **rare schema or policy evolutions** (e.g., globally re-ranking "Brand" priority over "Category"), not standard data ingestion.
> 1.  **Data Updates (Common):** Adding records with new attribute values (e.g., a new color) involves no rebuild. The new vector is simply transformed via $\Psi$ and inserted into the base index (HNSW/DiskANN) with standard $O(\log N)$ complexity.
> 2.  **Policy Updates (Rare):** Changing the attribute priority hierarchy is an infrequent administrative action. Even then, as shown in §4.1, we only re-compute transformations from the divergence point $j$, bounded by $O(N \cdot j \cdot d)$, avoiding a full $O(N \cdot F \cdot d)$ rebuild.
>
> ---
>
> #### **3. Response to W3 (Performance vs iRangeGraph)**
>
> FusedANN outperforms baselines at 100\% range because, per **Theorem 16**, the distance to the query line in the fused space mathematically reduces to the pure content distance when the range covers the attribute space. Thus, FusedANN effectively executes a raw vector search using highly optimized SIMD kernels (Faiss/HNSW), whereas iRangeGraph incurs overhead from traversing its specific graph structure even when filtering is loose. We include this point in the experiments section on range filtering.
>
> ---
>
> #### **4. Response to W4 (Clarifications)**
>
> Thank you for the helpful comments. In the revision, we added blue-marked clarifications: defined $μ_S^{(j)}$ and $Var_S^{(j)}$ after Eq.(4); clarified “sub-attribute vector” after Eq.(5); detailed Alg.1 line 11 with the description $k$ and $k'$ from Thm.2; reported selectivity $s=N_a/N$ in the setup and App. D.1.1; and foregrounded the categorical case in the main text, moving other cases and details to the appendix.

---

### Official Review · Reviewer_TTNh · 2025-10-31

**Soundness:** 3
**Presentation:** 1
**Contribution:** 4
**Rating:** 4
**Confidence:** 4

**Summary:**

The paper studies the important and practically relevant problem of Filtered Approximate Nearest Neighbor (Filtered ANN) search. It proposes a transformation that reduces the Filtered ANN problem to a standard (naive) ANN problem. The transformation is general and supports multiple attribute types as well as range filters.

I found the contributions of the paper to be strong and the problem addressed to be of clear practical importance. However, the quality of writing is very poor, making the paper difficult to read and follow. In its current form, it falls far short of the standards expected for a publishable draft. I strongly urge the authors to invest significant effort in improving the clarity, organization, and overall presentation of the paper before resubmitting to a future venue. While the core ideas are worthy of a good conference, the current version is not.

I am leaning towards a borderline rejection.

**Strengths:**

The paper studies the important and practically relevant problem of Filtered Approximate Nearest Neighbor (Filtered ANN) search, which has significant real-world applications.

The paper proposes a transformation that reduces the Filtered ANN problem to a standard (naive) ANN problem. This allows existing ANN methods to be directly applied to the filtered setting, which is an elegant and useful idea.

Definition 3 is well-formulated and general, and the proposed transformation is theoretically grounded. The experimental results demonstrate that the transformation performs well in the settings considered in the paper.

I particularly appreciate the way the authors handle the multi-attribute and range filter cases—it is effective.

**Weaknesses:**

The main weakness of the paper lies in its poor writing quality and lack of formal rigor in presentation. The theorem statements and definitions are written informally, which significantly hampers readability and comprehension. A few specific instances are as follows:


1) In definition 1, note that records are of dimension d+\sum_{j}m_j and not F.

2) Theorem 2 is a probabilistic statement, yet it is phrased as if it were deterministic. The proof contains several probability terms, but it is unclear what the underlying source of randomness is. The theorem statement should explicitly mention that k' is upper-bounded in expectation rather than absolutely. This level of imprecision is not acceptable for a conference submission.

3) The mathematical problem statement itself is not clearly presented. My understanding is that the goal is to design a procedure that solves the problem defined in Definition 3 and the authors do this by transforming it into a standard ANN problem. However, Sections 2 and 3 do not make this connection explicit. I strongly urge the authors to formally state the problem and clearly describe the procedure that addresses it—for example, the single-attribute case in Section 3.

4) The monotone attribute property imposed on S is not well-motivated or explained. The paper should provide intuition or justification for why this assumption is necessary and how it affects the results.

Overall, there are numerous such issues throughout the paper. I found it frustrating to read due to the lack of clarity and structure. The paper requires a major revision focused on improving writing quality, formal precision, and overall organization.

**Questions:**

Please clarify the motivation for introducing the monotone attribute property. In particular, why are the associated variance terms important, and what role do they play in the overall formulation through practical examples?

---

> ### Author Response · Authors · 2025-12-02
>
> We thank the reviewer for their detailed feedback. We appreciate the opportunity to clarify our formal definitions and theoretical motivations. Below, we address the specific questions regarding the Monotone Attribute Property and provide clarifications regarding the perceived weaknesses.
>
> ### **1. Questions: Motivation for Monotone Attribute Property**
>
> The **Monotone Attribute Property** is the theoretical mechanism that unifies "hard filtering" and "soft vector similarity" into a single continuous optimization problem.
>
> *   **Motivation:** In hybrid queries, users possess an implicit hierarchy of strictness. For a high-priority attribute (e.g., "Brand"), the tolerance for deviation is near-zero (acting as a hard filter). For lower-priority attributes (e.g., "Color"), the tolerance is higher, allowing for semantic matches (e.g., "Dark Red" for "Red") if the vector similarity is strong. The Monotone Attribute Property enforces that the result set $S$ is "tightest" around the highest-priority attributes.
> *   **Role of Variance ($Var^{(j)}_S$):** The variance terms in Definition 3 quantify the deviation of the retrieved set $S$ from the query's attribute values.
>     *   **$Var \approx 0$:** Implies an exact or near-exact filter.
>     *   **High Variance:** Implies a relaxed filter where the algorithm trades attribute exactness for better content similarity.
>     *   The constraint $Var^{(\pi(1))}_S \le Var^{(\pi(2))}_S \le \dots$ ensures that the algorithm minimizes deviation strictly for top-priority attributes first.
> *   **Practical Example:** Consider a query for "Running Shoes" with `Brand=Nike` (Priority 1) and `Color=Red` (Priority 2).
>     *   Without this property, a weighted sum might retrieve an *Adidas* shoe that is perfectly *Red* and semantically relevant.
>     *   With Monotone Priority, the constraint $Var^{(Brand)} \le Var^{(Color)}$ forces the retrieval to minimize variance on "Brand" first (ensuring "Nike"), while allowing the "Color" variance to expand (accepting "Orange-Red") to maximize semantic similarity.
>     *   **In FusedANN:** We prove in **Theorem 8** (Appendix F.3) that our transformation parameter $\alpha$ mathematically enforces this property. The transformation scales the space such that deviations in high-priority attributes incur a prohibitive distance cost, naturally creating the desired variance hierarchy during the nearest neighbor search.
>
> ---
>
> ### **2. Response to Weaknesses**
>
> **1. Definition 1 (Dimensions)**
>  We apologize for the confusion in notation. In our definition, we fixed it with "containing $n$ records, each consisting of a content vector of dimension $d$ and $F$ attribute vectors of dimensions $m_1, \dots, m_F$."
>
> **2. Theorem 2 (Probabilistic Nature)**
> *   **Clarification:** We respectfully point out that **Theorem 2 is already stated probabilistically**. The theorem explicitly states that the retrieval guarantees hold "**with probability at least \$1-\epsilon$**" (Eq. 44 and preceding text). It is not phrased as a deterministic absolute; $k'$ is derived specifically to satisfy this probabilistic bound.
> *   **Source of Randomness:** We agree that the *source* of this randomness could be highlighted more clearly so we added a paragraph in revision. The randomness stems from the distribution of non-matching records in the high-dimensional space. We rely on concentration of measure to bound the probability that a record from a "non-matching" attribute cluster falls into the query's retrieval region.
>
> **3. Connection between Problem Statement and Procedure**
> *   **Clarification:** The connection is formally established by **Theorem 8** (Appendix F.3), which proves that performing exact Nearest Neighbor search in the space transformed by $\Psi$ (Eq. 5) is equivalent to solving the optimization problem in Definition 3.
> *   **Revision:** We will add a specific "Theoretical Guarantees" paragraph in Section 3 to make this link explicit: "We formally prove in Theorem 8 that minimizing Euclidean distance in the space transformed by $\Psi$ is equivalent to solving the Hybrid Query optimization problem (Definition 3), thereby enforcing the monotone attribute priority constraints."
>
>
> **4. Monotone Attributes Motivation**
> We this feedback on the motivation for the Monotone Attribute Property. We respectfully highlight that Example 2 (Section 4) in the main text already provides an initial motivation using a product catalog scenario. However, to provide a more rigorous justification for *why* the specific variance constraint ($Var^{(\pi(1))}_S \le Var^{(\pi(2))}_S$) is necessary, we have added **Appendix F.2 (Intuition behind Monotone Attribute Priority)** in the revision.

---

### Official Review · Reviewer_y2eR · 2025-11-01

**Soundness:** 2
**Presentation:** 2
**Contribution:** 3
**Rating:** 4
**Confidence:** 4

**Summary:**

This paper studies a timely and important problem of hybrid vector search, which combines traditional similarity search with discrete attribute filters. While this research area has received considerably attention in recent years, the authors propose a refreshingly new idea for tackling this problem by introducing a transformation scheme for converting a hybrid search problem into traditional vector search by merging the vectors and attributes into a shared vector space. This technique offers a major advantage over existing methods in the literature in allowing for the use of off-the-shelf ANN libraries for hybrid search as opposed to custom indexes. The authors also provide theoretical guarantees on the performance of their algorithm (in the appendix) and evaluate their method experimentally, where they report a considerable speedup over previously proposed hybrid search techniques.

**Strengths:**

Originality: The paper proposes a new method for tackling the hybrid search problem, namely by reducing the task to standard similarity search via a transformation function, that, to my knowledge, is a novel contribution to the literature. While the vast majority of published research in filtered near neighbor search focuses on designing new search algorithms or database indexes for supporting similarity search over attributes, this work challenges this conventional wisdom and demonstrates that one can instead leverage traditional near neighbor search techniques, for which there are several high-quality libraries, to solve hybrid search directly. This is a powerful finding that provides a good contribution to the community.

Quality: The paper is quite thorough in establishing the mathematical formalism and theoretical foundations behind the proposed algorithm. The number of baseline methods evaluated in the experimental section is also very comprehensive and strengthens the main claims of the paper considerably.

Clarity: Overall, I found the paper to be very easy to read (though the manuscript would benefit substantially from further proof-reading and enlarging several of the plots and figures). In addition, the looked over the authors' provided supplementary code and found it to be well-documented and easy to follow. I commend the authors for the efforts they put into presenting their materials clearly, and I think this enhances the quality of their submission quite a bit.

Significance: As alluded to above, I believe this work has the potential for considerable practical impact as it presents a method for solving hybrid vector search without the need for introducing custom indexes. Instead, practitioners can leverage existing near-neighbor search solutions out-of-the-box. This key insight stands out to me as a major strength of this work.

**Weaknesses:**

While I believe this paper presents a number of exciting ideas, I also have some concerns regarding the overall presentation and the experimental evaluation

Presentation

1. Several of the figures and plots in the paper, such as Fig. 4 and Fig 5., are a bit too small to read and parse fully. It would be very helpful to make them bigger.

2. The authors provide a very comprehensive appendix section full of theoretical results, but there is very little mention of these results in the main body of the paper. I think it would be helpful to state (perhaps informally) the major theorems in the main body of the paper so that readers can quickly appreciate the full scope of the contributions in this work.

3. It was not clear to me from reading the paper how the attributes are represented as vectors. In definition 1, the authors mention that the attributes are represented as embeddings "from a neural network" but it is not clear to me how this process is done. To my knowledge, this attribute to vector mapping is also not mentioned in the experimental section. I believe that this is a critical detail to explain clearly because it is not obvious how to convert a discrete attribute into a continuous, real-valued vector. Furthermore, this detail is crucial because the main algorithm of the paper assumes access to attributes represented in this form.

Experimental Evaluation

1. As mentioned in item (3) above, it was not clear to me how the attributes are represented as vectors in the benchmark datasets used in the paper. More generally, I don't believe the authors provided sufficient details on how the attributes themselves were constructed for these evaluations. To my knowledge, these benchmark data sets like SIFT and Glove do not come with attributes, so these attributes must have been generated via some process. However, I don't believe these details are currently mentioned in the paper.

2. The authors mention that their technique provides efficiency gains when the "number of distinct attributes is much smaller than the dataset size." I think it would strengthen the paper considerably if the authors could "stress test" their method and consider experimental evaluations where the universe of attributes is large and then compare different techniques. Understanding the practical limitations of this proposed algorithm would be very useful in understanding when it provides concretely efficient gains and when it may not be the right tool.

3. A discussion of the preprocessing time and the construction time of the authors' proposed algorithm would be very helpful. It is not clear from reading the experiments section how long it takes to compute the transformation as well as the time it takes to convert the attributes into vector representations. Comparing methods in terms of construction time would be a very valuable addition to the paper.

4. In the appendix, the authors briefly discuss the limitations of their technique with respect to supporting dynamic updates to the index. I think a deeper discussion into this limitation would also be helpful in understanding when the authors' proposed technique may nor may not be suitable for a given task.

**Questions:**

1. How are the attributes converted into vectors? What neural network models were used for this task in the experimental section of the paper?

2. Do you have a sense of how many distinct attributes is too many for the proposed algorithm to lose effectiveness in practice? Would it be possible to evaluate this experimentally?

3. I do not quite understand the use of the term "Convexified" in the title of this paper beyond the analogy to lagrange multipliers in convex optimization. Is there a deeper application of convexity in the main algorithm proposed in the paper? If not, I would suggest considering changing the title to focus on the main techniques used in the paper.

4. Have you tried to evaluate this algorithm on dynamic datasets with insertions/deletions? Do you have a sense of where the algorithm might break in this setting?

---

> ### Author Response · Authors · 2025-12-02
>
> We thank the reviewer for their helpful feedback. We appreciate their recognition of our ideas and will address the noted issues. With the extra pages, we’ll add clarifications and move key details from the appendix to the main text.
>
> ### **Presentation**
>
> **1. Readability of Figures (Fig. 4 and Fig. 5):** We recognize the figures are small and will enlarge Figures 4 and 5 to full-page width in the 10-page revision.
>
> **2. Theoretical Results in Main Body:** We agree that the theoretical contributions should be more prominent. We will use the extra space to include informal statements of the key theorems in Section 3, specifically Theorem 1 (Order Preservation) and Theorem 4 (Parameter Selection), to help readers immediately grasp the theoretical guarantees without flipping to the appendix.
>
> **3. Attribute Vector Representation:** Thank you for raising this point. As described in §3 (lines 172-175) and Appendix D.1.2, we embed attributes using pre-trained models. Text attributes (e.g., tags) are encoded with BERT and then reduced using PCA (e.g., m=10), mapping discrete values (e.g., “Red,” “Blue”) into a metric space that preserves or distinguishes semantics as appropriate. We will make it more clear in the Model Overview (§3).
>
> ### **Experimental Evaluation**
>
> **1. Construction of Attributes for Benchmark Datasets:** The reviewer correctly notes that SIFT1M and GloVe lack native attributes, so as noted in Appendix D.1.1.1, we followed the standard protocol from Filtered-DiskANN and NHQ (Wang et al., 2023). We added synthetic attributes to mimic real-world selectivity by assigning random labels to subsets for Zipfian or uniform distributions.
>
> **2. Stress Testing (Large Universe of Attributes):** We appreciate this suggestion. Our theoretical complexity analysis in **Section 3** indicates that preprocessing requires $O(|F|^2 N)$ to compute cluster statistics (where $|F|$ is the number of distinct attributes).
> *   **Efficiency Gain Limit:** If $|F|$ approaches $N$ (e.g., unique IDs), the preprocessing overhead increases, and the problem reduces to point lookups.
> *   **Search Effectiveness:** However, search effectiveness does not degrade; **Theorem 2** shows that if $N_a$ (count of records with attribute $a$) is 1, the method simply retrieves the exact match ($k'=k$).
> **Action:** We will add a "Scalability Discussion" in **Section 6** or the Appendix, noting that while search accuracy remains robust as $|F|$ grows (reducing to exact search), the preprocessing time for statistics $O(|F|^2)$ becomes the bottleneck for extremely high-cardinality attributes.
>
> **3. Space & Time Exp:** We have updated Table 8 in Appendix D.7 to explicitly separate preprocessing from index construction time, showing transform overhead is negligible and runtime remains competitive
>
> **4. Dynamic Updates:** We discuss updates in **Section 4.1** and limitations in **Section 7**. The index supports appending new records ($O(N)$) or updating priorities efficiently. In the Discussion, we note that updating a record’s attribute needs delete-insert, and frequent global shifts require recomputing ($R_a, \gamma_a$) to keep $k'$ optimal, limiting use in dynamic settings.
>
> ### **Answers to Specific Questions**
> **Q1:** We used BERT to embed textual attributes (e.g., tags) into high-dimensional vectors, followed by PCA to reduce them to a fixed dimension ($m=10$) to satisfy the $m < d$ constraint. This ensures attributes have semantically meaningful metric relationships before fusion.
>
> **Q2:** The algorithm effectively "breaks" in terms of *preprocessing efficiency* rather than effectiveness. The complexity to compute cluster statistics is $O(|F|^2)$. For millions of distinct attributes, this step becomes slow. However, in terms of retrieval, the method gracefully degrades to exact filtering (where $N_a$ is small) without losing accuracy, as shown in **Theorem 2**. We will add a remark that the method is best suited for cases where $|F|\ll N$.
>
> **Q3:** The term refers to the relaxation of the hard filtering constraint into a convex objective function (a linear combination of norms) via the parameters $\alpha$ and $\beta$, analogous to Lagrangian relaxation. The transformation creates a space where the "hard" attribute constraint becomes a "soft" penalty that convexifies the search landscape, allowing gradient-based or greedy ANN algorithms to function correctly.
>
> **Q4:** As discussed in **Section 4.1**, our method supports updates. Adding new attributes(fields) with high priority requires an additional transformation layer, while lower priority updates require partial reconstruction. For insertions/deletions of records, the underlying ANN index (e.g., HNSW) handles the dynamic changes. The "break point" for our algorithm specifically is when the *distribution* of attributes changes significantly (drifting $\gamma_a$ and $R_a$), which would trigger a need to re-calculate the global statistics for optimal parameter selection.

---

### Official Review · Reviewer_hKWW · 2025-11-08

**Soundness:** 3
**Presentation:** 3
**Contribution:** 3
**Rating:** 6
**Confidence:** 3

**Summary:**

### Paper Summary

This paper introduces a new framework called FUSEDANN to address the important problem of vector search with attribute filtering. Its core idea is to perform a transformation at the data representation level. It introduces a geometric transformation that  preserves nearest-neighbor ordering within attribute classes, supports dynamic attribute priorities, and allows efficient partial index updates. Experiments show that FUSEDANN and its variants comprehensively outperform existing SOTA methods on the QPS-Recall curve across standard hybrid query benchmarks.

**Strengths:**

### Paper Strengths

S1. Transforming the filtering problem into a geometric space transformation is a novel and powerful idea.

S2. The experimental evaluation is comprehensive, covering single-attribute, multi-attribute, and range query scenarios. In all cases, the FUSEDANN variants demonstrate clear superiority on the QPS-Recall tradeoff against all SOTA baselines, including Filtered-DiskANN, NHQ, DEG, and SeRF.

S3. The method is supported by solid theoretical derivations, including proofs for attribute separation, preservation of intra-cluster order, multi-attribute prioritization, and the geometric mapping of range queries.

**Weaknesses:**

### Paper Weaknesses

W1.  The framework cannot natively support query logic. (See D1)

W2. The core transformation seems to require a strict dimensional constraint. (See D2)

W3. The algorithm may introduce extra storage and computational overhead. (See D3)

W4. There are several formatting errors. (See D4)

### Detailed Comments

D1.  The paper admits in the limitations (Section 7, C) that the framework cannot natively support DNF predicates (i.e., "OR" conditions) as this violates metric axioms. While it claims this can be handled via "query planning," this often implies executing multiple queries and merging the results, which can be inefficient in practice.

D2. The core transformation requires attributes to be pre-embedded into a metric space and imposes a strict dimensional constraint, which is inconvenient for common scalar attributes and adds un-discussed complexity.

D3. The algorithm relies on a re-ranking step using original distances, which adds extra storage overhead and computational overhead.

D4. In section 8, the title of conclusion ends with an extra period.

**Questions:**

Please refer to the weakness and detailed comments.

---

> ### Author Response · Authors · 2025-12-02
>
> Thank you for the constructive and thorough review. We appreciate the recognition of our core idea, empirical coverage, and theory. Below we address each weakness and provide concise edit suggestions suitable for the limited-space rebuttal and camera-ready.
>
> **W1/D1. DNF/OR support and efficiency**
> - Principle: We intentionally do not encode OR as a single metric, since that would violate metric axioms. Instead, as stated in the paper, we follow the standard query-optimizer approach: decompose DNF into a small number of shared conjunctive subqueries, evaluate these pre-materialized conjunctive blocks once, and then handle OR/NOT via query planning that merges their results, avoiding repeated full base searches and aligning with classical DBMS predicate compilation and subplan reuse.
>
> - Efficiency in practice: Each disjunct probes shared blocks on the same fused index, followed by a lightweight O(k log k) dedup/merge over at most O(k′) candidates. Candidate sizes are bounded by our analysis and shrink as α increases, with k′→k, so the overhead is close to standard top‑k re‑ranking. The planner uses precomputed per‑attribute statistics Ra, dmin and γa to control k′ and probe effort.
>
> - Range disjunctions: An OR over ranges maps to a union of line segments in the fused space. Our hierarchical line index plus cylindrical search achieves O(log N + k log(1/ε) + k log k) expected time per interval, after which results are merged in O(k log k) over a small k′, ensuring practical efficiency even with multiple intervals.
>
>
> **W2/D2. Dimensional constraint and attribute embeddings**
> - Clarification: The requirement m≤d is natural in practice. Scalars use m=1; categorical/text attributes are embedded and (if required m>d) reduced via PCA. Our implementation already states BERT embeddings with PCA to m=10, which satisfies m≤d across datasets. We also assume m_j|d for exposition; this is not essential—zero‑padding the final block when m_j∤d preserves guarantees and implementation simplicity.
> - Action: We minimally clarified in Implementation Details that scalars use m=1, categorical/text use BERT+PCA to a small m≤d, and that we zero‑pad when $m_j∤d$. We will add a short parenthetical after the $m_j|d$ assumption to note padding as the general case.
>
> **D3. Re‑ranking overhead and storage**
> - Clarification: We do not duplicate base vectors. The index stores the transformed vector and a reference to the original record; re‑ranking follows the pointer and costs $O(k′d)$. Storage is $O(N)$ for vectors plus $O(|F|^2)$ for statistics, and empirically our memory footprint is competitive or smaller than most baselines on 1M‑scale datasets.
> - Practical impact: As α increases, k′→k, so post‑processing is akin to standard ANN re‑ranking.
> - Action: We will append one sentence in Complexity to explicitly state pointer‑only re‑ranking and no duplication.
>
> **W4/D4. Formatting errors**
> - We fixed the extra period in the Conclusion section title and the duplicated phrases in Discussion and some typos.

---

### Meta-Review · Area_Chair_WGS1 · 2026-01-04

**Summary:**

Reviewers raised various concerns regarding limitations of the proposed method, for example inability to support DNF predicates, high update cost, inability to support different filter priorities, and concerns about theoretical claims. The authors made a strong effort to respond to these comments; however, several concerns may not have been fully addressed (e.g., no new experiments were presented to support claims, for updates, for iRangeGraph etc.). Even if we assume 1-2 of the reviewers would raise their scores, the overall score would be within the borderline scores, making it difficult to be accepted this time. I hope the authors will address the comments and re-submit to an appropriate venue.

**Reviewer Concerns:**

Authors attempted to address all comments, however, several of the claims could have been supported by experiments to strengthen their arguments.

**Reviewer Scores:**

The authors did a good job addressed the comments and I could assume 1-2 of the reviewers could have raised their scores. Unfortunately, the overall score would be in the borderline era and hence unlikely to be accepted in the end.

---

### Decision · Program_Chairs · 2026-01-26

Reject